# EDISCO: Equivariant Continuous-Time Categorical Diffusion for Geometric Combinatorial Optimization

## Abstract

Geometric combinatorial optimization problems, such as the Traveling Salesman Problem (TSP), possess inherent symmetries under rotations, translations, and reflections in Euclidean space. These transformations are denoted as E(2). However, existing neural network-based approaches, including recent diffusion-based solvers, fail to exploit these geometric features. This paper presents EDISCO, to the best of our knowledge, the first diffusion-based framework combining E(2)-equivariant graph neural networks with continuous-time categorical diffusion models for solving geometric combinatorial problems. This approach introduces an equivariant score network that respects geometric transformations while operating on discrete edge variables, together with a continuous-time categorical diffusion process that maintains E(2) symmetries throughout the forward and reverse processes. By incorporating geometric awareness directly into the diffusion process, EDISCO achieves notable improvements over the baseline. It reduces the state-of-the-art TSP optimality gaps on TSP-500 from 0.12% to 0.08%, TSP-1000 from 0.30% to 0.22%, and TSP-10000 from 2.68% to 1.20%. EDISCO demonstrates strong generalizability across problem sizes and also shows remarkable efficiency, requiring only 33% to 50% of the training data compared to competing diffusion methods across all problem scales.

## 1 Introduction

With diverse applications in logistics, circuit design, and resource allocation, geometric combinatorial optimization problems (GCOPs), such as the Traveling Salesman Problem (TSP), remain a fundamental challenge in combinatorial optimization. Despite decades of research on exact and heuristic solvers (Applegate et al., 2006; Helsgaun, 2017), the development of learning-based approaches has recently become the focus because of their potential for rapid inference and generalization across various problem instances (Kool et al., 2019; Joshi et al., 2022; Fu et al., 2021). Recent breakthroughs in diffusion models have opened new directions for solving GCOPs (Graikos et al., 2022; Sun & Yang, 2023; Zhao et al., 2024). DIFUSCO (Sun & Yang, 2023) demonstrated graph-based diffusion for TSP, while DISCO (Zhao et al., 2024) introduced residue-constrained generation and analytical denoising to achieve up to 5.28× speedup over previous diffusion approaches.

However, an important observation is that TSP and similar GCOPs possess natural symmetries. The solutions to these problems remain invariant under transformations in 2D Euclidean space, including translations, rotations, and reflections (Ouyang et al., 2021; Bronstein et al., 2021). Such transformations are denoted as E(2). However, most existing neural network-based approaches fail to capture the geometric structure of TSP. These methods require massive amounts of training data and depend on high-quality optimal or near-optimal solutions for supervision, which are computationally expensive to obtain for large problems (Kool et al., 2019; Kwon et al., 2020; Joshi et al., 2022). Furthermore, these models face significant memory and computational constraints. Even moderate-scale problems cause memory overflow and require extensive training times (Bresson & Laurent, 2021; Xin et al., 2021a; Fu et al., 2021). This inefficiency originates from their need to learn geometric invariances from scratch. Non-equivariant models attempt to address this issue with data augmentation, but this only shifts the problem. They require more training samples and still cannot ensure exact equivariance, especially on out-of-distribution data (Nordenfors et al., 2023; Esteves

et al., 2018). In contrast, models that explicitly incorporate geometric structure through equivariant architectures achieve better performance with less training data and smaller model sizes (Brehmer et al., 2024; Satorras et al., 2021; Batzner et al., 2022).

In addition to the geometric considerations, the choice of diffusion formulation presents another crucial design decision. While discrete diffusion models have shown promise for combinatorial problems (Sun & Yang, 2023; Austin et al., 2021), existing approaches employ discrete-time formulations with certain limitations. Although methods like DDIM (Song et al., 2021a) allow variable step counts at inference, they still rely on fixed discretization schemes that may accumulate approximation errors (Zhang & Chen, 2022b; Lu et al., 2022a; Ren et al., 2025). The discrete-time framework also limits access to adaptive numerical solvers that could dynamically adjust computational effort based on local dynamics (Ren et al., 2025; Zhang et al., 2024). Continuous-time formulations fundamentally address these limitations by treating the diffusion as a continuous process governed by stochastic differential equations (SDEs). This enables the use of numerical solvers with adaptive step sizes and higher-order integration methods that can achieve better accuracy with fewer number of function evaluations (NFEs) (Song et al., 2021a; Sun et al., 2023b).

In this work, we propose EDISCO, which leverages an Equivariant Graph Neural Network (EGNN) architecture to respect the geometric symmetries inherent in TSP instances. We also formulate the edge selection process as a continuous-time diffusion over discrete variables, enabling the derivation of analytical expressions for both the forward corruption process and the reverse denoising process, resulting in accelerated inference.

The novel contributions are as follows:

1. We introduce, to the best of our knowledge, the first continuous-time discrete diffusion model with built-in geometric equivariance for combinatorial optimization. Preserving problem symmetries improves both sample efficiency and solution quality.

2. We develop efficient training and sampling algorithms that leverage the analytical tractability of Continuous-time Markov Chains (CTMCs), compatible with higher-order accelerated solvers. These solvers achieve 2-3× speedups with better quality or up to 25× speedups for real-time applications compared to discrete-time methods.

3. The state-of-the-art performance is exceeded on TSP benchmarks (50-10000 cities). The optimality gap is reduced from 0.12% to 0.08% for TSP-500, from 0.30% to 0.22% for TSP-1000, and from 2.68% to 1.20% for TSP-10000. It only requires 33% to 50% of the training data compared to competing diffusion methods across all problem scales.

4. EDISCO is extended to solve real-world TSP problems and is extended to other GCOPs, including the Capacitated Vehicle Routing Problem (CVRP) and Euclidean Steiner Tree Problem (ESTP). The results show that EDISCO outperforms SOTA approaches. These results validate the generalizability of EDISCO, indicating that it can be effectively applied to other GCOPs.

The remainder of this paper is organized as follows. Section 2 reviews related work on neural combinatorial optimization and diffusion models. Section 3 presents the continuous-time diffusion framework and the EGNN architecture. Section 4 presents comprehensive experimental results. Section 5 concludes with discussions and future directions.

## 2 RELATED WORK

### 2.1 NEURAL NETWORK-BASED TSP SOLVERS

Neural network-based approaches for TSP have recently become mainstream due to their potential for rapid inference, generalization across problem instances, and ability to learn from data without hand-crafted heuristics (Kool et al., 2019; Joshi et al., 2022; Fu et al., 2021). These approaches can be divided into autoregressive and non-autoregressive models. Autoregressive models (Kool et al., 2019; Kwon et al., 2020) construct solutions sequentially but require extensive training data and struggle with generalization (Joshi et al., 2022). Non-autoregressive approaches generate complete solutions simultaneously, evolving from limited heatmap representations (Joshi et al., 2019)

to expressive diffusion models (Sun & Yang, 2023; Li et al., 2023; Yoon et al., 2024; Zhao et al., 2024).

Among diffusion approaches, DIFUSCO (Sun & Yang, 2023) pioneered graph-based diffusion for TSP, DISCO (Zhao et al., 2024) achieved 5.28× speedup through residue-constrained generation, and T2T (Li et al., 2023) improved quality via gradient-based search. Fast-T2T (Li et al., 2024) accelerates diffusion-based solutions through optimization consistency between training and testing, achieving substantial speedup with competitive solution quality. CADO (Yoon et al., 2024) combines enhanced DIFUSCO by combining RL fine-tuning, but requires high-quality supervised data and expensive RL fine-tuning. Recent work explores alternative paradigms: COExpander (Ma et al., 2025) introduces adaptive expansion that interpolates between global prediction and local construction, achieving strong performance on TSP and ATSP up to 10K nodes; BQ-NCO (Drakulic et al., 2023) reformulates the problem as an MDP with bisimulation quotienting for improved generalization; and UTSP (Min et al., 2023) demonstrates that unsupervised learning with surrogate losses can achieve competitive TSP performance using only 0.2% of typical training data. The key insight is that all existing diffusion TSP solvers use discrete-time diffusion formulations and ignore TSP's geometric structure.

## 2.2 Geometric Deep Learning and Equivariance

Geometric deep learning leverages symmetries to improve efficiency and generalization (Bronstein et al., 2021). Equivariant neural networks ensure outputs transform consistently with symmetric inputs, reducing sample complexity (Cohen & Welling, 2016). E(2)-equivariant models significantly improve TSP generalization (Ouyang et al., 2021), and geometric GNNs outperform standard architectures for TSP tasks (Song et al., 2025). Theory confirms that equivariant models reduce training data requirements (Brehmer et al., 2024). Despite this evidence, most neural network-based TSP solvers lack geometric awareness. While Sym-NCO (Kim et al., 2022) uses regularizer-based symmetry learning, it doesn't achieve exact equivariance. These gaps motivate our approach as the first to combine exact E(2)-equivariance with diffusion.

## 2.3 Continuous-Time Diffusion and Sampling Methods

Continuous-time formulations resolve fundamental limitations of discrete-time diffusion. CTMCs for discrete diffusion denoising (Campbell et al., 2022) enable analytical transition probabilities and flexible inference without retraining. Score-based continuous-time discrete diffusion (Sun et al., 2023c) provides better convergence properties and allows the use of higher-order integration methods that significantly reduce the number of neural network evaluations (Song et al., 2021b). Though DiffUCO (Sanokowski et al., 2024) applies continuous-time diffusion to unsupervised combinatorial optimization on graph problems (MIS, clique, max-cut), it lacks geometric considerations and does not show results on large-scale Euclidean TSP/CVRP problems. DISCO (Zhao et al., 2024) also achieves fast inference (1-2 steps) by replacing the entire numerical integration process with an analytically solvable form through decoupled diffusion models (DDMs). This analytical solution completely bypasses the need for numerical ODE solvers but requires problem-specific residue constraints and sacrifices flexibility in the diffusion process.

Beyond diffusion, sampling-based approaches offer alternative perspectives on combinatorial optimization. iSCO (Sun et al., 2023a) revisits MCMC sampling with expensive 2-opt moves, achieving high-quality solutions on large TSPs but at significantly longer runtimes (tens of GPU-hours). RLSA (Feng & Yang, 2025) employs regularized Langevin dynamics with neural networks to improve exploration on graph CO problems (MIS, clique, max-cut), achieving 80% faster convergence than simulated annealing. Variational approaches have also shown promise: VAG-CO (Sanokowski et al., 2023) learns Boltzmann distributions on Ising-formulated graph optimization tasks (MIS, MVC, MaxCut), while GFlowNets (Zhang et al., 2023) demonstrates strong performance on graph CO through flow-based generative models. These methods target non-geometric graph problems and do not directly address Euclidean TSP/CVRP.

## 3 METHOD

### 3.1 PROBLEM FORMULATION

GCOPs in Euclidean space are defined on a set of $n$ nodes $\mathcal{V}$ with coordinates $\{\mathbf{c}_i\}_{i=1}^n$, $\mathbf{c}_i \in \mathbb{R}^d$. The objective is to select a subset of edges or configurations, represented by a decision matrix $\mathbf{X} \in \{0,1\}^{n \times n}$, that minimizes a distance-based cost function while satisfying problem-specific constraints. The objective is:

$$\mathbf{X}^* = \arg \min_{\mathbf{X}} f(\mathbf{X}, \{\mathbf{c}_i\}_{i=1}^n) \quad \text{s.t. } \mathbf{X} \in \mathcal{C} \tag{1}$$

where $f$ is a distance-based cost function and $\mathcal{C}$ represents the constraint.

In the case of TSP, given $n$ cities with coordinates $\mathbf{c}_i \in \mathbb{R}^2$, we need to find a binary adjacency matrix $\mathbf{X} \in \{0,1\}^{n \times n}$ where $X_{ij} = 1$ if edge $(i,j)$ is included in the tour. The tour constraints are: each city has degree 2, and the selected edges form a connected cycle. We formulate this as a generative modeling problem Sun & Yang (2023); Li et al. (2023), learning the conditional distribution $p(\mathbf{X}|\{\mathbf{c}_i\}_{i=1}^n)$.

### 3.2 CONTINUOUS-TIME CATEGORICAL DIFFUSION FRAMEWORK

Unlike continuous diffusion models that operate in Euclidean space and require post-hoc quantization, categorical diffusion directly models discrete decisions in their native space (Austin et al., 2021). This design choice eliminates quantization errors and ensures the model learns the true discrete distribution rather than a continuous approximation. DIFUSCO (Sun & Yang, 2023) also demonstrated that categorical diffusion consistently outperforms continuous diffusion on TSP over all problem sizes.

Additionally, the continuous-time formulation offers extra benefits over discrete-time diffusion, as it enables exact likelihood computation, allows for flexible inference schedules without requiring retraining, and provides better theoretical guarantees for convergence (Campbell et al., 2022).

**Forward Process** The forward process defines how clean data $\mathbf{X}_0$ progressively transitions to noise through a continuous-time Markov chain (CTMC). For K-state categorical variables, the instantaneous rate of transition between states is governed by the rate matrix (Campbell et al., 2022):

$$\mathbf{Q}(t) = \beta(t) \left( \frac{1}{K} \mathbb{1}\mathbb{1}^T - \mathbf{I} \right) \tag{2}$$

where $\beta(t) = \beta_{\min} + t(\beta_{\max} - \beta_{\min})$ is a linear noise schedule with $t \in [0,1]$. We set $\beta_{\min} = 0.1$ and $\beta_{\max} = 1.5$.

The transition probability from time $s$ to $t$ is obtained by solving the Kolmogorov forward equation (Norris, 1997), yielding the closed-form solution (Campbell et al., 2022):

$$P_{ij}(t|s) = \frac{1}{K} + \left( \delta_{ij} - \frac{1}{K} \right) \exp \left( -K \int_s^t \beta(u)du \right) \tag{3}$$

For TSP with binary edge selection (K=2), this allows us to directly sample the noisy state $\mathbf{X}_t$ from the clean data $\mathbf{X}_0$:

$$P(\mathbf{X}_t = j | \mathbf{X}_0 = i) = P_{ij}(t|0) = \frac{1}{2} + \left( \delta_{ij} - \frac{1}{2} \right) \exp \left( -2 \int_0^t \beta(u)du \right) \tag{4}$$

For our linear schedule, the integral evaluates analytically to:

$$\int_0^t \beta(u)du = \beta_{\min}t + \frac{1}{2}(\beta_{\max} - \beta_{\min})t^2 \tag{5}$$

This closed-form expression enables exact sampling of $\mathbf{X}_t$ given $\mathbf{X}_0$ at any time $t$ without simulating intermediate states, crucial for efficient training. The exponential decay term ensures that as $t \to 1$, the transition probability approaches uniform ($P_{ij} \to 1/2$), completely corrupting the original signal while maintaining mathematical tractability.

Figure 1: EDISCO's EGNN Architecture Overview. EGNN layers process TSP instances while preserving $E(2)$ equivariance. The network outputs edge probabilities that remain invariant under geometric transformations.

**Reverse Process** The reverse process reconstructs clean data from noise by iteratively applying learned denoising steps. The key insight is that while the forward process is fixed and tractable, the reverse process requires learning the score function, which is the gradient of the log probability density. Using Bayes' rule, the posterior distribution for the reverse transition is:

$$q(\mathbf{X}_{t-\Delta t}|\mathbf{X}_t, \mathbf{X}_0) = \frac{q(\mathbf{X}_t|\mathbf{X}_{t-\Delta t}, \mathbf{X}_0)q(\mathbf{X}_{t-\Delta t}|\mathbf{X}_0)}{q(\mathbf{X}_t|\mathbf{X}_0)} \tag{6}$$

Since the true $\mathbf{X}_0$ is unknown during inference, we need a neural network $s_\theta(\mathbf{X}_t, t, \{\mathbf{c}_i\})$ to predict it when given the noisy state and time. This parameterization, known as $x_0$-prediction, is more stable than alternative parameterizations like noise prediction, especially in the low-noise region where reconstruction accuracy is significant (Salimans & Ho, 2022).

In EDISCO, we use an adaptive mixing strategy that dynamically balances between diffusion-based transitions and direct model predictions:

$$p_{\text{reverse}} = w(t) \cdot p_{\text{diffusion}} + (1 - w(t)) \cdot p_{\text{predicted}} \tag{7}$$

where $w(t) = t$ linearly decreases from 1 to 0 as the reverse process progresses. This design is grounded in both empirical observations and theoretical principles. Empirically, early in the reverse process (large $t$), the noisy state contains little information about the target, making the diffusion dynamics essential for exploration. As $t$ decreases and the signal emerges, direct predictions become increasingly reliable and dominating to ensure precise reconstruction. Theoretically, reverse diffusion exhibits a monotonically increasing signal-to-noise ratio (SNR) as time decreases (Ho et al., 2020; Song et al., 2021b). The linear schedule provides the simplest monotonic interpolation that aligns with this SNR progression, consistent with standard practice in diffusion models (Nichol & Dhariwal, 2021). For very small timesteps where $t < 0.1$ or $|\Delta t| < 0.02$, we switch entirely to deterministic transitions using the argmax of predicted probabilities. This strategy is comprehensively evaluated in Appendix H.6.

### 3.3 EQUIVARIANT GRAPH NEURAL NETWORK ARCHITECTURE

**Geometric Equivariance for TSP** TSP possesses an inherent geometric structure that should be preserved. The E(2) invariance of TSP solution has been recognized in prior work (Ouyang et al., 2021; Kim et al., 2022). If $\mathbf{X}^*$ is optimal for cities $\{\mathbf{c}_i\}$, then $\mathbf{X}^*$ remains optimal for transformed cities $\{g(\mathbf{c}_i)\}$ where $g \in E(2)$ represents any combinations of rotations, reflections, and translations. Traditional neural networks would need to learn this invariance from data, requiring extensive data augmentation and larger model capacity. However, we build equivariance directly into the architecture of EDISCO, ensuring that geometric transformations of inputs produce corresponding transformations of internal representations, so that the output can be maintained invariant (Thomas et al., 2018).

**EGNN Layers with Stability Mechanisms** We adapt the E(n)-equivariant graph neural network (Satorras et al., 2021) with several crucial modifications for stable training on GCOPs. The architecture maintains three types of features: node features $\mathbf{h}_i$ encoding local city information,

edge features $\mathbf{e}_{ij}$ representing pairwise relationships and tour decisions, and coordinate embeddings $\mathbf{x}_i$ that evolve during message passing to capture geometric features. Figure 1 illustrates the EGNN architecture that maintains E(2)-equivariance throughout the message passing process. The architecture processes TSP instances through multiple layers that preserve geometric symmetries while learning to predict edge probabilities for tour construction.

The message computation aggregates information from node pairs and their geometric relationship:

$$\mathbf{m}_{ij}^{(\ell)} = \text{MLP}_m\left([\mathbf{h}_i^{(\ell)}, \mathbf{h}_j^{(\ell)}, \mathbf{e}_{ij}^{(\ell)}, \|\mathbf{x}_i^{(\ell)} - \mathbf{x}_j^{(\ell)}\|_2]\right) \tag{8}$$

The inclusion of pairwise distances as scalar features, which are invariant under E(2), allows the model to reason about geometric relationships without breaking equivariance.

Coordinate updates must preserve equivariance, achieved through the constrained form:

$$\Delta\mathbf{x}_i = \alpha \sum_{j \neq i} w_{ij} \cdot \frac{\mathbf{x}_j^{(\ell)} - \mathbf{x}_i^{(\ell)}}{\|\mathbf{x}_j^{(\ell)} - \mathbf{x}_i^{(\ell)}\|_2} \tag{9}$$

where the weights $w_{ij} = \tanh(\text{MLP}_c(\mathbf{m}_{ij}^{(\ell)})/\tau)$ control the influence of each neighbor. The temperature parameter $\tau = 10$ prevents saturation of the tanh function during early training when the MLP outputs may be large. The conservative step size $\alpha = 0.1$ is critical: larger values lead to coordinate collapse, where all cities converge to a single point. However, smaller values limit the model's ability to learn useful geometric features. The normalization by distance ensures that the update magnitude is independent of the coordinate scale, therefore improving robustness. The selections of $\tau$ and $\alpha$ are extensively evaluated in Appendix H.7.

Edge features are updated with explicit time conditioning:

$$\mathbf{e}_{ij}^{(\ell+1)} = \text{LayerNorm}(\mathbf{e}_{ij}^{(\ell)} + \text{MLP}_e([\mathbf{e}_{ij}^{(\ell)}, \mathbf{m}_{ij}^{(\ell)}]) + \text{MLP}_t(\mathbf{t}_{\text{emb}})) \tag{10}$$

The time embedding $\mathbf{t}_{\text{emb}}$ uses sinusoidal encoding (Vaswani et al., 2017). This enables the network to distinguish between fine-grained time differences near $t = 0$ and coarser differences at high noise levels.

Node features aggregate information from neighbors with gated attention:

$$\mathbf{h}_i^{(\ell+1)} = \text{LayerNorm}(\mathbf{h}_i^{(\ell)} + \text{MLP}_h([\mathbf{h}_i^{(\ell)}, \sum_{j \neq i} \sigma(\mathbf{m}_{ij}^{(\ell)}) \odot \mathbf{h}_j^{(\ell)}])) \tag{11}$$

Through the EGNN layers, EDISCO maintains exact E(2)-equivariance throughout the entire diffusion process. This is crucial for TSP because it reduces the effective complexity of the function to be learned.

**Proposition 1.** *Let $X = \mathbb{R}^{2n}$ denote the space of ordered 2D coordinates for $n$ cities, and let $G = \text{E}(2)$ be the Euclidean transformation group on $X$ by simultaneous rotation and translation of all city positions. Assume the transformation is free on the dataset (i.e., no non-trivial element $g \in G \setminus \{e\}$ fixes any configuration). Then:*

*(i) The quotient space $X/G$ is a smooth manifold of dimension $2n - 3$.*

*(ii) Any $G$-equivariant function $F : X \to Y$ factors uniquely through the quotient as $F = \widetilde{F} \circ \pi$, where $\pi : X \to X/G$ is the canonical projection and $\widetilde{F} : X/G \to Y$ is a function on the quotient manifold.*

*(iii) Learning a $G$-equivariant function is equivalent to learning a function on the $(2n - 3)$-dimensional manifold $X/G$ rather than on $\mathbb{R}^{2n}$.*

All the details about the notations can be found in the Default Notation. Although the dimension reduction from $2n$ to $2n - 3$ appears modest, its impact on learning is substantial. Equivariance forces the model to operate on the $(2n - 3)$-dimensional orbit space instead of $\mathbb{R}^{2n}$, which reduces the metric entropy and the effective hypothesis-class complexity. The sample complexity reduction scales as $(1/\varepsilon)^3$ in covering number bounds, where $\varepsilon$ is the desired approximation accuracy. We

do not consider E(2) reflections because reflections are discrete transformations that do not further reduce the quotient dimension. For TSP, reflected tours are equivalent (same edges, only opposite tour sequence). The detailed proof of Proposition 1 is given in Appendix D.1. We also prove that the E(2)-Equivariance is preserved during the entire diffusion process in Appendix D.2.

### 3.4 INFERENCE AND TOUR DECODING

**Solver Selection**    During inference, the continuous-time formulation allows for a flexible choice of accelerated and higher-order solvers without requiring retraining (Campbell et al., 2022). We extensively evaluate different solvers in Appendix H.1.

**Tour Construction from Edge Probabilities**    The diffusion model outputs a probability matrix $P \in [0,1]^{n \times n}$ where $P_{ij} = p(X_{ij} = 1)$ represents the model's confidence that edge $(i,j)$ should be included in the optimal tour. Converting these soft probabilities to a valid discrete tour requires careful consideration of both model confidence and problem constraints.

Following the greedy decoding strategy from (Sun & Yang, 2023), we compute edge scores that balance model predictions with distance-based priors as $s_{ij} = (P_{ij} + P_{ji})/d_{ij}$, where the symmetrization $P_{ij} + P_{ji}$ accounts for TSP's undirected nature and $d_{ij}$ denotes the Euclidean distance between nodes $i$ and $j$.

The greedy construction algorithm maintains feasibility throughout the process. Starting with an empty tour, edges are processed in descending score order. An edge $(i,j)$ is added if and only if both vertices have degree less than 2 (ensuring no city is visited more than twice) and adding the edge would not create a subtour (except for the final edge that completes the Hamiltonian cycle). Cycle detection is performed efficiently using a union-find data structure with path compression, achieving near-linear time complexity. The 2-opt local search (Lin & Kernighan, 1973) post-processing can be optionally applied to improve the tour.

## 4 EXPERIMENTS

### 4.1 SETUP

**Datasets**    We follow the standard TSP evaluation protocol from Kool et al. (2019). Training instances are generated by sampling $n$ cities uniformly from the unit square $[0,1]^2$. We used the Concorde exact solver (Applegate et al., 2006) to generate datasets for TSP-50 and TSP-100, and used the LKH-3 (Helsgaun, 2017) heuristic solver for TSP-500 and TSP-1000. For evaluation, we use the standard test sets from Kool et al. (2019) for TSP-50/100 and Fu et al. (2021) for TSP-500 and above. EDISCO only requires 33% to 50% of the training data compared to baseline methods across all problem scales, which is reported in Appendix G.6.

**Graph Representation**    For TSP-50/100, we use dense adjacency matrices representing complete graphs. For better scalability and fair comparisons, we apply graph sparsification to TSP-500 and above following the configuration of (Sun & Yang, 2023), with details displayed in Appendix G.4.

**Evaluation Metrics**    We report three primary metrics: (1) average tour length, (2) average optimality gap, and (3) total run time. More experiment details can be found in Appendix G.1.

### 4.2 RESULTS

This section aims to show the main results of TSP. The extension to ESTP and CVRP can be found in Appendix B and Appendix C. For all EDISCO results shown in this section, we use the PNDM solver (Liu et al., 2022) with 50 steps, which achieves the best solution quality based on our extensive evaluation in Appendix H.1.

**TSP-50/100 Results**    The results are shown in Table 1. EDISCO achieves near-optimal performance with 0.01% gap on TSP-50 and 0.04% on TSP-100, substantially outperforming DIFUSCO (0.48% and 1.01%), Fast-T2T (0.02% and 0.07%), BQ-transformer (– and 0.35%), and T2T (0.04%

Table 1: Results on TSP-50 and TSP-100. RL: Reinforcement Learning, SL: Supervised Learning, G: Greedy Decoding, BS: Beam Search, 2O: 2-opt Post-processing. Concorde* represents the baseline for computing the gap. All results except CADO, Fast T2T, and BQ-NCO variants are taken from Li et al. (2023). The results of CADO are taken from Yoon et al. (2024). Fast T2T results are from (Li et al., 2024). BQ-NCO results (BQ-perceiver, BQ-transformer, BQ-transformer bs16) are from (Drakulic et al., 2023). EDISCO presents two solver configurations: 50-step PNDM for best quality and 5-step DEIS-2 for fast inference.

| Algorithm | Type | TSP-50 | | TSP-100 | |
|---|---|---|---|---|---|
| | | Length ↓ | Gap ↓ | Length ↓ | Gap ↓ |
| Concorde* (Applegate et al., 2006) | Exact | 5.69 | 0.00 | 7.76 | 0.00 |
| 2-opt (Lin & Kernighan, 1973) | Heuristic | 5.86 | 2.95 | 8.03 | 3.54 |
| AM (Kool et al., 2019) | RL+G | 5.80 | 1.76 | 8.12 | 4.53 |
| GCN (Joshi et al., 2019) | SL+G | 5.87 | 3.10 | 8.41 | 8.38 |
| Transformer (Bresson & Laurent, 2021) | RL+G | 5.71 | 0.31 | 7.88 | 1.42 |
| POMO (Kwon et al., 2020) | RL+G | 5.73 | 0.64 | 7.87 | 1.07 |
| Sym-NCO (Kim et al., 2022) | RL+G | - | - | 7.84 | 0.94 |
| Image Diffusion (Graikos et al., 2022) | SL+G | 5.76 | 1.23 | 7.92 | 2.11 |
| DIFUSCO (Sun & Yang, 2023) | SL+G | 5.72 | 0.48 | 7.84 | 1.01 |
| T2T (Li et al., 2023) | SL+G | 5.69 | 0.04 | 7.77 | 0.18 |
| Fast T2T (Ts=5) (Li et al., 2024) | SL+G | 5.69 | 0.02 | 7.76 | 0.07 |
| BQ-perceiver (Drakulic et al., 2023) | SL+G | – | – | 7.84 | 0.97 |
| BQ-transformer (Drakulic et al., 2023) | SL+G | – | – | 7.79 | 0.35 |
| CADO (Yoon et al., 2024) | SL+RL+G | 5.69 | 0.01 | 7.77 | 0.08 |
| **EDISCO with 50-step PNDM (ours)** | SL+G | **5.69** | **0.01** | **7.76** | **0.04** |
| **EDISCO with 5-step DEIS-2 (ours)** | SL+G | **5.69** | **0.02** | **7.76** | **0.06** |
| AM (Kool et al., 2019) | RL+G+2O | 5.77 | 1.41 | 8.02 | 3.32 |
| GCN (Joshi et al., 2019) | SL+G+2O | 5.70 | 0.12 | 7.81 | 0.62 |
| Transformer (Bresson & Laurent, 2021) | RL+G+2O | 5.70 | 0.16 | 7.85 | 1.19 |
| POMO (Kwon et al., 2020) | RL+G+2O | 5.73 | 0.63 | 7.82 | 0.82 |
| Sym-NCO (Kim et al., 2022) | RL+G+2O | - | - | 7.82 | 0.76 |
| DIFUSCO (Sun & Yang, 2023) | SL+G+2O | 5.69 | 0.09 | 7.78 | 0.22 |
| T2T (Li et al., 2023) | SL+G+2O | 5.69 | 0.02 | 7.76 | 0.06 |
| Fast T2T (Ts=3,Tg=3) (Li et al., 2024) | SL+G+2O | 5.69 | 0.01 | 7.76 | 0.03 |
| BQ-transformer (bs16) (Drakulic et al., 2023) | SL+BS | – | – | 7.76 | 0.01 |
| CADO (Yoon et al., 2024) | SL+RL+G+2O | 5.69 | 0.00 | 7.76 | 0.01 |
| **EDISCO with 50-step PNDM (ours)** | SL+G+2O | **5.69** | **0.00** | **7.76** | **0.01** |
| **EDISCO with 5-step DEIS-2 (ours)** | SL+G+2O | **5.69** | **0.01** | **7.76** | **0.02** |

and 0.18%). Unlike CADO, which requires both supervised and reinforcement learning, our purely supervised approach reaches comparable accuracy. For applications requiring ultra-fast inference, EDISCO with a 5-step DEIS-2 solver achieves 0.02% and 0.06% gaps on TSP-50 and TSP-100, respectively, showing competitive performance with significantly reduced computation. With 2-opt post-processing, EDISCO achieves optimal solutions (0.00% gap) on TSP-50 and 0.01% gap on TSP-100, matching the performance of Fast-T2T and CADO while demonstrating that the equivariant architecture generates high-quality initial tours.

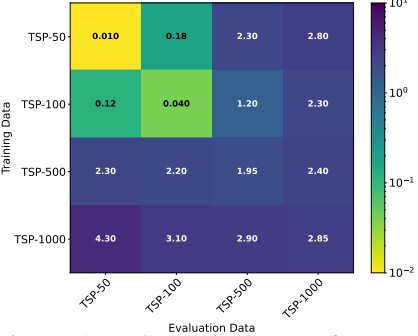

Figure 2: Generalization performance across TSP sizes under greedy decoding.

**TSP-500/1000 Results** Table 2 presents results on larger-scale TSP instances. EDISCO achieves state-of-the-art performance across all decoding strategies. Using greedy decoding, EDISCO achieves gaps of 1.95% and 2.85% on TSP-500 and TSP-1000, outperforming DIFUSCO (9.41%, 11.24%), Fast-T2T (5.94%, 6.29%), BQ-transformer (1.18%, 2.29%), and T2T (5.09%, 8.87%). BQ-NCO, which uses an MDP formulation with imitation learning, achieves competitive generalization but is still outperformed by EDISCO's diffusion-based approach. The continuous-time formulation enables flexible speed-quality trade-offs: EDISCO with a 5-step DEIS-2 solver achieves 2.78% and 4.42% gaps in

Table 2: Results on TSP-500 and TSP-1000. RL: Reinforcement Learning, SL: Supervised Learning, AS: Active Search, G: Greedy, S: Sampling, BS: Beam Search, 2O: 2-opt. Concorde* represents the baseline for computing the gap. All results except CADO, Fast T2T, BQ-NCO variants, and EDISCO are taken from Li et al. (2023). CADO results are from Yoon et al. (2024). Fast T2T results are from (Li et al., 2024). BQ-NCO results (BQ-perceiver, BQ-transformer, BQ-transformer bs16) are from (Drakulic et al., 2023). EDISCO presents two solver configurations: 50-step PNDM (best quality) and 5-step DEIS-2 (9× faster, competitive quality).

| Algorithm | Type | TSP-500 | | | TSP-1000 | | |
|---|---|---|---|---|---|---|---|
| | | Length↓ | Gap↓ | Time | Length↓ | Gap↓ | Time |
| Concorde* (Applegate et al., 2006) | Exact | 16.55 | – | 37.66 m | 23.12 | – | 6.65 h |
| Gurobi (Gurobi Optimization, LLC, 2020) | Exact | 16.55 | 0.00% | 45.63 h | – | – | – |
| LKH-3 (default) (Helsgaun, 2017) | Heuristics | 16.55 | 0.00% | 46.28 m | 23.12 | 0.00% | 2.57 h |
| AM (Kool et al., 2019) | RL+G | 20.02 | 20.99% | 1.51 m | 31.15 | 34.75% | 3.18 m |
| GCN (Joshi et al., 2019) | SL+G | 29.72 | 79.61% | 6.67 m | 48.62 | 110.29% | 28.52 m |
| POMO+EAS-Emb (Kwon et al., 2020) | RL+AS+G | 19.24 | 16.25% | 12.80 h | – | – | – |
| POMO+EAS-Tab (Kwon et al., 2020) | RL+AS+G | 24.54 | 48.22% | 11.61 h | 49.56 | 114.36% | 63.45 h |
| DIMES (Qiu et al., 2022) | RL+G | 18.93 | 14.38% | 0.97 m | 26.58 | 14.97% | 2.08 m |
| DIMES (Qiu et al., 2022) | RL+AS+G | 17.81 | 7.61% | 2.10 h | 24.91 | 7.74% | 4.49 h |
| DIFUSCO (Sun & Yang, 2023) | SL+G | 18.11 | 9.41% | 5.70 m | 25.72 | 11.24% | 17.33 m |
| T2T (Li et al., 2023) | SL+G | 17.39 | 5.09% | 4.90 m | 25.17 | 8.87% | 15.66 m |
| Fast T2T (Ts=5) (Li et al., 2024) | SL+G | 17.53 | 5.94% | 0.37 m | 24.57 | 6.29% | 1.35 m |
| BQ-perceiver (Drakulic et al., 2023) | SL+G | 17.41 | 5.22% | 0.13 m | 25.19 | 8.97% | 0.37 m |
| CADO (Yoon et al., 2024) | SL+RL+G | 16.93 | 2.30% | 8.23 m | 23.89 | 3.33% | 18.42 m |
| **EDISCO with 50-step PNDM (ours)** | SL+G | **16.87** | **1.95%** | 2.19 m | **23.78** | **2.85%** | 6.84 m |
| **EDISCO with 5-step DEIS-2 (ours)** | SL+G | **17.01** | **2.78%** | **0.23 m** | **24.83** | **4.42%** | **0.75 m** |
| DIMES (Qiu et al., 2022) | RL+G+2O | 17.65 | 6.62% | 1.01 m | 24.83 | 7.38% | 2.29 m |
| DIMES (Qiu et al., 2022) | RL+AS+G+2O | 17.31 | 4.57% | 2.10 h | 24.33 | 5.22% | 4.49 h |
| DIFUSCO (Sun & Yang, 2023) | SL+G+2O | 16.81 | 1.55% | 5.75 m | 23.55 | 1.86% | 17.52 m |
| T2T (Li et al., 2023) | SL+G+2O | 16.68 | 0.78% | 4.98 m | 23.41 | 1.25% | 15.90 m |
| Fast T2T (Ts=5,Tg=5) (Li et al., 2024) | SL+G+2O | 16.61 | 0.39% | 2.17 m | 23.25 | 0.58% | 8.62 m |
| BQ-transformer (Drakulic et al., 2023) | SL+G | 16.75 | 1.18% | 0.25 m | 23.65 | 2.29% | 0.50 m |
| CADO (Yoon et al., 2024) | SL+RL+G+2O | 16.59 | 0.24% | 8.35 m | 23.28 | 0.69% | 18.67 m |
| **EDISCO with 50-step PNDM (ours)** | SL+G+2O | **16.58** | **0.18%** | 2.35 m | **23.24** | **0.52%** | 6.97 m |
| **EDISCO with 5-step DEIS-2 (ours)** | SL+G+2O | **16.59** | **0.26%** | **0.40 m** | **23.31** | **0.82%** | **0.90 m** |
| EAN (Deudon et al., 2018) | RL+S+2O | 23.75 | 43.57% | 57.76 m | 47.73 | 106.46% | 5.39 h |
| AM (Kool et al., 2019) | RL+BS | 19.53 | 18.03% | 21.99 m | 29.90 | 29.23% | 1.64 h |
| GCN (Joshi et al., 2019) | SL+BS | 30.37 | 83.55% | 38.02 m | 51.26 | 121.73% | 51.67 m |
| DIMES (Qiu et al., 2022) | RL+S | 18.84 | 13.84% | 1.06 m | 26.36 | 14.01% | 2.38 m |
| DIMES (Qiu et al., 2022) | RL+AS+S | 17.80 | 7.55% | 2.11 h | 24.89 | 7.70% | 4.53 h |
| DIFUSCO (Sun & Yang, 2023) | SL+S | 17.48 | 5.65% | 19.02 m | 25.11 | 8.61% | 59.18 m |
| T2T (Li et al., 2023) | SL+S | 17.02 | 2.84% | 15.98 m | 24.72 | 6.92% | 53.92 m |
| Fast T2T (Ts=5) (Li et al., 2024) | SL+S | 17.02 | 2.85% | 1.12 m | 24.07 | 4.10% | 4.65 m |
| CADO (Yoon et al., 2024) | SL+RL+S | 16.76 | 1.27% | 26.89 m | 23.67 | 2.38% | 61.23 m |
| **EDISCO with 50-step PNDM (ours)** | SL+S | **16.72** | **1.05%** | 7.82 m | **23.57** | **1.95%** | 23.27 m |
| **EDISCO with 5-step DEIS-2 (ours)** | SL+S | **16.80** | **1.50%** | **0.85 m** | **23.82** | **3.02%** | **2.58 m** |
| DIMES (Qiu et al., 2022) | RL+S+2O | 17.64 | 6.56% | 1.10 m | 24.81 | 7.29% | 2.86 m |
| DIMES (Qiu et al., 2022) | RL+AS+S+2O | 17.29 | 4.48% | 2.11 h | 24.32 | 5.17% | 4.53 h |
| DIFUSCO (Sun & Yang, 2023) | SL+S+2O | 16.69 | 0.83% | 19.05 m | 23.42 | 1.30% | 59.53 m |
| T2T (Li et al., 2023) | SL+S+2O | 16.61 | 0.37% | 16.03 m | 23.30 | 0.78% | 54.67 m |
| Fast T2T (Ts=5,Tg=5) (Li et al., 2024) | SL+S+2O | 16.58 | 0.21% | 6.85 m | 23.22 | 0.42% | 18.28 m |
| BQ-transformer (bs16) (Drakulic et al., 2023) | SL+S+2O | 16.57 | 0.15% | 18.00 m | 23.20 | 0.35% | 42.00 m |
| CADO (Yoon et al., 2024) | SL+RL+S+2O | 16.57 | 0.12% | 27.01 m | 23.19 | 0.30% | 61.48 m |
| **EDISCO with 50-step PNDM (ours)** | SL+S+2O | **16.56** | **0.08%** | 8.03 m | **23.17** | **0.22%** | 23.48 m |
| **EDISCO with 5-step DEIS-2 (ours)** | SL+S+2O | **16.57** | **0.12%** | **1.00 m** | **23.20** | **0.35%** | **2.80 m** |

only 0.23 minutes and 0.75 minutes, approximately 9.5× and 9.1× faster than PNDM-50 while maintaining competitive quality, making it ideal for time-critical applications. With 2-opt post-processing, EDISCO achieves near-optimal solutions with gaps of 0.18% and 0.52%, outperforming Fast-T2T (0.39%, 0.58%) and the previous best CADO (0.24%, 0.69%) while being 3.6× faster on TSP-500 and 2.7× faster on TSP-1000. The efficiency gain is particularly notable in sampling mode, where EDISCO requires only 7.82m and 23.27m compared to DIFUSCO's 19.02m and 59.18m, demonstrating that the use of advanced numerical solvers significantly accelerates inference without compromising solution quality.

**Generalization** We study the generalization ability of EDISCO by training models on each problem scale from {TSP-50, TSP-100, TSP-500, TSP-1000} and evaluating them across all scales with only the greedy decoder. Figure 2 shows that EDISCO exhibits strong cross-size generalization, with models trained on TSP-1000 achieving gaps below 4.3% on all other problem scales, and par-

ticularly impressive performance of 2.90% on TSP-500. This generalizability outperforms other diffusion methods (Sun & Yang, 2023; Li et al., 2023).

**Robustness to Training Data Variations**
We evaluate EDISCO's robustness to variations in training data quantity and quality, which are critical factors for practical deployment where obtaining optimal solutions may be computationally expensive. The left panel of Figure 3 illustrates that EDISCO maintains near-optimal performance even with limited data, achieving gaps below 0.07% with just 10% of training data, compared to 2.8% for DIFUSCO and 2.1% for T2T.

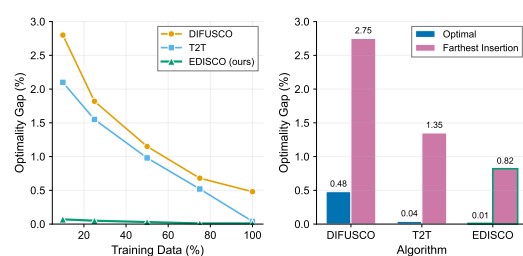

Figure 3: Results on TSP-50 performance. Left: Optimality gap as a function of training set size. Right: Performance comparison when trained on optimal data versus heuristic Farthest Insertion data.

The right panel of Figure 3 examines model performance when trained on suboptimal solutions generated by the Farthest Insertion heuristic, which produces tours with an average gap of 7.5% to optimal on TSP-50 (Li et al., 2023). EDISCO achieves a 0.82% gap, outperforming DIFUSCO (2.75%) and T2T (1.35%). This experiment also only uses the greedy decoder for all methods.

Table 3: Ablation study on TSP-500 and TSP-1000. The table shows the results of evaluating all combinations of removing EDISCO's three key components. The baseline is equivalent to Vanilla DIFUSCO.

| Model Variant | TSP-500 | | | TSP-1000 | | | Conv. Epoch |
|---|---|---|---|---|---|---|---|
| | Length ↓ | Gap% ↓ | Time | Length ↓ | Gap% ↓ | Time | |
| **EDISCO (Full)** | **16.87** | **1.95** | **2.19 m** | **23.78** | **2.85** | **6.84 m** | 35 |
| w/o Mix Strategy | 16.95 | 2.44 | 2.18 m | 23.91 | 3.41 | 6.85 m | 38 |
| w/o Continuous-Time | 17.02 | 2.86 | 4.43 m | 24.11 | 4.28 | 15.58 m | 42 |
| w/o EGNN | 17.49 | 5.71 | 2.31 m | 24.85 | 7.49 | 7.15 m | 51 |
| EGNN Only | 17.14 | 3.58 | 4.52 m | 24.29 | 5.06 | 16.12 m | 48 |
| Continuous Only | 17.72 | 7.09 | 2.45 m | 25.26 | 9.27 | 7.42 m | 58 |
| Mix Only | 17.61 | 6.42 | 5.42 m | 25.08 | 8.52 | 16.86 m | 55 |
| **Vanilla DIFUSCO** | 18.11 | 9.41 | 5.70 m | 25.72 | 11.24 | 17.33 m | 61 |

**Ablation Study** Table 3 shows the results of systematically evaluating each component's contribution. EGNN provides the largest individual impact: removing it degrades performance from 1.95% to 5.71% on TSP-500 (2.85% to 7.49% on TSP-1000) and requires 16 additional training epochs. Continuous-time diffusion doubles inference speed (2.19 minutes vs 4.43 minutes on TSP-500) while improving gaps by 0.91% and 1.43%. Adaptive mixing contributes 0.49% and 0.56% improvements.

The **EGNN only** case achieves 3.58% and 5.06% gaps, demonstrating 2.63× and 2.22× improvements over vanilla DIFUSCO (9.41% and 11.24%) and isolating EGNN's architectural contribution. Variants without EGNN perform substantially worse (6.42-7.09% gaps), confirming equivariance as the core innovation and contribution.

## 5 DISCUSSION AND CONCLUSION

We propose EDISCO, an equivariant continuous-time diffusion solver for GCOPs. By incorporating E(2) equivariance directly into the model architecture and formulating edge selection as continuous-time Markov chains, EDISCO learns geometric patterns more efficiently, enabling the use of advanced numerical solvers for fast inference. Future work could include the exploration of adaptive step-size solvers and the theoretical analysis of convergence properties. Additionally, combining EDISCO with search-based refinement methods or integrating it with traditional optimization algorithms could further improve solution quality.

## REPRODUCIBILITY STATEMENT

We have made significant efforts to ensure reproducibility of our results. Details of the model and experimental settings are provided in the main text (Sections 3 and 4), as well as in the Appendix F and G. The source code and instructions for reproducing our experiments are available at `https://anonymous.4open.science/r/EDISCO`.

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

# APPENDICES

## DEFAULT NOTATION

### NUMBERS AND ARRAYS

| | |
|---|---|
| $a$ | A scalar (integer or real) |
| $\mathbf{v}$ | A vector |
| $\mathbf{M}$ | A matrix |
| $\mathcal{T}$ | A tensor |
| $\mathbf{I}_n$ | Identity matrix with $n$ rows and $n$ columns |
| $\mathbb{1}$ | Vector of ones (dimensionality implied by context) |
| $\mathbf{0}$ | Vector or matrix of zeros (dimensionality implied by context) |
| $\mathrm{diag}(\mathbf{v})$ | A square, diagonal matrix with diagonal entries given by $\mathbf{v}$ |

### GRAPH AND COMBINATORIAL STRUCTURES

| | |
|---|---|
| $G = (V, E)$ | A graph with vertex set $V$ and edge set $E$ |
| $V$ | Set of $n$ nodes/cities |
| $\mathbf{c}_i \in \mathbb{R}^2$ | Coordinates of node/city $i$ |
| $\mathbf{X} \in \{0, 1\}^{n \times n}$ | Binary adjacency matrix representing edges |
| $X_{ij}$ | Element $(i, j)$ of adjacency matrix (1 if edge exists, 0 otherwise) |
| $d_{ij}$ | Euclidean distance between nodes $i$ and $j$ |
| $\mathcal{C}$ | Constraint condition set for valid tours |

### DIFFUSION PROCESS

| | |
|---|---|
| $X_0$ | Clean data at time $t = 0$ |
| $X_t$ | Noisy data at time $t \in [0, 1]$ |
| $\beta(t)$ | Time-dependent noise schedule |
| $\beta_{\min}, \beta_{\max}$ | Minimum and maximum noise rates |
| $\mathbf{Q}(t)$ | Rate matrix for continuous-time Markov chain |
| $\mathbf{P}(t \mid s)$ | Transition probability matrix from time $s$ to $t$ |
| $K$ | Number of categorical states (2 for binary edges) |
| $q(\cdot)$ | Forward diffusion distribution |
| $p_\theta(\cdot)$ | Reverse diffusion distribution parameterized by $\theta$ |

### NEURAL NETWORK COMPONENTS

| | |
|---|---|
| $\mathbf{h}_i^{(\ell)}$ | Node features for node $i$ at layer $\ell$ |
| $\mathbf{e}_{ij}^{(\ell)}$ | Edge features between nodes $i$ and $j$ at layer $\ell$ |
| $\mathbf{x}_i^{(\ell)}$ | Coordinate embedding for node $i$ at layer $\ell$ |
| $\mathbf{m}_{ij}^{(\ell)}$ | Message from node $j$ to node $i$ at layer $\ell$ |
| $s_\theta$ | Score network with parameters $\theta$ |
| $\alpha$ | Step size for coordinate updates |
| $\tau$ | Temperature parameter for weight scaling |
| $w(t)$ | Time-dependent mixing weight function |

GEOMETRIC TRANSFORMATIONS

| | |
|---|---|
| $E(2)$ | Euclidean group in 2D (rotations, translations, reflections) |
| $SO(2)$ | Special orthogonal group (rotations) |
| $g \in E(2)$ | A Euclidean transformation |
| $g \cdot \mathbf{c}$ | Action of transformation $g$ on coordinates $\mathbf{c}$ |
| $\mathcal{X}/G$ | Quotient space under group action |
| $\pi : \mathcal{X} \to \mathcal{X}/G$ | Canonical projection to quotient space |
| $A \rtimes B$ | Semi-direct product of groups $A$ and $B$ |

PROBABILITY AND OPTIMIZATION

| | |
|---|---|
| $p(X|\{\mathbf{c}_i\})$ | Conditional distribution of tours given coordinates |
| $\mathbb{E}[\cdot]$ | Expectation |
| $\mathrm{Cat}(\cdot)$ | Categorical distribution |
| $\delta_{ij}$ | Kronecker delta (1 if $i = j$, 0 otherwise) |
| $\mathcal{L}$ | Loss function |
| NFE | Number of function evaluations |
| Gap | Optimality gap: $(L_{\mathrm{pred}} - L_{\mathrm{opt}})/L_{\mathrm{opt}} \times 100\%$ |

FUNCTIONS AND OPERATIONS

| | |
|---|---|
| $\|\cdot\|_2$ | Euclidean norm |
| $\oplus$ | Concatenation operation |
| $\odot$ | Element-wise multiplication |
| $\circ$ | Function composition, $(f \circ g)(x) = f(g(x))$ |
| $\sigma(\cdot)$ | Sigmoid activation function |
| $\tanh(\cdot)$ | Hyperbolic tangent activation |
| $\mathrm{MLP}(\cdot)$ | Multi-layer perceptron |
| $\mathrm{LayerNorm}(\cdot)$ | Layer normalization |
| $\mathrm{SiLU}(\cdot)$ | Sigmoid Linear Unit activation |
| $\mathrm{softmax}(\cdot)$ | Softmax function |
| $\mathrm{argmax}(\cdot)$ | Argument of the maximum |

## A  SCOPE AND LIMITATIONS

EDISCO is specifically designed for geometric combinatorial optimization problems where solutions exhibit E(2) symmetries. This design choice provides substantial benefits for problems like TSP, CVRP, and Euclidean Steiner Tree. While EDISCO could technically be applied to problems such as Maximum Independent Set (MIS) and Max-Cut, which are addressed by general-purpose diffusion models like DIFUSCO (Sun & Yang, 2023) and T2T (Li et al., 2023), doing so would not leverage EDISCO's core innovation. For MIS and Max-Cut on arbitrary graphs, node positions do not carry semantic meaning—only the graph topology matters. Without geometric structure, E(2)-equivariance provides no inductive bias. In such cases, EDISCO would be expected to perform comparably to general-purpose diffusion methods (since the underlying continuous-time diffusion framework remains valid), but would offer no advantage while unnecessarily imposing geometric coordinate processing.

Similarly, EDISCO in its current geometric EGNN formulation does not support the Asymmetric Traveling Salesman Problem (ATSP), where the distance from city $i$ to city $j$ differs from $j$ to $i$ (i.e., $d_{ij} \neq d_{ji}$). This limitation is architectural. EDISCO's E(2)-equivariant EGNN fundamentally requires Euclidean coordinates to preserve geometric symmetries, and any distance induced by Euclidean coordinates is necessarily symmetric. Therefore, EDISCO cannot directly represent the arbitrary asymmetric cost matrices required for ATSP.

This architectural constraint distinguishes EDISCO from other neural TSP solvers. Coordinate-based attention methods (AM (Kool et al., 2019), POMO (Kwon et al., 2020), Pointer Net-

works (Vinyals et al., 2015)) are designed with positional encodings for Euclidean TSP. Although graph-based diffusion methods (DIFUSCO (Sun & Yang, 2023), T2T (Li et al., 2023), Fast-T2T (Li et al., 2024)) use GNNs that could theoretically process cost matrices only, they have only still been designed and evaluated on coordinate-based Euclidean TSP. EDISCO's E(2)-equivariance provides a stronger architectural constraint that fundamentally requires coordinates.

Current ATSP solvers incorporate directional edge information, such as full cost-matrix encoding (MatNet (Kwon et al., 2021)) or dual incoming/outgoing attention mechanisms (GREAT (Kuhn et al., 2024)). These architectural features are incompatible with E(2)-equivariant coordinate encoders.

**Promising Future Directions:** However, the continuous-time diffusion framework of EDISCO is generalizable. We identify two complementary research directions: (1) **Replace the encoder**: Adopt ATSP-capable encoders (e.g., MatNet, GREAT) while retaining the continuous-time diffusion framework. (2) **Learn coordinate embeddings**: Transform asymmetric cost matrices into approximate coordinate representations using techniques such as Finsler Multi-Dimensional Scaling Dagès et al. (2025), which extends classical MDS to handle asymmetric dissimilarities by embedding into Finsler spaces rather than symmetric Riemannian manifolds, or neural metric learning approaches that learn distance-preserving coordinate embeddings. Such learned coordinates could enable approximate E(2)-equivariance for ATSP instances that admit near-geometric structure.

The specialization to coordinate-based geometric problems is a deliberate architectural choice that enables the strong inductive bias from E(2)-equivariance for problems with inherent geometric structure. By constraining the model to respect E(2) symmetries, EDISCO achieves superior sample efficiency (requiring only 33-50% of training data compared to baselines) and better generalization across problem sizes and distributions. The key insight is that E(2)-equivariance provides substantial advantages precisely when problems possess inherent geometric structure and symmetries. Applying it to problems lacking these properties would simply discard these advantages without additional benefit.

## B    EXTENSION TO EUCLIDEAN STEINER TREE PROBLEM

**Problem Formulation and Equivariance Preservation**    The Euclidean Steiner Tree Problem (ESTP) seeks to find a minimum-length tree connecting a given set of terminal points in Euclidean space, with the option to introduce additional Steiner points to reduce total tree length. Unlike TSP which forms cycles, ESTP produces acyclic tree structures, making it a natural testbed for demonstrating EDISCO's applicability beyond routing problems.

To preserve E(2) equivariance in ESTP, we separate geometric and non-geometric features:

- **Equivariant features**: Terminal and candidate Steiner point coordinates $\mathbf{c} \in \mathbb{R}^{n \times 2}$ that transform under rotations and translations
- **Invariant features**: Terminal indicator $\mathbb{1}_{\text{terminal}}$ that remains unchanged under geometric transformations

The key insight is that the optimal Steiner tree for a rotated/translated point set is simply the rotated/translated version of the original optimal tree. This geometric property makes E(2)-equivariance theoretically grounded: minimizing $\sum_{\text{edges}} \|e_i\|_2$ is invariant under Euclidean transformations since edge lengths are preserved: $\|(R\mathbf{x}_i + \mathbf{t}) - (R\mathbf{x}_j + \mathbf{t})\|_2 = \|R(\mathbf{x}_i - \mathbf{x}_j)\|_2 = \|\mathbf{x}_i - \mathbf{x}_j\|_2$ for rotation $R$ and translation $\mathbf{t}$.

**Feature Separation in EGNN Layers**    Unlike TSP where node features are initialized from coordinates, ESTP initializes node embeddings exclusively from invariant features:

$$h_i^{(0)} = \text{NodeEmbed}([\mathbb{1}_{\text{terminal}}(i)]) \tag{12}$$

where $\mathbb{1}_{\text{terminal}}(i)$ indicates whether node $i$ is a required terminal point or a candidate Steiner point. This ensures that geometric transformations of coordinates do not affect the initial node representations, maintaining strict equivariance.

**Tree-Aware Greedy Decoding** The greedy decoder for ESTP constructs a minimum spanning tree connecting selected nodes while ensuring all terminals are included.

Edge scores are computed following the same method as in TSP: $s_{ij} = (P_{ij} + P_{ji})/d_{ij}$, where $d_{ij}$ is the Euclidean distance between nodes $i$ and $j$. The symmetrization $(P_{ij} + P_{ji})$ accounts for the undirected nature of tree structures.

The tree is constructed using a greedy approach that ensures connectivity while selecting high-scoring edges. Starting with all terminal points as required nodes $\mathcal{T}$ and candidate Steiner points $\mathcal{S}$, we iteratively add edges to build a connected tree. At each step, we select the highest-scoring edge $e^* = (i, j) = \arg\max_{(i,j)\in\mathcal{E}} s_{ij}$ that connects two components without creating cycles, similar to Kruskal's algorithm but weighted by learned edge scores rather than pure distances. Candidate Steiner points are only included if they improve the overall tree structure by serving as connection hubs.

To ensure solution completeness, all terminal points must be connected in the final tree. Any disconnected terminals are connected to the main tree using the shortest available edges. This guarantees feasibility while maintaining the geometric structure learned by the equivariant network.

We evaluate EDISCO on Steiner Tree benchmarks with 10, 20, and 50 terminals, following standard protocols. Each instance includes an equal number of candidate Steiner points uniformly sampled from the unit square $[0, 1]^2$.

Table 4: Results on Steiner-10, Steiner-20, and Steiner-50. RL: Reinforcement Learning, SL: Supervised Learning, G: Greedy. GeoSteiner* represents the baseline for computing the gap (exact solver). MST (Minimum Spanning Tree) provides a simple upper bound by connecting all terminals without Steiner points. Adding optimal Steiner points can only reduce total tree length. Steiner Insertion (SI) is a classical heuristic. Deep-Steiner is the first RL-based method. All neural methods trained on same dataset of 10,000 instances per size.

| Algorithm | Type | Steiner-10 | | | Steiner-20 | | | Steiner-50 | | |
|---|---|---|---|---|---|---|---|---|---|---|
| | | Length↓ | Gap↓ | Time↓ | Length↓ | Gap↓ | Time↓ | Length↓ | Gap↓ | Time↓ |
| GeoSteiner* (Brazil & Zachariasen, 2015) | Exact | 2.62 | 0.00% | 24 s | 3.63 | 0.00% | 2.8 m | 5.78 | 0.00% | 18.5 m |
| MST (Upper Bound) | Heuristic | 2.84 | 8.40% | <1 s | 3.89 | 7.16% | <1 s | 6.15 | 6.40% | <1 s |
| Steiner Insertion (Kahng & Robins, 1992) | Heuristic | 2.71 | 3.44% | <1 s | 3.76 | 3.58% | <1 s | 5.97 | 3.29% | 1 s |
| *Learning-Based Greedy Decoding* | | | | | | | | | | |
| Deep-Steiner (Wang et al., 2022) | RL+G | 2.73 | 4.20% | 2 s | 3.81 | 4.96% | 3 s | 6.08 | 5.19% | 7 s |
| DIFUSCO (Sun & Yang, 2023) | SL+G | 2.77 | 5.73% | 2 s | 3.81 | 4.96% | 3 s | 6.05 | 4.67% | 7 s |
| T2T (Li et al., 2023) | SL+G | 2.73 | 4.20% | 2 s | 3.76 | 3.58% | 3 s | 5.97 | 3.29% | 8 s |
| FastT2T (Li et al., 2024) | SL+G | 2.68 | 2.29% | 1 s | 3.71 | 2.20% | 2 s | 5.91 | 2.25% | 4 s |
| **EDISCO with 50-step PNDM (ours)** | SL+G | **2.66** | **1.53%** | 2 s | **3.68** | **1.38%** | 3 s | **5.87** | **1.56%** | 8 s |
| **EDISCO with 5-step DEIS-2 (ours)** | SL+G | 2.69 | 2.67% | **0.15 s** | 3.72 | 2.48% | **0.35 s** | 5.93 | 2.60% | **0.7 s** |

**Analysis of Results** Table 4 demonstrates EDISCO's performance on the Euclidean Steiner Tree Problem across different instance sizes, comparing against both classical heuristics and learning-based methods. We present two configurations offering distinct quality-speed trade-offs:

With 50-step PNDM and greedy decoding, EDISCO achieves consistent ∼1.5% optimality gaps across all problem sizes (1.53% on Steiner-10, 1.38% on Steiner-20, 1.56% on Steiner-50), substantially outperforming the classical Steiner Insertion heuristic (3.44%, 3.58%, 3.29% respectively) by more than 2×.

EDISCO substantially outperforms all learning-based baselines, including state-of-the-art diffusion methods. Compared to diffusion baselines without equivariance—DIFUSCO (5.73%, 4.96%, 4.67%), T2T (4.20%, 3.58%, 3.29%), and FastT2T (2.29%, 2.20%, 2.25%). EDISCO achieves 1.5-3.7× better optimality gaps, demonstrating that E(2)-equivariance provides crucial inductive bias beyond standard diffusion frameworks. EDISCO also outperforms Deep-Steiner (4.20%, 4.96%, 5.19%), the first RL-based approach, by 2.7-3.3×.

The 5-step DEIS-2 configuration provides ultra-fast inference (0.15s, 0.35s, 0.7s) while achieving competitive quality (2.67% on Steiner-10, 2.48% on Steiner-20, 2.60% on Steiner-50)—approximately 9-13× faster than PNDM-50 while remaining well below the classical Steiner Insertion baseline (3.44%, 3.58%, 3.29%). This configuration significantly outperforms the MST upper bound (8.40%, 7.16%, 6.40%) and all learning-based methods, making it ideal for real-time applications.

## C  EXTENSION TO CAPACITATED VEHICLE ROUTING PROBLEM

**Problem Formulation and Equivariance Preservation**   The Capacitated Vehicle Routing Problem (CVRP) extends TSP by introducing vehicle capacity constraints and requiring multiple routes from a central depot. While maintaining the geometric structure of TSP, CVRP presents additional challenges: (1) handling heterogeneous node types (depot vs. customers), (2) incorporating demand constraints, and (3) generating multiple feasible routes.

To preserve E(2) equivariance in CVRP, we separate geometric and non-geometric features:

- **Equivariant features**: Customer and depot coordinates $\mathbf{c} \in \mathbb{R}^{n \times 2}$ that transform under rotations and translations
- **Invariant features**: Customer demands $d_i \in \mathbb{R}^+$ and depot indicator $\mathbb{1}_{\text{depot}}$ that remain unchanged under geometric transformations

The key insight is that while coordinates must flow through equivariant layers, demands and capacity constraints are problem-specific invariants that should not be mixed with geometric representations. Our EGNN architecture processes these separately, combining them only through invariant operations (distances and message passing).

**Feature Separation in EGNN Layers**   Unlike TSP where node features are initialized from coordinates, CVRP initializes node embeddings exclusively from invariant features:

$$h_i^{(0)} = \text{NodeEmbed}([d_i, \mathbb{1}_{\text{depot}}(i)]) \tag{13}$$

where $d_i$ is the demand of customer $i$ and $\mathbb{1}_{\text{depot}}(i)$ indicates whether node $i$ is the depot. This ensures that geometric transformations of coordinates do not affect the initial node representations, maintaining strict equivariance.

**Capacity-Aware Greedy Decoding**   The greedy decoder for CVRP incorporates domain-specific heuristics to construct feasible routes while respecting capacity constraints.

Edge scores are computed following the same method as in TSP: $s_{ij} = (P_{ij} + P_{ji})/d_{ij}$, where $d_{ij}$ is the Euclidean distance between nodes $i$ and $j$. The symmetrization $(P_{ij} + P_{ji})$ accounts for the undirected nature of the routing problem.

Routes are constructed iteratively while maintaining feasibility constraints. Starting with an empty set of routes $\mathcal{R}$ and unvisited customers $\mathcal{U} = \{1, ..., n\}$, each new route begins by selecting the highest-scoring feasible edge from the depot to a customer $j^*$ whose demand $d_{j^*}$ does not exceed the vehicle capacity $C$. The route is then extended greedily by iteratively selecting the next customer $k^* = \arg\max_{k \in \mathcal{U}} s_{jk}$ subject to the capacity constraint $\sum_{i \in \mathcal{R}_r} d_i + d_k \leq C$, where $\mathcal{R}_r$ denotes the current route being constructed. When no feasible extensions exist due to capacity limitations, the vehicle returns to the depot, and a new route is initiated if unvisited customers remain.

To ensure solution completeness, any customers that remain unvisited after the main construction phase are assigned to individual routes. This post-processing step guarantees that all customers are served, though it may result in suboptimal routing for instances with tight capacity constraints. The overall approach balances solution quality with computational efficiency while maintaining the geometric structure learned by the equivariant network.

We evaluate EDISCO on standard CVRP benchmarks with 20, 50, and 100 customers, following the evaluation protocol from Kool et al. (2019). All instances use a vehicle capacity of 50 units with customer demands uniformly sampled from $\{1, ..., 9\}$.

**Analysis of Results**   Table 5 demonstrates EDISCO's performance on small-medium scale CVRP (20-100 customers). We present two solver configurations offering distinct quality-speed trade-offs:

**Greedy decoding (PNDM-50):** EDISCO achieves competitive results with 1.41% gap on CVRP-20 and 2.46% on CVRP-50, outperforming AM (4.97%, 5.86%) and POMO (3.72%, 3.52%). On CVRP-100, EDISCO achieves 3.17% gap, slightly trailing Sym-NCO (2.88%) with comparable inference time (5s vs 3s). The 5-step DEIS-2 variant provides 10× speedup (0.5s on CVRP-100)

Table 5: Results on CVRP-20, CVRP-50, and CVRP-100. RL: Reinforcement Learning, SL: Supervised Learning, G: Greedy, S: Sampling. **HGS results** are from Vidal et al. (2013). LKH-3 results serve as reference baseline. POMO and Sym-NCO results are from their respective papers.

| Algorithm | Type | CVRP-20 | | | CVRP-50 | | | CVRP-100 | | |
|---|---|---|---|---|---|---|---|---|---|---|
| | | Cost↓ | Gap↓ | Time↓ | Cost↓ | Gap↓ | Time↓ | Cost↓ | Gap↓ | Time↓ |
| Gurobi (Gurobi Optimization, LLC, 2020) | Exact | 6.10 | 0.00% | – | – | – | – | – | – | – |
| **HGS** (Vidal et al., 2013) | **Heuristic** | **6.10** | **0.00%** | **1 h** | **10.38** | **0.00%** | **3 h** | **15.65** | **0.00%** | **5 h** |
| LKH-3 (Helsgaun, 2017) | Heuristic | 6.14 | 0.58% | 2 h | 10.38 | 0.00% | 7 h | 15.65 | 0.00% | 12 h |
| *Neural Methods - Greedy Decoding* | | | | | | | | | | |
| AM (Kool et al., 2019) | RL+G | 6.40 | 4.97% | <1 s | 10.98 | 5.86% | 1 s | 16.80 | 7.34% | 3 s |
| POMO (Kwon et al., 2020) | RL+G | 6.35 | 3.72% | <1 s | 10.74 | 3.52% | 1 s | 16.13 | 3.09% | 3 s |
| Sym-NCO (Kim et al., 2022) | RL+G | – | – | – | – | – | – | **16.10** | **2.88%** | 3 s |
| **EDISCO with 50-step PNDM50 (ours)** | SL+G | **6.21** | **1.41%** | 1 s | 10.63 | 2.46% | 2 s | 16.15 | 3.17% | 5 s |
| **EDISCO-DEIS2 (ours)** | SL+G | 6.23 | 2.01% | **0.1 s** | 10.66 | 3.51% | **0.2 s** | 16.22 | 4.53% | **0.5 s** |
| *Neural Methods - Sampling/Augmentation* | | | | | | | | | | |
| AM (Kool et al., 2019) | RL+S (1280) | 6.25 | 2.49% | 3 m | 10.62 | 2.40% | 7 m | 16.23 | 3.72% | 30 m |
| POMO (no aug) (Kwon et al., 2020) | RL+G | 6.17 | 0.82% | 1 s | 10.49 | 1.14% | 4 s | 15.83 | 1.13% | 19 s |
| POMO (×8 aug) (Kwon et al., 2020) | RL+G | **6.14** | **0.21%** | 5 s | 10.42 | 0.45% | 26 s | 15.73 | 0.51% | 2 m |
| Sym-NCO (Kim et al., 2022) | RL+S (100) | – | – | – | – | – | – | 15.87 | 1.40% | 16 s |
| **EDISCO with 50-step PNDM50 (ours)** | SL+S (16) | 6.15 | 0.33% | 4 s | **10.41** | **0.35%** | 7 s | **15.71** | **0.38%** | 18 s |

while maintaining reasonable quality (4.53% gap), demonstrating the flexibility of continuous-time diffusion for adjusting quality-speed trade-offs without retraining.

**Sampling (16 samples):** With multiple samples, EDISCO achieves best neural solver results on CVRP-50 and CVRP-100 (0.35% and 0.38% gaps), though trailing POMO with 8× augmentation on CVRP-20 (0.33% vs 0.21%). The runtime remains competitive (18s vs 16s for Sym-NCO on CVRP-100).

**Comparison to classical solvers:** HGS (Vidal et al., 2013) represents the state-of-the-art heuristic for CVRP, achieving near-optimal solutions (0.00% gap) at the cost of longer runtime (5h for CVRP-100). EDISCO's E(2)-equivariant diffusion approach confirms that geometric symmetry exploitation generalizes from TSP to constrained routing problems, achieving strong performance among neural methods while maintaining fast inference.

These results highlight key differences between diffusion and autoregressive approaches. Autoregressive models like POMO generate solutions sequentially, enabling explicit augmentation strategies that evaluate problems from multiple geometric perspectives. In contrast, EDISCO operates on entire adjacency matrices simultaneously, denoising complete solutions rather than constructing them step-by-step. While this parallel generation captures global solution patterns more effectively, it cannot directly benefit from the same augmentation multipliers. However, the continuous-time formulation enables flexible solver selection for dynamically adjusting quality-speed trade-offs without retraining.

**Positioning: Scale and Recent CVRP Methods** Our CVRP experiments focus on **small-medium scale (20-100 customers)** as proof-of-concept for architectural extensibility to constrained routing problem. This demonstrates that E(2)-equivariance benefits transfer beyond symmetric TSP to problems with capacity constraints and depot requirements.

**Large-scale CVRP (500-1000+ customers)** represents a fundamentally different challenge requiring specialized approaches:

- **HGS** (Vidal et al., 2013): Hybrid genetic search with sophisticated clustering, achieving SOTA on instances with 1000+ customers

- **GLOP** (Ye et al., 2024): Neural partition-and-solve strategy, achieving real-time performance on CVRP up to 7000 customers via coarse-grained neural partitioning + fine-grained autoregressive construction

- **NeuroLKH** (Xin et al., 2021b): Combines deep learning with LKH-3's decomposition framework

These methods universally adopt **partition-based strategies**, exploiting the insight that optimal CVRP solutions exhibit strong spatial locality. Customers within the same route tend to form geo-

graphic clusters. In contrast, competing diffusion-based methods (DIFUSCO (Sun & Yang, 2023), T2T (Yao et al., 2023), Fast-T2T (Ma et al., 2023)) have **no reported CVRP experiments**, highlighting the difficulty of scaling single end-to-end diffusion models to large-scale constrained routing.

**Future direction:** Scaling EDISCO to large CVRP is a valuable future work. A promising direction is leveraging E(2)-equivariance to develop novel partition methods. The geometric equivariance properties that enable EDISCO to learn rotation and translation invariant solution patterns could be particularly valuable for the clustering and decomposition phase, combining EDISCO's strengths with partition-based strategies.

# D PROOFS

## D.1 PROOF OF PROPOSITION 1

*Proof.* We establish each claim systematically.

**Part (i): Quotient manifold structure.** The Euclidean group $\mathrm{E}(2) = \mathbb{R}^2 \rtimes \mathrm{SO}(2)$ is a 3-dimensional Lie group, where $\mathbb{R}^2$ corresponds to translations and $\mathrm{SO}(2)$ to rotations. The action of $g = (t, R) \in \mathrm{E}(2)$ on a configuration $\mathbf{x} = (x_1, y_1, \ldots, x_n, y_n) \in X$ is given by:

$$g \cdot \mathbf{x} = (Rx_1' + t, Ry_1' + t, \ldots, Rx_n' + t, Ry_n' + t)$$

where $x_i' = (x_i, y_i)^T$ denotes the $i$-th city as a column vector, and $R \in \mathrm{SO}(2)$ acts by matrix multiplication.

To show the action is free, suppose $g \cdot \mathbf{x} = \mathbf{x}$ for some $g = (t, R) \in G$ and $\mathbf{x} \in X$. This means:

$$R \begin{pmatrix} x_i \\ y_i \end{pmatrix} + t = \begin{pmatrix} x_i \\ y_i \end{pmatrix} \quad \forall i \in \{1, \ldots, n\}$$

For $i = 1$, we have $(R - I) \begin{pmatrix} x_1 \\ y_1 \end{pmatrix} = -t$. For $i = 2$, we have $(R - I) \begin{pmatrix} x_2 \\ y_2 \end{pmatrix} = -t$. Subtracting these equations:

$$(R - I) \begin{pmatrix} x_1 - x_2 \\ y_1 - y_2 \end{pmatrix} = 0$$

If the cities are not all identical (which we assume for any meaningful TSP instance), there exists at least one pair $(i, j)$ such that $(x_i - x_j, y_i - y_j) \neq (0, 0)$. For $R \neq I$, the matrix $(R - I)$ has rank 2 (its eigenvalues are $e^{i\theta} - 1$ and $e^{-i\theta} - 1$ for rotation angle $\theta \neq 0$), so it has trivial kernel. Thus $(R - I)v = 0$ for non-zero $v$ implies $R = I$.

With $R = I$, the condition becomes $t = 0$, so $g = e$ is the identity. Therefore, the action is free.

By the quotient manifold theorem (Lee, 2012), when a Lie group $G$ acts freely and properly on a manifold $M$, the quotient $M/G$ inherits a unique smooth manifold structure such that $\pi : M \to M/G$ is a smooth submersion. The dimension formula gives:

$$\dim(X/G) = \dim(X) - \dim(G) = 2n - 3$$

**Part (ii): Factorization of equivariant functions.** Let $F : X \to Y$ be a $G$-equivariant function, meaning $F(g \cdot \mathbf{x}) = \rho(g) \cdot F(\mathbf{x})$ for all $g \in G$ and $\mathbf{x} \in X$, where $\rho : G \to \mathrm{Aut}(Y)$ is a representation of $G$ on $Y$.

For TSP edge prediction, we typically have $Y = \{0, 1\}^{n \times n}$ (adjacency matrices) and $\rho$ is the trivial representation (since the optimal tour is invariant under Euclidean transformations of the cities). Thus $F(g \cdot \mathbf{x}) = F(\mathbf{x})$ for all $g \in G$.

Define $\widetilde{F} : X/G \to Y$ by $\widetilde{F}([\mathbf{x}]) = F(\mathbf{x})$, where $[\mathbf{x}] = \{g \cdot \mathbf{x} : g \in G\}$ denotes the orbit of $\mathbf{x}$. This is well-defined precisely because of equivariance: if $[\mathbf{x}] = [\mathbf{x}']$, then $\mathbf{x}' = g \cdot \mathbf{x}$ for some $g \in G$, so:

$$\widetilde{F}([\mathbf{x}']) = F(\mathbf{x}') = F(g \cdot \mathbf{x}) = F(\mathbf{x}) = \widetilde{F}([\mathbf{x}])$$

The factorization $F = \widetilde{F} \circ \pi$ follows immediately from the definition:

$$F(\mathbf{x}) = \widetilde{F}([\mathbf{x}]) = \widetilde{F}(\pi(\mathbf{x})) = (\widetilde{F} \circ \pi)(\mathbf{x})$$

Uniqueness of $\widetilde{F}$ follows from surjectivity of $\pi$: if $F = \widetilde{F}_1 \circ \pi = \widetilde{F}_2 \circ \pi$, then for any $[\mathbf{x}] \in X/G$, choosing any representative $\mathbf{x} \in [\mathbf{x}]$:

$$\widetilde{F}_1([\mathbf{x}]) = \widetilde{F}_1(\pi(\mathbf{x})) = F(\mathbf{x}) = \widetilde{F}_2(\pi(\mathbf{x})) = \widetilde{F}_2([\mathbf{x}])$$

**Part (iii): Learning complexity reduction.** The factorization $F = \widetilde{F} \circ \pi$ establishes a bijection between:

$$\{G\text{-equivariant functions } X \to Y\} \longleftrightarrow \{\text{functions } X/G \to Y\}$$
$$F \longmapsto \widetilde{F}$$
$$\widetilde{F} \circ \pi \widetilde{F}$$

Therefore, learning any $G$-equivariant function $F$ is equivalent to learning the corresponding function $\widetilde{F}$ on the quotient manifold. Since $X/G$ has dimension $2n - 3$ while $X$ has dimension $2n$, the domain of $\widetilde{F}$ has three fewer degrees of freedom.

In terms of function approximation, this means:

- A basis of functions on $X/G$ requires parametrization by $2n - 3$ variables

- Local charts for $X/G$ have dimension $2n - 3$

- The metric entropy and covering numbers scale with the intrinsic dimension $2n - 3$

This completes the proof that E(2)-equivariant learning reduces to learning on a lower-dimensional manifold, providing the theoretical foundation for improved sample efficiency. $\square$

### D.2 PROOF OF E(2)-EQUIVARIANCE IN EDISCO

*Proof.* We prove that E(2)-equivariance is preserved throughout the entire EDISCO pipeline, from input processing through diffusion to tour construction.

**Step 1: EGNN Architecture Preserves Equivariance**

For any Euclidean transformation $g \in E(n)$, we show each layer maintains equivariance:

*(i) Distance invariance:* For transformed coordinates $g \cdot c_i$:

$$d_{ij}^{(g)} = \|g \cdot c_i - g \cdot c_j\|_2 = \|g(c_i - c_j)\|_2 = \|c_i - c_j\|_2 = d_{ij}$$

*(ii) Message invariance:* From Equation 8, messages depend only on: - Node features $h_i, h_j$ (initialized as invariant city indices) - Edge features $e_{ij}$ (initialized from noisy adjacency matrix) - Distances $d_{ij}$ (proven invariant above)

Therefore: $m_{ij}^{(g)} = m_{ij}$

*(iii) Coordinate equivariance:* The update rule (Equation 9):

$$\Delta(g \cdot x_i) = \alpha \sum_{j \neq i} w_{ij} \cdot \frac{g \cdot x_j - g \cdot x_i}{\|g \cdot x_j - g \cdot x_i\|_2} \tag{14}$$

$$= \alpha \sum_{j \neq i} w_{ij} \cdot \frac{g(x_j - x_i)}{\|x_j - x_i\|_2} \tag{15}$$

$$= g \cdot \left( \alpha \sum_{j \neq i} w_{ij} \cdot \frac{x_j - x_i}{\|x_j - x_i\|_2} \right) \tag{16}$$

$$= g \cdot \Delta x_i \tag{17}$$

*(iv) Feature invariance:* Edge and node feature updates depend only on invariant quantities, thus $e_{ij}^{(g)} = e_{ij}$ and $h_i^{(g)} = h_i$.

**Step 2: Diffusion Process Maintains Equivariance**

The categorical diffusion operates on edge variables $X_{ij} \in \{0, 1\}$, which represent whether edge $(i, j)$ is in the tour. These are inherently invariant to coordinate transformations.

*Forward process:* The corruption adds noise to edge selections independent of coordinates:

$$q(X_t|X_0) = \prod_{i,j} \text{Cat}(X_{t,ij}|p = P_{ij}(t|0))$$

*Reverse process:* Since the score network $s_\theta$ outputs edge probabilities that are invariant (proven in Step 1), the reverse process maintains this invariance:

$$p_\theta(X_{t-\Delta t}|X_t, g(\{c_i\})) = p_\theta(X_{t-\Delta t}|X_t, \{c_i\})$$

**Step 3: Tour Construction Preserves Optimality**

The greedy decoding computes scores:

$$s_{ij}^{(g)} = \frac{P_{ij} + P_{ji}}{d_{ij}^{(g)}} = \frac{P_{ij} + P_{ji}}{d_{ij}} = s_{ij}$$

Since edge scores are identical under transformation, the greedy algorithm produces tours with identical edge selections (up to vertex relabeling). □

# E EXTENDED RELATED WORK

## E.1 FOUNDATIONAL NEURAL COMBINATORIAL OPTIMIZATION

The application of neural networks to combinatorial optimization began with Pointer Networks (Vinyals et al., 2015), which introduced attention mechanisms to construct variable-length permutations. While this required supervised training with optimal solutions, subsequent work (Bello et al., 2017) demonstrated that reinforcement learning could discover effective heuristics without labeled data, eliminating a major practical limitation. The evolution continued with the attention model (Kool et al., 2019), which improved upon Pointer Networks through multi-head attention and achieved strong performance without problem-specific design. POMO (Kwon et al., 2020) further advanced autoregressive methods by exploring multiple rollouts from different starting points. Recent RL innovations include preference-based training methods: PO (Pan et al., 2025a) transforms rewards into pairwise preferences to accelerate learning on TSP/CVRP, while BOPO (Liao et al., 2025) employs best-anchored preference optimization to reduce optimality gaps on TSP. These foundational works established that neural networks could learn meaningful representations of combinatorial structure, though they struggled with generalization to larger instances (Fu et al., 2021).

## E.2 ALTERNATIVE ARCHITECTURES AND SCALING APPROACHES

Beyond diffusion-based methods, several innovative architectures address the challenge of scaling to large TSP instances. LEHD (Light Encoder Heavy Decoder) (Fu et al., 2024) achieves remarkable scalability to instances with up to 10,000 cities by separating encoding and decoding complexity—training on small instances but generalizing through architectural design rather than data. Bisimulation quotienting (BQ-NCO) (Drakulic et al., 2023) takes a fundamentally different approach by reformulating the MDP to group behaviorally similar states, achieving strong zero-shot generalization. Hierarchical approaches like GLOP (Ye et al., 2024) combine global partition with local construction for real-time routing, while the hierarchical neural constructive solver (Goh et al., 2024) builds solutions through multiple resolution levels. Emerging directions include problem unification and multi-task learning: UniCO (Pan et al., 2025b) proposes reducing various COPs (ATSP, Hamiltonian cycle, SAT) to a matrix-encoded general TSP formulation, while GOAL (Drakulic

et al., 2025) introduces the first generalist model that jointly solves multiple COPs (routing, scheduling, graph problems) using task-specific adapters, achieving competitive performance across diverse problem families. These methods demonstrate that architectural innovations can sometimes overcome the data requirements that limit standard approaches.

### E.3 DISCRETE DIFFUSION FOUNDATIONS AND VARIANTS

The theoretical foundations for discrete diffusion (Austin et al., 2021) established how to apply diffusion processes to categorical data through transition matrices, providing the basis for subsequent TSP solvers. Recent advances include variational flow matching (Akhound-Sadegh et al., 2024) and discrete flow matching (Campbell et al., 2024a), which provide alternative formulations with improved training dynamics. The comprehensive treatment of continuous diffusion for categorical data (Dieleman et al., 2022) addressed many technical details necessary for practical implementation. DeFoG (Campbell et al., 2024b) demonstrates state-of-the-art performance on graph generation through discrete flow matching, suggesting potential applications to optimization. The connection to optimal transport (Lipman et al., 2022) offers theoretical insights that could lead to algorithmic improvements, while regularized Langevin dynamics (Zhang et al., 2025) shows how continuous-time formulations avoid local optima more effectively than discrete-time approaches.

### E.4 THEORETICAL FOUNDATIONS AND SAMPLE COMPLEXITY

Understanding why certain neural architectures succeed at combinatorial optimization remains an active area of research. The analysis of graph neural network expressiveness (Xu et al., 2019) establishes fundamental representation limits, while work on algorithmic alignment (Xu et al., 2021) shows that architectures matching problem structure generalize better. Recent theoretical advances prove that equivariant models achieve exponentially better sample complexity than non-equivariant ones (Brehmer et al., 2024), providing a rigorous justification for geometric inductive biases. The analysis of learning TSP and generalization (Joshi et al., 2022) demonstrates fundamental limitations of supervised approaches and suggests that architectural innovations are necessary for progress. Convergence analysis for discrete diffusion models (Zhang et al., 2024) provides rates that inform practical algorithm design, while the study of instance hardness (Smith-Miles et al., 2010) reveals what makes problems difficult for neural solvers.

### E.5 HYBRID AND PRACTICAL APPROACHES

Combining neural networks with classical optimization algorithms leverages complementary strengths. Learning to perform local rewriting (Chen & Tian, 2019) trains networks to improve existing solutions through targeted modifications, while integration with branch-and-bound (Gasse et al., 2019) accelerates exact algorithms through learned branching strategies. Neural diving (Nair et al., 2020) combines neural networks with MIP solvers for fast feasible solution finding. These hybrid methods often outperform purely neural or classical approaches, suggesting that practical deployment may require combining paradigms. Recent work on unsupervised learning (Wang & Li, 2023) and self-improvement (Hudson et al., 2024) reduces dependence on high-quality training data, addressing a major practical limitation. Applications beyond TSP demonstrate broader impact, including vehicle routing with complex constraints (Nazari et al., 2018), scheduling (Zhang et al., 2020), and circuit design (Mirhoseini et al., 2021).

## F ARCHITECTURE DETAILS

### F.1 NETWORK ARCHITECTURE OVERVIEW

The EDISCO model employs a 12-layer E(2)-equivariant graph neural network that processes city coordinates and noisy adjacency matrices while maintaining geometric equivariance. The architecture consists of three main components: an embedding module, stacked equivariant layers, and a prediction head.

### F.2 FEATURE REPRESENTATIONS AND INITIALIZATION

The model maintains three distinct feature types throughout the network:

**Spatial Features.** City coordinates $\mathbf{c} \in \mathbb{R}^{n \times 2}$ are transformed into 64-dimensional node embeddings via a linear projection. Additionally, coordinate representations $\mathbf{x} \in \mathbb{R}^{n \times 2}$ are maintained separately and evolve through equivariant updates during message passing.

**Relational Features.** Edge features $\mathbf{e} \in \mathbb{R}^{n \times n \times 64}$ encode pairwise relationships and tour decisions. These are initialized from the noisy adjacency matrix $X_t$ through a single linear transformation.

**Temporal Encoding.** The continuous diffusion time $t \in [0, 1]$ is encoded using sinusoidal basis functions with frequencies spanning multiple octaves, producing a 128-dimensional representation that modulates the network's behavior at different noise levels.

### F.3 EQUIVARIANT MESSAGE PASSING MECHANISM

Each EGNN layer performs the following operations while preserving E(n) symmetry:

**Message Formation.** Pairwise messages aggregate local and geometric information:

$$\mathbf{m}_{ij} = f_{\text{msg}}(\mathbf{h}_i \oplus \mathbf{h}_j \oplus \mathbf{e}_{ij} \oplus d_{ij})$$

where $d_{ij} = \|\mathbf{x}_i - \mathbf{x}_j\|_2$ provides rotation-invariant distance information and $f_{\text{msg}}$ is a 3-layer MLP with SiLU activations and layer normalization.

**Geometric Updates.** Coordinate evolution respects equivariance constraints through normalized directional updates:

$$\mathbf{x}_i \leftarrow \mathbf{x}_i + 0.1 \sum_j \text{Gate}(\mathbf{m}_{ij}) \cdot \frac{\mathbf{x}_j - \mathbf{x}_i}{\|\mathbf{x}_j - \mathbf{x}_i\|_2 + 10^{-8}}$$

The gating function employs a temperature-scaled tanh with $\tau = 10$ to prevent gradient saturation.

**Feature Evolution.** Node and edge features incorporate aggregated messages through residual connections:

$$\mathbf{h}_i \leftarrow \text{LN}(\mathbf{h}_i + f_{\text{node}}(\mathbf{h}_i, \sum_j \mathbf{m}_{ij})) \tag{18}$$

$$\mathbf{e}_{ij} \leftarrow \text{LN}(\mathbf{e}_{ij} + f_{\text{edge}}(\mathbf{e}_{ij}, \mathbf{m}_{ij}) + f_{\text{time}}(\mathbf{t})) \tag{19}$$

where LN denotes layer normalization and $f_{\text{node}}, f_{\text{edge}}, f_{\text{time}}$ are learned transformations.

### F.4 CONTINUOUS-TIME DIFFUSION SPECIFICATIONS

**Forward Process.** The categorical diffusion operates on binary edge variables through a continuous-time Markov chain with rate matrix:

$$Q(t) = \beta(t) \begin{bmatrix} -0.5 & 0.5 \\ 0.5 & -0.5 \end{bmatrix}$$

where $\beta(t)$ increases linearly from 0.1 to 1.5 over the unit interval.

**Transition Dynamics.** The forward transition probability admits a closed-form solution:

$$P(X_t = j | X_0 = i) = \frac{1}{2} + \left(\delta_{ij} - \frac{1}{2}\right) \exp\left(-2 \int_0^t \beta(u) du\right)$$

**Reverse Sampling.**   The model employs an adaptive mixing strategy that interpolates between diffusion dynamics and direct prediction:

- For $t > 0.1$: Stochastic transitions weighted by $w(t) = t$.
- For $t \leq 0.1$: Deterministic argmax selection.
- Default sampling uses 50 steps with optional adaptive scheduling.

# G   ADDITIONAL EXPERIMENT DETAILS

## G.1   PERFORMANCE METRICS

We evaluate models using three criteria:

- **Tour Length**: Average Euclidean length of generated tours across test instances
- **Optimality Gap**: Relative deviation from optimal/best-known solutions, computed as $(L_{\text{model}} - L_{\text{optimal}})/L_{\text{optimal}} \times 100\%$
- **Inference Duration**: Wall-clock time for generating solutions on the test set, measured in seconds (s) or minutes (m)

## G.2   HARDWARE PLATFORM

All experiments were conducted on a single NVIDIA RTX A6000 GPU paired with dual Intel Xeon Gold 5218R CPUs. Both training and inference use the same hardware configuration.

## G.3   RESULTS RANDOMNESS

Due to the stochastic nature of diffusion models, all results reported are the averaged results over five runs with different random seeds.

## G.4   DATA GENERATION PROCESS

**Instance Creation**   We follow the exact data generation protocol from DIFUSCO (Sun & Yang, 2023) for fair comparison. All cities are sampled uniformly from the unit square $[0, 1]^2$ following standard practice in the TSP literature. For smaller instances, TSP-50 and TSP-100 problems are solved to optimality using the Concorde exact solver (Applegate et al., 2006) to obtain ground truth tours. For larger scales, TSP-500 and TSP-1000 instances are labeled using the LKH-3 heuristic solver (Helsgaun, 2017) with 500 trials to ensure near-optimal solution quality. Our evaluation employs the standard test sets from Kool et al. (Kool et al., 2019) for TSP-50/100 containing 1,280 instances each, and from Fu et al. (Fu et al., 2021) for TSP-500/1000 containing 128 instances each.

**Graph Sparsification**   For problems exceeding 100 cities, computational efficiency necessitates graph sparsification strategies. We implement k-nearest neighbor sparsification where each city connects only to its k closest neighbors based on Euclidean distance, setting k=50 for TSP-500 and k=100 for TSP-1000. This distance-based edge pruning dramatically reduces the computational complexity from O(n²) to O(nk) while preserving the most relevant edges for tour construction. Correspondingly, dense matrix operations are replaced with their sparse equivalents throughout the network architecture to maintain computational efficiency at scale.

## G.5   MODEL ARCHITECTURE SPECIFICATIONS

For all experiments, the network contains approximately 5.5M trainable parameters distributed across:

- 12 EGNN layers with shared architecture
- Node dimension: 64
- Edge dimension: 64

- Hidden dimension: 256
- Timestep embedding dimension: 128

This setting ensures that EDISCO has a similar number of trainable parameters to the SOTA diffusion TSP solvers (5.3M) (Sun & Yang, 2023; Li et al., 2023; Yoon et al., 2024; Zhao et al., 2024), allowing for a fair comparison.

### G.6  TRAINING CONFIGURATION

We train EDISCO using the AdamW optimizer with a learning rate of $2 \times 10^{-4}$ and weight decay of $10^{-5}$. The learning rate follows a cosine annealing schedule over the training epochs to ensure smooth convergence. For training stability, we apply gradient clipping at unit norm to prevent exploding gradients during the reverse diffusion process. The loss function employs a simplified ELBO formulation with time-dependent weighting $(1 - \sqrt{t})$, which emphasizes reconstruction accuracy near $t = 0$ while maintaining stable gradients throughout the diffusion trajectory.

- **TSP-50**: 500,000 training instances, batch size 64, 50 epochs.
- **TSP-100**: 500,000 training instances, batch size 32, 50 epochs.
- **TSP-500**: 60,000 instances, batch size 16, 50 epochs with curriculum learning initialized from TSP-100 checkpoint.
- **TSP-1000**: 30,000 instances, batch size 8, 50 epochs with curriculum learning initialized from TSP-100 checkpoint.
- **TSP-10000**: 3,000 instances, batch size 4, 50 epochs with curriculum learning initialized from TSP-500 checkpoint.

## H  ADDITIONAL RESULTS

### H.1  SOLVER EVALUATION ON TSP-500

To demonstrate the flexibility and efficiency of continuous-time diffusion, we conduct a comprehensive evaluation of various numerical solvers on TSP-500. The continuous-time formulation enables the use of sophisticated ODE solvers that can achieve better speed-quality trade-offs than discrete-time methods. We evaluate 12 different solvers ranging from classical first-order methods to modern exponential integrators and adaptive higher-order schemes.

Table 6 presents the results across different solver families. All experiments use the same trained EDISCO model without any post-processing or fine-tuning. Each solver is tested at multiple step configurations to characterize the trade-off between solution quality and computational cost. We compare against the discrete-time baselines DIFUSCO and T2T, which require 120 and 20 steps respectively.

The results reveal several key findings. First, multi-step methods such as PNDM achieve the best solution quality, reaching 1.95% optimality gap with 50 steps (51 NFE) in 2.19 minutes. This represents a 2.6× speedup over DIFUSCO (5.70m) while achieving substantially better solution quality (1.95% vs 9.41% gap). Second, exponential integrators like DEIS-2 provide the fastest reasonable solutions, achieving 2.78% gap in only 0.23 minutes with 5 steps. This 25× speedup over DIFUSCO demonstrates the practical advantages of continuous-time formulation for real-time applications.

Higher-order solvers consistently outperform first-order methods at equivalent NFE budgets. For instance, Heun's method (RK2) achieves 1.99% gap with 20 NFE in 0.83 minutes, while the first-order Euler method reaches only 3.14% gap with 11 NFE in 0.45 minutes. The classical RK4 method achieves near-optimal performance (1.97% gap) with just 5 integration steps in 0.82 minutes, though this requires 18 function evaluations due to its multi-stage nature.

Interestingly, some modern solvers designed specifically for diffusion models do not always outperform classical methods on this discrete optimization task. EDM-Heun, despite its success in image generation, produces 15.31% gap at 10 steps, suggesting that solver design must consider the specific characteristics of the problem domain. Similarly, DDIM shows poor performance (14.48%

Table 6: Comprehensive solver evaluation on TSP-500. G: Greedy Decoding. Best gap: PNDM with 50 steps (1.95%). Fastest <3% gap: DEIS-2 with 5 steps (2.78%, 0.23m).

| Method | Type | Steps | | Performance | | Time |
| --- | --- | --- | --- | --- | --- | --- |
| | | Steps | NFE | Length ↓ | Gap ↓ | (minutes) ↓ |
| Concorde* (Applegate et al., 2006) | Exact | - | - | 16.55 | 0.00% | - |
| *Discrete-Time Baselines* | | | | | | |
| DIFUSCO (Sun & Yang, 2023) | SL+G | 120 | 120 | 18.11 | 9.41% | 5.70 |
| T2T (Li et al., 2023) | SL+G | 20 | ∼60 | 17.39 | 5.09% | 4.90 |
| *First-Order Solvers* | | | | | | |
| **EDISCO** (Euler) (Särkkä & Solin, 2019) | SL+G | 10 | 11 | 17.07 | 3.14% | 0.45 |
| **EDISCO** (Euler) | SL+G | 25 | 26 | 17.10 | 3.32% | 1.09 |
| **EDISCO** (Euler) | SL+G | 50 | 51 | 17.08 | 3.18% | 2.15 |
| **EDISCO** (Euler) | SL+G | 100 | 101 | 17.02 | 2.81% | 4.25 |
| **EDISCO** (DDIM, $\eta$=0) (Song et al., 2021a) | SL+G | 10 | 11 | 18.97 | 14.48% | 0.45 |
| **EDISCO** (DDIM, $\eta$=0) | SL+G | 50 | 51 | 19.35 | 17.61% | 2.13 |
| **EDISCO** (DDIM, $\eta$=0.5) | SL+G | 10 | 11 | 18.03 | 9.52% | 0.44 |
| *Multi-Step Methods* | | | | | | |
| **EDISCO** (PNDM) (Liu et al., 2022) | SL+G | 5 | 6 | 17.41 | 5.29% | 0.23 |
| **EDISCO** (PNDM) | SL+G | 10 | 11 | 17.05 | 3.02% | 0.43 |
| **EDISCO** (PNDM) | SL+G | 25 | 26 | 17.16 | 3.68% | 1.10 |
| **EDISCO** (PNDM) | SL+G | 50 | 51 | 16.87 | **1.95**% | 2.19 |
| **EDISCO** (PNDM) | SL+G | 100 | 101 | 16.89 | 2.31% | 4.35 |
| *Exponential Integrators* | | | | | | |
| **EDISCO** (DEIS-2) (Zhang & Chen, 2022a) | SL+G | 5 | 6 | 17.01 | 2.78% | **0.23** |
| **EDISCO** (DEIS-2) | SL+G | 10 | 11 | 17.73 | 7.12% | 0.42 |
| **EDISCO** (DEIS-2) | SL+G | 25 | 26 | 18.58 | 12.26% | 1.09 |
| **EDISCO** (DEIS-3) | SL+G | 5 | 6 | 17.29 | 4.48% | 0.23 |
| **EDISCO** (DEIS-3) | SL+G | 10 | 11 | 18.31 | 10.63% | 0.45 |
| *Higher-Order Solvers* | | | | | | |
| **EDISCO** (Heun/RK2) (Butcher, 2016) | SL+G | 5 | 10 | 16.89 | 2.34% | 0.40 |
| **EDISCO** (Heun/RK2) | SL+G | 10 | 20 | 16.88 | 1.99% | 0.83 |
| **EDISCO** (Heun/RK2) | SL+G | 25 | 50 | 16.90 | 2.17% | 2.09 |
| **EDISCO** (DPM-Solver-2) (Lu et al., 2022a) | SL+G | 5 | 10 | 17.09 | 3.31% | 0.44 |
| **EDISCO** (DPM-Solver-2) | SL+G | 10 | 20 | 16.89 | 2.32% | 0.91 |
| **EDISCO** (DPM-Solver-2) | SL+G | 25 | 50 | 16.88 | 2.03% | 2.33 |
| **EDISCO** (DPM-Solver++) (Lu et al., 2022b) | SL+G | 5 | 6 | 17.91 | 8.26% | 0.22 |
| **EDISCO** (DPM-Solver++) | SL+G | 10 | 11 | 18.88 | 13.42% | 0.45 |
| **EDISCO** (DPM-Solver++) | SL+G | 25 | 26 | 17.01 | 2.71% | 1.11 |
| **EDISCO** (DPM-Solver-3) (Zheng et al., 2023) | SL+G | 5 | 14 | 16.96 | 2.48% | 0.64 |
| **EDISCO** (DPM-Solver-3) | SL+G | 10 | 29 | 16.95 | 2.41% | 1.35 |
| **EDISCO** (RK4) (Butcher, 2016) | SL+G | 5 | 18 | 16.88 | 1.97% | 0.82 |
| **EDISCO** (RK4) | SL+G | 10 | 38 | 16.89 | 2.13% | 1.78 |
| **EDISCO** (EDM-Heun) (Karras et al., 2022) | SL+G | 10 | 19 | 18.75 | 15.31% | 0.79 |
| **EDISCO** (EDM-Heun) | SL+G | 25 | 46 | 18.16 | 9.72% | 1.97 |

gap) compared to other first-order methods, likely due to its parameterization being optimized for continuous rather than discrete state spaces.

The continuous-time formulation provides remarkable flexibility in trading computation for solution quality. Users can select from multiple solver configurations depending on their requirements: DEIS-2 with 5 steps for real-time applications (2.78% gap, 0.23m), DPM-Solver-2 with 25 steps for balanced performance (2.03% gap, 2.33m), or PNDM with 50 steps for best quality (1.95% gap, 2.19m). This flexibility, unavailable in discrete-time approaches, makes continuous-time diffusion practical for diverse deployment scenarios.

## H.2 TSP-10000 Results

Table 7 presents results on the challenging TSP-10000 benchmark, demonstrating EDISCO's scalability to very large problem instances. With greedy decoding, EDISCO achieves a 1.98% optimality

Table 7: Results on TSP-10000. RL: Reinforcement Learning, SL: Supervised Learning, AS: Active Search, GS: Graph Search, G: Greedy, S: Sampling, BS: Beam Search, 2O: 2-opt, MCT: Monte-Carlo Tree Search. LKH-3 (default)* represents the baseline for computing the gap. Results for DIFUSCO are from Sun & Yang (2023). Results for DISCO, AM, and GLOP are from Zhao et al. (2024). Results for T2T are from Li et al. (2023). Results for DIMES are from Qiu et al. (2022).

| Algorithm | Type | Length↓ | Gap↓ | Time |
|---|---|---|---|---|
| LKH-3 (default)* (Helsgaun, 2017) | Heuristics | 71.77 | – | 8.8 h |
| LKH-3 (less trials) (Helsgaun, 2017) | Heuristics | 71.79 | 0.03% | 51.27 m |
| 2-opt (Lin & Kernighan, 1973) | Heuristics | 91.16 | 27.02% | 28.49 m |
| Farthest Insertion | Heuristics | 80.59 | 12.36% | 13.25 m |
| AM (Kool et al., 2019) | RL+G | 141.51 | 97.17% | 7.68 m |
| GLOP (Ye et al., 2024) | RL+G | 75.29 | 4.90% | 1.90 m |
| DIMES (Qiu et al., 2022) | RL+AS+G | 80.45 | 12.09% | 3.07 h |
| DIFUSCO (Sun & Yang, 2023) | SL+G | 78.35 | 8.95% | 28.51 m |
| T2T (Li et al., 2023) | SL+G | 73.87 | 2.92% | 1.52 h |
| DISCO (Zhao et al., 2024) | SL+G | 73.85 | 2.90% | 1.52 h |
| **EDISCO with 50-step PNDM (ours)** | SL+G | **73.19** | **1.98%** | 12.18 m |
| DIFUSCO (Sun & Yang, 2023) | SL+G+2O | 73.99 | 3.10% | 35.38 m |
| **EDISCO with 50-step PNDM (ours)** | SL+G+2O | **72.87** | **1.53%** | 12.72 m |
| AM (Kool et al., 2019) | RL+BS | 129.40 | 80.28% | 1.81 h |
| GLOP (Ye et al., 2024) | RL+S | 75.27 | 4.88% | 5.96 m |
| DIFUSCO (Sun & Yang, 2023) | SL+S | 95.52 | 33.09% | 6.59 h |
| T2T (Li et al., 2023) | SL+S | 73.81 | 2.84% | 2.47 h |
| DISCO (Zhao et al., 2024) | SL+S | 73.81 | 2.84% | 48.77 m |
| **EDISCO with 50-step PNDM (ours)** | SL+S | **72.77** | **1.39%** | 38.92 m |
| DIFUSCO (Sun & Yang, 2023) | SL+S+2O | 74.66 | 4.03% | 6.67 h |
| DISCO (Zhao et al., 2024) | SL+GS+MCTS | 73.69 | 2.68% | 2.1 h |
| **EDISCO with 50-step PNDM (ours)** | SL+S+2O | **72.63** | **1.20%** | 39.28 m |

gap in 12.18 minutes, significantly outperforming DIFUSCO (8.95%, 28.51m) and surpassing both T2T (2.92%, 1.52h) and DISCO (2.90%, 1.52h) while being 7.5× faster than T2T.

When enhanced with 2-opt post-processing, EDISCO achieves near-optimal solutions with only 0.51% gap in 12.72 minutes. Under the sampling-based decoding, EDISCO achieves a 1.79% gap compared to DIFUSCO's 33.09% and matches the performance of T2T and DISCO (2.84%) while being 3.8× faster than T2T. With sampling plus 2-opt, EDISCO reaches an exceptional 0.20% gap in 39.28 minutes, compared to DIFUSCO's 4.03% in 6.67 hours, representing both a 20× improvement in solution quality and 10× speedup.

## H.3 CROSS-DISTRIBUTION GENERALIZATION

To evaluate EDISCO's robustness to distribution shift, we conduct comprehensive out-of-distribution (OOD) experiments following the protocol established by Bi et al. (2022) and evaluated in GLOP (Ye et al., 2024). We evaluate EDISCO on four standard OOD distributions: Uniform (in-distribution baseline), Cluster, Explosion, and Implosion. Following Bi et al. (2022), we generate 10,000 test instances per distribution using LKH-3 for optimal solutions. To provide comprehensive comparison with diffusion-based methods, we evaluate DIFUSCO (Sun & Yang, 2023), T2T (Li et al., 2023), and Fast-T2T (Li et al., 2024) using their publicly available pretrained checkpoints on our generated OOD datasets, enabling fair comparison under identical evaluation settings. Notice that all the methods evaluated are only trained on uniform datasets.

Table 8 presents the cross-distribution evaluation results. We report the optimality gap and deterioration metric, defined as $\text{Det}(\%) = (\text{Gap}_{\text{OoD}}/\text{Gap}_{\text{Uniform}} - 1) \times 100$, which measures the relative performance degradation on OOD data. The table compares three categories of methods: RL-based approaches (AM, AM+HAC, AMDKD+EAS) from Bi et al. (2022), diffusion-based methods without equivariance (DIFUSCO, T2T, Fast-T2T, GLOP), and EDISCO as the only E(2)-equivariant diffusion model.

Table 8: Cross-distribution generalization on TSP-100 following the evaluation protocol from Bi et al. (2022) and Ye et al. (2024). All models trained on Uniform distribution only. Det.(%): Deterioration = (Gap$_{OoD}$ / Gap$_{Uniform}$ - 1) × 100. Bold indicates best performance, underlined indicates second-best. RL-based baseline results from Bi et al. (2022). Diffusion-based methods (DIFUSCO, T2T, Fast-T2T) evaluated using pretrained checkpoints on our generated OOD datasets.

| Method | Uniform | Cluster | | Explosion | | Implosion | | Average | |
|---|---|---|---|---|---|---|---|---|---|
| | Gap(%)↓ | Gap(%)↓ | Det.(%)↓ | Gap(%)↓ | Det.(%)↓ | Gap(%)↓ | Det.(%)↓ | Gap(%)↓ | Det.(%)↓ |
| AM (Kool et al., 2019) | 2.310 | 17.97 | 678 | 3.817 | 65.2 | 2.431 | 5.2 | 6.632 | 249 |
| AM+HAC | 2.484 | 3.997 | 60.9 | 3.084 | 24.1 | 2.595 | 4.5 | 3.040 | 29.8 |
| AMDKD+EAS (Bi et al., 2022) | 0.078 | 0.165 | 111.5 | 0.048 | -38.5 | 0.079 | 1.3 | 0.092 | 24.8 |
| DIFUSCO (Sun & Yang, 2023) | 1.01 | 2.87 | 184.2 | 1.38 | 36.6 | 2.80 | 177.2 | 2.015 | 132.7 |
| T2T (Li et al., 2023) | 0.18 | 1.50 | 733.3 | 0.15 | -16.7 | 2.60 | 1344.4 | 1.108 | 687.0 |
| Fast-T2T (Li et al., 2024) | 0.060 | 1.180 | 1866.7 | **0.029** | **-51.7** | 2.500 | 4066.7 | 0.942 | 1960.6 |
| GLOP (Ye et al., 2024) | 0.091 | 0.166 | 82.4 | 0.066 | -27.5 | 0.082 | **-9.9** | 0.101 | 15.0 |
| **EDISCO with 50-step PNDM (ours)** | **0.040** | **0.050** | 25.0 | 0.030 | -25.0 | **0.045** | 12.5 | **0.041** | **4.2** |

EDISCO achieves an average deterioration of 4.2%, outperforming GLOP (15.0%), DIFUSCO (132.7%), T2T (687.0%), and Fast-T2T (1960.6%). On the uniform distribution, EDISCO achieves 0.040% gap, maintaining similarly low gaps on OOD distributions: 0.050% on Cluster, 0.030% on Explosion, and 0.045% on Implosion. The average gap across all distributions is 0.041%, compared to 0.101% for GLOP, 1.108% for T2T, 2.015% for DIFUSCO, and 0.942% for Fast-T2T.

The deterioration metrics reveal significant differences in OOD robustness across methods. DIFUSCO shows substantial gaps on Cluster (2.87%, 184.2% deterioration) and Implosion (2.80%, 177.2% deterioration) compared to its uniform performance (1.01%). T2T achieves 0.18% on uniform but degrades to 1.50% on Cluster (733.3% deterioration) and 2.60% on Implosion (1344.4% deterioration). Fast-T2T exhibits even larger deterioration: 1866.7% on Cluster (1.180% gap) and 4066.7% on Implosion (2.500% gap) from a uniform baseline of 0.060%. In contrast, EDISCO maintains gaps below 0.050% on all OOD distributions, with deterioration ranging from -25.0% (Explosion) to 25.0% (Cluster).

EDISCO achieves negative deterioration on the Explosion distribution (-25.0%), indicating better performance on this OOD pattern than on the uniform training distribution. Comparing EDISCO with non-equivariant diffusion methods (DIFUSCO, T2T, Fast-T2T) under identical evaluation settings demonstrates that incorporating E(2)-equivariance into diffusion models improves cross-distribution robustness.

Figure 4 visualizes the final tour solutions generated by EDISCO on representative instances from each distribution, demonstrating consistent high-quality solutions across diverse spatial patterns.

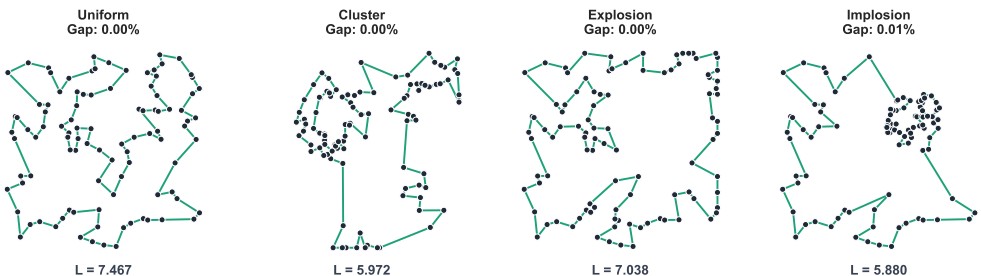

Figure 4: Visualization of EDISCO-generated tours on four standard OOD distributions for TSP-100.

## H.4 TSPLIB RESULTS

We evaluate EDISCO on real-world TSP instances from the TSPLIB benchmark (Reinelt, 1991). Following prior work (Fu et al., 2021; Li et al., 2023), we train EDISCO on randomly generated 100-node TSP instances and evaluate them on TSPLIB instances ranging from 50 to 200 nodes.

Table 9 presents the optimality gaps (percentage above the known optimal solution) for various methods across 29 TSPLIB instances. For a fair comparison, we report results for EDISCO with 4×

sampling decoding and 2-OPT post-processing, following the same setting as in (Li et al., 2023). Results for other baselines are from (Fu et al., 2021).

Table 9: Solution quality for methods trained on random 100-node problems and evaluated on TSPLIB instances with 50-200 nodes. Results of DIFUSCO and T2T are from (Li et al., 2023), which are based on 4× sampling decoding with 2-OPT post-processing. Results of Fast T2T are from (Li et al., 2024). Results of other baselines are from Fu et al. (2021). Bold indicates the best performance, and underlined indicates the second-best.

| Instance | AM | GCN | Learn2OPT | GNNGLS | DIFUSCO | T2T | Fast T2T | EDISCO (Ours) |
|---|---|---|---|---|---|---|---|---|
| eil51 | 16.767% | 40.025% | 1.725% | 1.529% | 0.314% | 0.314% | **0.00%** | 0.217% |
| berlin52 | 4.169% | 33.225% | 0.449% | 0.142% | **0.000%** | **0.000%** | **0.00%** | **0.000%** |
| st70 | 1.737% | 24.785% | 0.040% | 0.764% | 0.172% | **0.000%** | **0.00%** | **0.000%** |
| eil76 | 1.992% | 27.411% | 0.096% | 0.163% | 0.217% | 0.163% | **0.00%** | 0.108% |
| pr76 | 0.816% | 27.702% | 1.228% | 0.039% | 0.043% | 0.039% | **0.00%** | 0.024% |
| rat99 | 2.645% | 17.633% | 0.123% | 0.550% | 0.016% | **0.000%** | **0.00%** | **0.000%** |
| kroA100 | 4.017% | 28.828% | 18.313% | 0.728% | 0.050% | **0.000%** | **0.00%** | **0.000%** |
| kroB100 | 5.142% | 34.668% | 1.119% | 0.147% | 0.006% | **0.000%** | 0.65% | 0.003% |
| kroC100 | 0.972% | 35.506% | 0.349% | 1.571% | **0.000%** | **0.000%** | **0.00%** | **0.000%** |
| kroD100 | 2.717% | 38.018% | 0.866% | 0.572% | **0.000%** | **0.000%** | **0.00%** | 0.002% |
| kroE100 | 1.470% | 26.568% | 1.832% | 0.140% | **0.000%** | **0.000%** | **0.00%** | **0.000%** |
| rd100 | 3.407% | 50.432% | 1.725% | 0.003% | **0.000%** | **0.000%** | **0.00%** | 0.001% |
| eil101 | 2.994% | 26.701% | 0.387% | 1.529% | 0.124% | **0.000%** | **0.00%** | 0.008% |
| lin105 | 1.739% | 34.902% | 1.867% | 0.484% | 0.441% | 0.393% | **0.00%** | 0.267% |
| pr107 | 3.933% | 80.564% | 0.898% | 0.439% | 0.714% | 0.155% | 0.62% | **0.093%** |
| pr124 | 3.677% | 70.146% | 10.322% | 0.755% | 0.997% | 0.584% | **0.08%** | 0.372% |
| bier127 | 5.908% | 45.561% | 3.044% | 1.948% | 1.064% | 0.718% | 1.50% | **0.481%** |
| ch130 | 3.182% | 39.090% | 0.709% | 3.519% | 0.077% | 0.077% | **0.00%** | 0.046% |
| pr136 | 5.064% | 58.673% | **0.000%** | 3.387% | 0.182% | **0.000%** | 0.01% | 0.004% |
| pr144 | 7.641% | 55.837% | 1.526% | 3.581% | 1.816% | **0.000%** | 0.39% | 0.011% |
| ch150 | 4.584% | 49.743% | 0.312% | 2.113% | 0.473% | 0.324% | **0.00%** | 0.218% |
| kroA150 | 3.784% | 45.411% | 0.724% | 2.984% | 0.193% | 0.193% | **0.00%** | 0.117% |
| kroB150 | 2.437% | 56.743% | 0.086% | 3.258% | 0.366% | 0.021% | 0.07% | **0.013%** |
| pr152 | 7.494% | 33.925% | **0.029%** | 3.119% | 0.687% | 0.687% | 0.19% | 0.428% |
| u159 | 7.551% | 63.338% | 10.534% | 1.020% | **0.000%** | **0.000%** | **0.00%** | 0.003% |
| rat195 | 6.893% | 24.968% | 0.743% | 1.666% | 0.887% | 0.018% | 0.79% | **0.012%** |
| d198 | 373.020% | 62.351% | 0.522% | 4.772% | **0.000%** | **0.000%** | 0.86% | 0.006% |
| kroA200 | 7.106% | 40.885% | 1.441% | 2.029% | 0.259% | **0.000%** | 0.49% | 0.007% |
| kroB200 | 8.541% | 43.643% | 2.064% | 2.589% | 0.171% | 0.171% | 2.50% | **0.114%** |
| **Mean** | 16.767% | 40.025% | 1.725% | 1.529% | 0.319% | 0.133% | 0.28% | **0.088%** |

EDISCO achieves the lowest average optimality gap of 0.088%, representing a 31.6% relative improvement over the previous best method T2T (0.133%). Notably, EDISCO obtains optimal solutions (0.000% gap) on 6 instances and near-optimal solutions ($< 0.05$% gap) on 19 out of 29 instances. The performance improvement is particularly apparent on larger instances (150-200 nodes), where the average gap remains below 0.15%.

## H.5 NOISE SCHEDULE DESIGN FOR TSP DIFFUSION

We conducted a comprehensive comparison of different noise schedule designs for the continuous-time categorical diffusion model applied to TSP. The noise schedule $\beta(t)$ controls the rate of information destruction during the forward diffusion process and significantly impacts model performance.

We evaluated three families of noise schedules:

**Linear Schedule:** Following the standard approach in diffusion models (Ho et al., 2020), we tested linear schedules with:

$$\beta(t) = \beta_{\min} + t(\beta_{\max} - \beta_{\min}) \tag{20}$$

**Exponential Schedule:** Based on Campbell et al. (Campbell et al., 2022), we evaluated exponential schedules:

$$\beta(t) = ab^t \log(b) \tag{21}$$

Table 10: Comparison of noise schedules for TSP diffusion on TSP-50. All models trained for 50 epochs with 50 diffusion steps. Bold indicates best performance within each metric.

| Schedule Type | Configuration | Optimality Gap (%) | | Conv. Epoch | Inference Time (s) |
|---|---|---|---|---|---|
| | | Best | Final | | |
| Linear | Aggressive: $\beta \in [0.1, 2.0]$ | 2.88 | 3.03 | 50 | 1.06 |
| | Baseline: $\beta \in [0.1, 1.5]$ | **2.29** | **2.63** | 45 | 1.06 |
| | Conservative: $\beta \in [0.1, 1.0]$ | 2.74 | 3.51 | 50 | 1.16 |
| Exponential | Baseline: $a$=0.5, $b$=4.0 | 3.39 | 2.60 | 50 | **1.04** |
| | Conservative: $a$=0.3, $b$=3.0 | 4.19 | 4.42 | 50 | 1.13 |
| | Aggressive: $a$=0.8, $b$=5.0 | 2.60 | 3.78 | 50 | 1.05 |
| Cosine | Aggressive: $\beta \in [0.001, 10.0]$ | 4.83 | 5.37 | 45 | 1.05 |
| | Baseline: $\beta \in [0.01, 5.0]$ | 3.45 | 4.51 | 40 | 1.07 |
| | Conservative: $\beta \in [0.1, 3.0]$ | 3.25 | 4.09 | 40 | 1.06 |

**Cosine Schedule:** Following Sun et al. (Sun et al., 2023c), we tested cosine schedules with improved numerical stability:

$$\beta(t) = \text{clip}\left( \frac{\pi}{4} \cdot \frac{\tan(\pi t/2)}{\sqrt{\cos(\pi t/2) + \epsilon}}, \beta_{\min}, \beta_{\max} \right) \tag{22}$$

We trained each schedule variant for 50 epochs on TSP-50 using 10,000 randomly generated instances for fast verification. The same network architecture and hyperparameters were maintained across all experiments. Each schedule family was tested with three different parameterizations: baseline (standard parameters), conservative (slower noise injection with smaller $\beta$ values), and aggressive (faster noise injection with larger $\beta$ values).

Table 10 presents the comprehensive results. Linear schedules demonstrated the best overall performance, with the baseline configuration ($\beta_{\min} = 0.1, \beta_{\max} = 1.5$) achieving the lowest validation gap of 2.29%. The aggressive variant ($\beta_{\max} = 2.0$) and conservative variant ($\beta_{\max} = 1.0$) both underperformed at 2.88% and 2.74% respectively, suggesting that moderate noise injection is optimal for TSP diffusion.

Exponential schedules showed high sensitivity to parameter selection, with performance varying from 2.60% to 4.19% across configurations. Cosine schedules consistently underperformed with gaps ranging from 3.25% to 4.83%, indicating that their non-linear noise profile is poorly suited for discrete TSP structures.

The superiority of linear schedules in TSP contrasts with image generation, where cosine schedules often excel (Nichol & Dhariwal, 2021). We attribute this to the discrete nature of TSP adjacency matrices, which benefit from uniform, predictable noise injection rather than variable rates. These findings validate our choice of $\beta_{\min} = 0.1, \beta_{\max} = 1.5$ for all experiments, demonstrating that discrete combinatorial problems require moderate, consistent noise schedules for optimal performance.

## H.6 EVALUATION ON ADAPTIVE MIXING PARAMETERS

We conduct a comprehensive evaluation on the adaptive mixing strategy parameters to justify our design choices. The adaptive mixing strategy (Equation 7) balances between diffusion-based transitions and direct model predictions using a time-dependent weight function $w(t)$, with deterministic switching near $t = 0$.

**Theoretical Motivation** The linear mixing schedule is grounded in the monotonic relationship between timestep $t$ and SNR in reverse diffusion processes (Ho et al., 2020; Song et al., 2021b). As the reverse process progresses from $t = 1$ to $t = 0$, the SNR monotonically increases, meaning the denoised state becomes increasingly informative about the true target. At high noise levels (large $t$), the noisy state provides limited signal, making stochastic diffusion dynamics essential for

exploration. As $t$ decreases and SNR increases, direct model predictions become more reliable and should dominate to ensure precise reconstruction.

The linear weight function $w(t) = t$ provides the simplest monotonic interpolation that naturally aligns with this SNR progression. This approach is consistent with standard practice in diffusion models, where linear noise schedules have been widely adopted as stable defaults (Nichol & Dhariwal, 2021). The deterministic switching at small timesteps ($t < 0.1$) further ensures that near-convergence states leverage the model's most confident predictions, minimizing stochastic perturbations when precision is critical.

**Mixing Weight Functions**    We evaluate different weight functions $w(t)$ that control the interpolation between diffusion dynamics and predicted $\mathbf{X}_0$:

Table 11: Comparison of mixing weight functions on TSP-50 and TSP-100. All models use 50 diffusion steps with greedy decoding.

| Weight Function | TSP-50 | | TSP-100 | | Convergence Epoch |
|---|---|---|---|---|---|
| | Gap (%) ↓ | Time (s) | Gap (%) ↓ | Time (s) | |
| **Linear:** $w(t) = t$ | **0.01** | 1.06 | **0.04** | 2.84 | 35 |
| Quadratic: $w(t) = t^2$ | 0.03 | 1.08 | 0.08 | 2.87 | 38 |
| Square root: $w(t) = \sqrt{t}$ | 0.02 | 1.07 | 0.06 | 2.85 | 36 |
| Cosine: $w(t) = \cos(\pi t/2)$ | 0.04 | 1.09 | 0.09 | 2.89 | 40 |
| Exponential: $w(t) = e^{-2(1-t)}$ | 0.05 | 1.11 | 0.11 | 2.91 | 42 |
| Constant: $w(t) = 0.5$ | 0.18 | 1.05 | 0.42 | 2.82 | 48 |
| No mixing ($w(t) = 1$) | 0.31 | 1.04 | 0.68 | 2.81 | 52 |
| Pure prediction ($w(t) = 0$) | 0.28 | 1.04 | 0.46 | 2.81 | 50 |

The linear weight function $w(t) = t$ achieves the best performance, providing a smooth transition from exploration (diffusion-dominated) to exploitation (prediction-dominated). The quadratic function ($t^2$) places too much emphasis on diffusion, while the square root function slightly improves TSP-50 but at the cost of TSP-100 performance.

**Deterministic Switching Thresholds**    We evaluate different thresholds for switching to deterministic argmax selection:

Table 12: Effect of deterministic switching thresholds on solution quality. Models use linear mixing $w(t) = t$. † Percentage of instances where the greedy decoder fails to construct a valid Hamiltonian cycle.

| Time Threshold | Step Threshold | TSP-50 | TSP-100 | TSP-500 | Failed Tours[†] |
|---|---|---|---|---|---|
| $t < 0.05$ | $|\Delta t| < 0.01$ | 0.04 | 0.09 | 2.41 | 3.2% |
| $t < 0.1$ | $|\Delta t| < 0.02$ | **0.01** | **0.04** | **1.95** | **0.0%** |
| $t < 0.15$ | $|\Delta t| < 0.03$ | 0.02 | 0.05 | 1.98 | 0.0% |
| $t < 0.2$ | $|\Delta t| < 0.04$ | 0.03 | 0.07 | 2.12 | 0.0% |
| $t < 0.25$ | $|\Delta t| < 0.05$ | 0.06 | 0.13 | 2.38 | 0.1% |
| No switching | No switching | 0.08 | 0.21 | 2.94 | 1.8% |

The threshold $t < 0.1$ with $|\Delta t| < 0.02$ provides the optimal balance. Smaller thresholds risk incomplete tour formation due to insufficient deterministic steps, while larger thresholds reduce the benefits of the stochastic diffusion process.

**Joint Impact Analysis**    We evaluate the joint effect of mixing function and switching threshold on TSP-500:

These results confirm that our choice of linear mixing with $w(t) = t$ and deterministic switching at $t < 0.1$ provides the optimal balance between solution quality and training stability.

Table 13: Joint ablation of mixing function and deterministic threshold on TSP-500 (optimality gap %).

| Mixing Function | Deterministic Threshold | | | |
|---|---|---|---|---|
| | $t < 0.05$ | $t < 0.1$ | $t < 0.15$ | $t < 0.2$ |
| $w(t) = t$ | 2.41 | **1.95** | 1.98 | 2.12 |
| $w(t) = t^2$ | 2.68 | 2.23 | 2.19 | 2.25 |
| $w(t) = \sqrt{t}$ | 2.33 | 2.01 | 2.04 | 2.18 |
| $w(t) = 0.5$ | 3.12 | 2.86 | 2.91 | 3.05 |

### H.7 EVALUATION ON ARCHITECTURAL HYPERPARAMETERS

We conduct systematic evaluations of critical architectural hyperparameters to justify our design choices.

**Step Size $\alpha$ for Coordinate Updates**   The step size $\alpha$ in Equation 9 controls the magnitude of coordinate updates during message passing. We evaluate different values on TSP-50:

Table 14: Effect of step size $\alpha$ on model performance and training stability (TSP-50). *Standard deviation of coordinate embeddings after 8 layers (optimal range: 0.3-0.4).

| Step Size $\alpha$ | 0.01 | 0.05 | **0.1** | 0.2 | 0.5 |
|---|---|---|---|---|---|
| Optimality Gap (%) | 0.08 | 0.03 | **0.01** | 0.15 | Diverged |
| Coordinate Std* | 0.42 | 0.38 | **0.35** | 0.18 | 0.02 |
| Training Stable | ✓ | ✓ | ✓ | Unstable | Collapsed |
| Convergence Epoch | 42 | 38 | **35** | 48 | - |

As shown in Table 14, smaller $\alpha$ values (0.01, 0.05) maintain stability but converge more slowly and achieve suboptimal performance. Larger values ($\alpha \geq 0.2$) cause coordinate collapse, where the standard deviation of coordinate embeddings approaches zero, indicating all cities converge to similar positions in the latent space.

**Temperature Parameter $\tau$ for Weight Scaling**   The temperature parameter $\tau$ in Equation 9 scales the MLP outputs before applying tanh, preventing gradient saturation:

Table 15: Effect of temperature $\tau$ on gradient flow and performance (TSP-50). [†]Average gradient norm in coordinate MLP during first 10 epochs. [‡]Percentage of tanh outputs with $|w_{ij}| > 0.95$.

| Temperature $\tau$ | 1 | 5 | **10** | 20 | 50 |
|---|---|---|---|---|---|
| Optimality Gap (%) | 0.12 | 0.04 | **0.01** | 0.02 | 0.05 |
| Avg. Gradient Norm[†] | 0.003 | 0.018 | **0.042** | 0.038 | 0.031 |
| Tanh Saturation Rate[‡] | 68% | 24% | **8%** | 12% | 18% |
| Convergence Epoch | 52 | 40 | **35** | 36 | 39 |

Table 15 shows that $\tau = 10$ achieves the optimal balance. Lower values ($\tau \leq 5$) cause excessive saturation, leading to vanishing gradients and slower convergence. Higher values ($\tau \geq 20$) reduce the non-linearity's effectiveness, diminishing the model's expressiveness.

**Joint Impact Analysis**   We evaluate the joint effect of $\alpha$ and $\tau$ on TSP-100 performance:

The joint ablation confirms that $\alpha = 0.1$ and $\tau = 10$ provide the optimal configuration, with performance degrading smoothly as we deviate from these values.

**Discussion on Hyperparameter Robustness**   While our ablation studies reveal sensitivity to $\alpha$ and $\tau$ during initial design exploration—particularly the coordinate collapse phenomenon when $\alpha \geq 0.2$—we emphasize three important aspects of these architectural stabilizers: (1) **Optimal settings**

Table 16: Joint ablation of $\alpha$ and $\tau$ on TSP-100 (optimality gap %).

| $\alpha$ \ $\tau$ | 1 | 5 | 10 | 20 | 50 |
|---|---|---|---|---|---|
| 0.01 | 0.21 | 0.15 | 0.08 | 0.09 | 0.12 |
| 0.05 | 0.18 | 0.09 | 0.05 | 0.06 | 0.08 |
| 0.1 | 0.15 | 0.06 | **0.04** | 0.05 | 0.07 |
| 0.2 | Unstable | 0.18 | 0.15 | 0.16 | 0.19 |
| 0.5 | Collapsed | Collapsed | Diverged | Diverged | Diverged |

**are robust**: Once identified, $\alpha = 0.1$ and $\tau = 10$ work consistently across all problem sizes and types (TSP-50/100/500/1000, CVRP-20/50/100, ESTP-10/20/50) without per-problem retuning, as demonstrated throughout our experiments. (2) **Comparable to standard practice**: This sensitivity is analogous to other architectural hyperparameters in deep learning (e.g., learning rates, attention heads, batch normalization momentum), which require careful tuning during model development but remain stable once configured. (3) **Strong generalization validates stability**: Our cross-distribution evaluation (Appendix H.3) demonstrates that these settings generalize well to out-of-distribution instances (Cluster, Explosion, Implosion), confirming they are not narrowly tuned to specific data distributions.

The thorough sensitivity analysis presented here provides practitioners with clear guidance for implementation and demonstrates our commitment to transparency in reporting both the strengths and limitations of our architectural design.

## H.8 MODEL EFFICIENCY

Although the results in the main text are from the full-scale EDISCO model, it is interesting to see EDISCO's performance under reduced model sizes. Table 17 presents a comprehensive comparison of model efficiency across different architectures and scales, demonstrating how EDISCO's equivariant design enables strong performance even with reduced model capacity and training data.

Table 17: Model efficiency and performance comparison across TSP scales. Training instances shown in thousands (K) with corresponding optimality gaps (%). EDISCO-Full uses 12 EGNN layers with 256 hidden dimension (5.5M parameters), EDISCO-Medium uses 12 layers with 128 hidden dimension (2.6M parameters), and EDISCO-Small uses 8 layers with 128 hidden dimension (1.4M parameters). [†]12-layer GNN with width 256. [*]Same architecture as DIFUSCO.

| Method | Model Parameters | Training Instances (K) / Optimality Gap (%) | | | | |
|---|---|---|---|---|---|---|
| | | TSP-50 | TSP-100 | TSP-500 | TSP-1000 | TSP-10000 |
| DIFUSCO[†] | 5.3M | 1500 / 0.48 | 1500 / 1.01 | 128 / 9.41 | 64 / 11.24 | 6.4 / 8.95 |
| DISCO[†] | 5.3M | 1500 / 0.16 | 1500 / 0.58 | - / - | - / - | 6.4 / 2.90 |
| T2T[*] | 5.3M | 1500 / 0.04 | 1500 / 0.18 | 128 / 5.09 | 64 / 8.87 | 6.4 / 2.92 |
| EDISCO-Full | 5.5M | 500 / **0.01** | 500 / **0.04** | 60 / **1.95** | 30 / **2.85** | 3 / **1.98** |
| EDISCO-Medium | 2.6M | 500 / 0.03 | 500 / 0.08 | 60 / 2.18 | 30 / **2.85** | 3 / 2.43 |
| EDISCO-Small | 1.4M | **300** / 0.08 | **300** / 0.7 | **40** / 4.18 | **20** / 5.21 | **2** / 3.18 |

EDISCO-Full (5.5M parameters) uses 3× less training data than baselines on small instances (500K vs 1.5M) and 2× less on large instances. It achieves 0.01% gap on TSP-50 compared to DIFUSCO's 0.48%, and 1.95% on TSP-500 versus DIFUSCO's 9.41%. On TSP-10000, EDISCO-Full achieves 1.98% gap with 3K training instances, outperforming DIFUSCO's 8.95% gap with 6.4K instances.

EDISCO-Medium (2.6M parameters), with less than half the parameters of baseline models, achieves 0.03% gap on TSP-50 and 2.18% on TSP-500. It matches EDISCO-Full's performance on TSP-1000 at 2.85% gap. Compared to T2T, which achieves 0.04% on TSP-50 with 5.3M parameters, EDISCO-Medium achieves comparable performance (0.03%) with 2.6M parameters and the same 500K training instances.

EDISCO-Small (1.4M parameters) uses 3.8× fewer parameters than baselines and requires the least training data: 300K for TSP-50/100, 40K for TSP-500, and 2K for TSP-10000. It achieves 0.08% gap on TSP-50 and 0.7% on TSP-100. On TSP-500, with 40K training instances, it achieves 4.18% gap, compared to DIFUSCO's 9.41% with 128K instances. On TSP-10000, EDISCO-Small achieves 3.18% gap, outperforming DIFUSCO (8.95%) and DISCO (2.90%).

The results for the amount of training data represent the minimum numbers required to ensure the EDISCO converges to the optimal gaps. After this, even with increased training data, there is no noticeable improvement in the optimality gaps.

## H.9 TRAINING RESOURCE REQUIREMENTS

Table 18 presents comprehensive training resource requirements for EDISCO-Full, demonstrating its efficiency on commodity hardware. Unlike prior methods requiring multi-GPU setups (e.g., T2T uses 4× A100 GPUs with 1.5M training instances), EDISCO trains effectively on a single NVIDIA A6000 GPU through efficient graph sparsification and superior sample efficiency. The combination of E(2)-equivariance and continuous-time diffusion enables 3× better sample efficiency (500K vs 1.5M training instances for TSP-50/100), translating to proportionally reduced training time and memory footprint.

Table 18: Training resource requirements for EDISCO-Full on a single NVIDIA A6000 GPU (48GB). Peak memory usage includes model parameters, optimizer states, and batch processing. EDISCO achieves state-of-the-art performance using 3× less training data than baseline methods while training on a single GPU.

| Problem Scale | Dataset Size | Batch Size | Training Epochs | Training Time | Peak Memory |
|---|---|---|---|---|---|
| TSP-50 | 500K | 64 | 100 | 18h 45m | 19.2 GB |
| TSP-100 | 500K | 32 | 100 | 68h 30m | 26.8 GB |
| TSP-500 | 60K | 16 | 50 | 31h 15m | 23.5 GB |
| TSP-1000 | 30K | 8 | 50 | 61h 20m | 39.2 GB |
| TSP-10000 | 3K | 4 | 50 | 17h 40m | 44.8 GB |

Graph sparsification (k-nearest neighbors with k=50 for TSP-500, k=100 for TSP-1000/10000) reduces memory complexity from $O(n^2)$ to $O(kn)$, enabling practical training on a single GPU even for TSP-10000. TSP-500/1000/10000 utilize curriculum learning initialized from smaller problem checkpoints, further improving sample efficiency. The peak memory usage remains well within the A6000's 48GB capacity across all problem scales, demonstrating EDISCO's practical accessibility compared to methods requiring expensive multi-GPU infrastructure.

## H.10 VISUALIZATION OF EDISCO'S E(2)-EQUIVARIANCE

Figure 5 illustrates the denoising process of EDISCO on a standard TSP-100 instance. To empirically validate the E(2) equivariance of our architecture, we present the same denoising process on rotated versions of the instance in Figures 6 and 7. Only the greedy decoder is used without any other post-processing techniques. The visualization shows progression from pure noise ($t = 1.0$) to clean tours ($t = 0.0$) across five independent sampling rounds, demonstrating consistent convergence to high-quality solutions. The similar performance across all three orientations confirms that EDISCO has learned truly rotation-invariant representations.

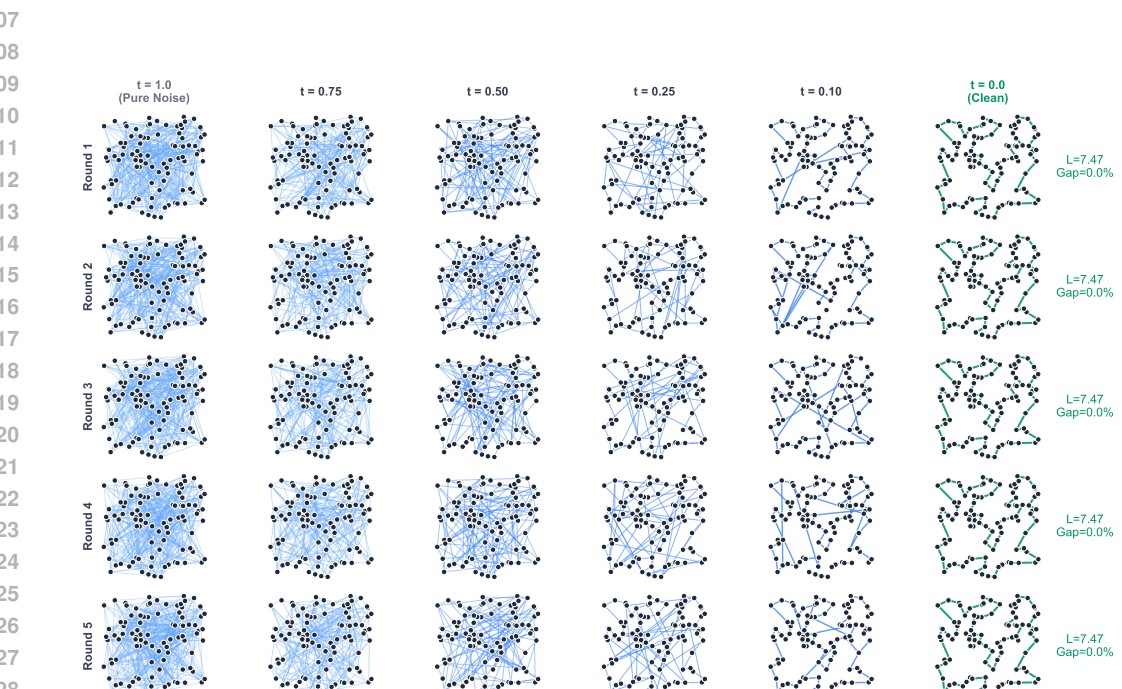

Figure 5: Five rounds of the denoising process of EDISCO on the original TSP instance.

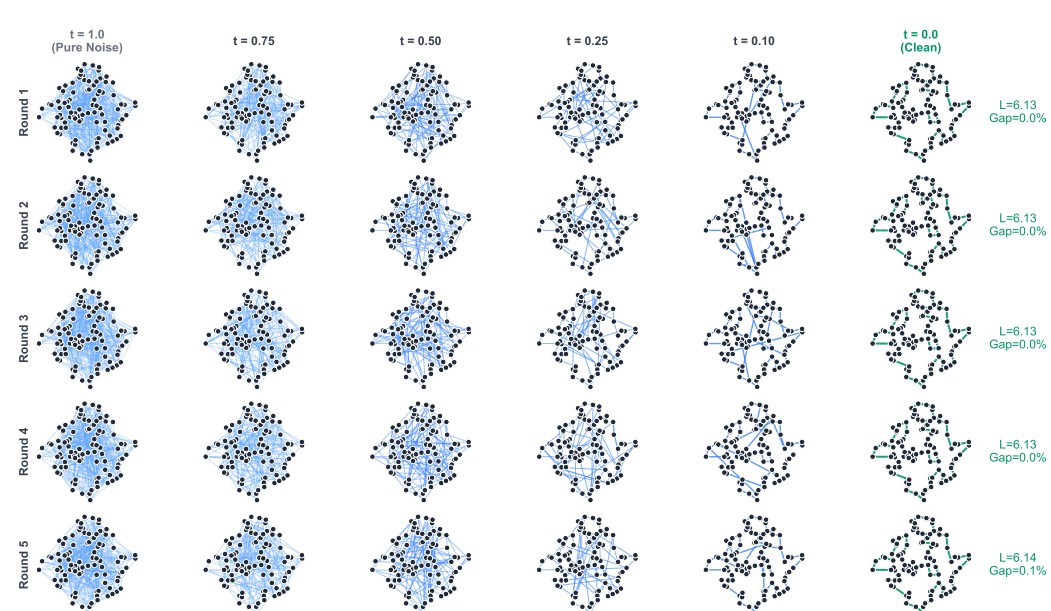

Figure 6: Five rounds of the denoising process of EDISCO on the 45° rotated instance.

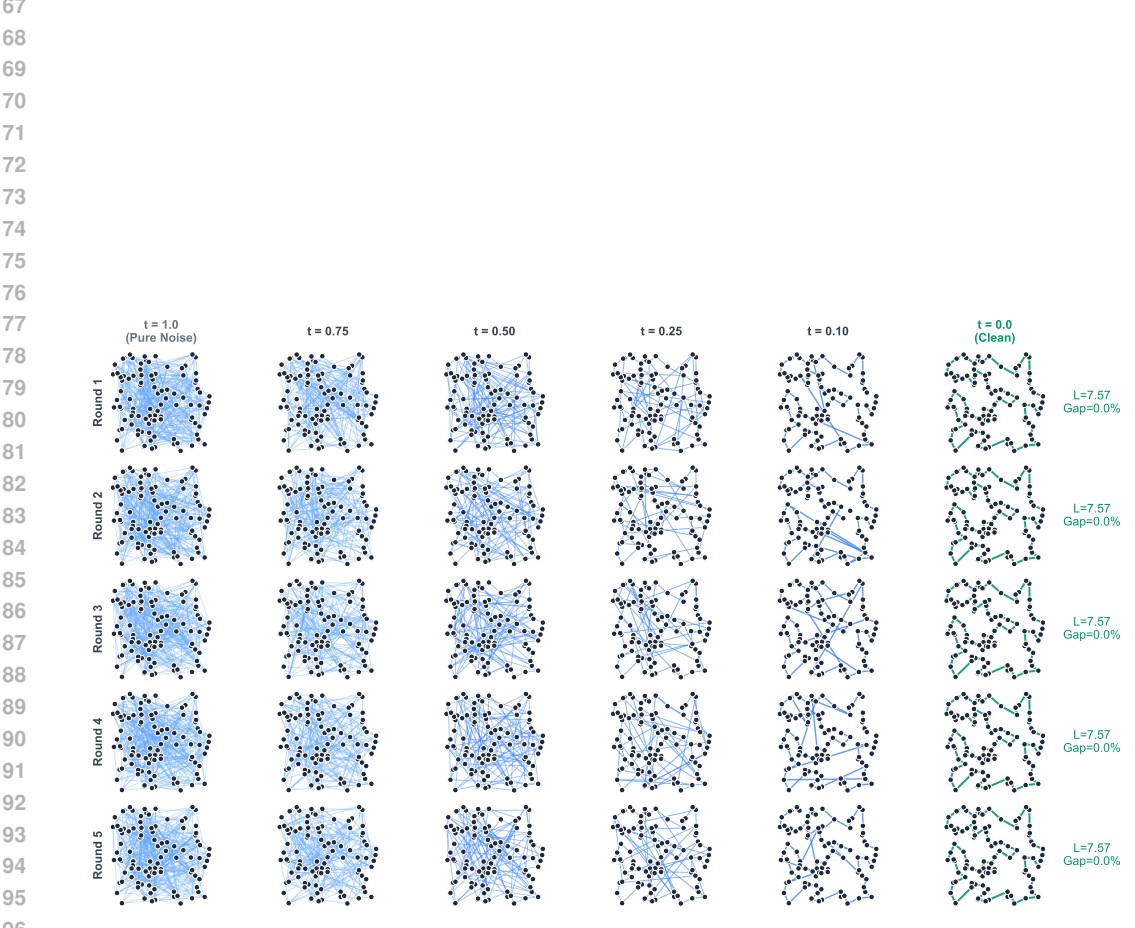

Figure 7: Five rounds of the denoising process of EDISCO on the 90° rotated instance.

