# OpenReview forum: "EDISCO: Equivariant Continuous-Time Categorical Diffusion for Geometric Combinatorial Optimization"
_ICLR.cc/2026/Conference — ICLR 2026 Conference Desk Rejected Submission_

### Official Review · Reviewer_ZQPE · 2025-10-19

**Soundness:** 2
**Presentation:** 3
**Contribution:** 2
**Rating:** 4
**Confidence:** 4

**Summary:**

This paper proposes EDISCO, a diffusion-based framework for Geometric Combinatorial Optimization Problems (GCOPs) (e.g., TSP) that integrates two key innovations: E(2)-equivariant Graph Neural Networks (EGNNs) and continuous-time categorical diffusion. E(2)-equivariance ensures the model respects geometric symmetries (rotations, translations, reflections) of GCOPs, while continuous-time diffusion (modeled as Continuous-Time Markov Chains, CTMCs) enables adaptive numerical solvers and avoids discrete-time approximation errors.

**Strengths:**

1. EDISCO combines E(2)-equivariance with continuous-time categorical diffusion for GCOPs, filling a gap where prior diffusion solvers ignore GCOPs' inherent symmetries, which is novel and practical.
2. The paper is well-structured and easy to follow.
3. The experimental studies on TSP are relatively thorough and the performance appears promising.

**Weaknesses:**

1. The evaluation scope is relatively limited. EDISCO is only evaluated on routing problems like TSP and CVRP. It would improve the quality of this work if the authors could provide some results on more types of GCOPs (**or at least a subset of**), e.g., node-focused tasks (MIS, Max Clique, Max Cut, Min Vertx Cover, etc., as studied by many NCO works [1-5]; note that DIMES/T2T/DIFUSCO (which you have included for comparison) have also evaluated their solvers on at least MIS), and VRPs with more complex constraints or asymmetric/binary TSPs. Also, current CVRP experiments only cover small scales (20–100 nodes) and lack comparison to more recent CVRP solvers like GLOP or PO[6], etc.
2. This work seems to have followed the research line of generative neural CO solving, e.g., DIFUSCO and T2T. To my knowledge, some follow-up works have emerged with stronger performance or more applicable geometric CO problems evaluated, e.g., Fast-T2T (Li et al. NeurIPS 2024), COExpander (Ma et al. ICML 2025), DiffUCO (Sanokowski et al. ICML 2024), etc. Also, the RL or local constructive/unsupervised field has also fostered approaches that are more advanced than DIMES and POMO, e.g., BQ-NCO (Drakulic et al. NeurIPS 2023), UTSP (Min et al. NeurIPS 2023), UniCO (Pan et al. ICLR 2025), GOAL (Drakulic et al. ICLR 2025), BOPO (Liao et al. ICML 2025), etc. I suggest the authors provide more comparative results (**or at least considerable discussions**) in this regard. I have listed several [6-13] below for your reference.
3. The main experiments are done on uniform TSP instances, while including (**at least a subset of**) supplementary evaluations on more diverse data distributions, e.g., Gaussian, cluster, explosive, implosion as in GLOP (Ye et al. AAAI 2024), are conducive to validating a better generalization capability of the proposed architecture and the training process. Also, I'm curious how EDISCO performs on ATSP cases, e.g., those defined in MatNet (Kwon et al. NeurIPS 2021), as 2D-coordinates are absent for input and the inherent symmetries are largely removed. Will these scenario shifts (occurring commonly in real-world applications) immensely harm the capability of the E(2)-equivariant tricks proposed in this paper?
4. Regarding the adaptive mixing strategy, the linear weight function $w(t)=t$ and deterministic switching threshold $t<0.1$ are chosen via empirical results like a grid search but lack rigorous/theoretical justification for why they outperform the alternatives (those listed  and beyond, e.g., other instance-aware adaptive strategies (e.g., adjusting $w(t)$ based on node density or problem scale)).
5. For the proposed equivariant architecture, it would be great if the authors apply it to replace the vanilla GNNs used in previous solvers so as to demonstrate the exact performance gain by advancing the neural backbone.

References:

[1] A Diffusion Model Framework for Unsupervised Neural Combinatorial Optimization (DiffUCO, ICML 2024)

[2] Regularized Langevin Dynamics for Combinatorial Optimization (RLSA, ICML 2025)

[3] Revisiting Sampling for Combinatorial Optimization (iSCO, ICML 2023)

[4] Variational Annealing on Graphs for Combinatorial Optimization (VAG-CO, NeurIPS 2023)

[5] Let the Flows Tell: Solving Graph Combinatorial Optimization Problems with GFlowNets (GFlowNets, NeurIPS 2023)

[6] Preference Optimization for Combinatorial Optimization Problems (PO, ICML 2025)

[7] Fast T2T: Optimization Consistency Speeds Up Diffusion-Based Training-to-Testing Solving for Combinatorial Optimization  (NeurIPS 2024)

[8] COExpander: Adaptive Solution Expansion for Combinatorial Optimization (ICML 2025)

[9] BQ-NCO: Bisimulation Quotienting for Efficient Neural Combinatorial Optimization (NeurIPS 2023)

[10] Unsupervised Learning for Solving the Travelling Salesman Problem (UTSP, NeurIPS 2023)

[11] UniCO: On Unified Combinatorial Optimization via Problem Reduction to Matrix-Encoded General TSP (ICLR 2025)

[12] GOAL: A Generalist Combinatorial Optimization Agent Learner (ICLR 2025)

[13] BOPO: Neural Combinatorial Optimization via Best-anchored and Objective-guided Preference Optimization (ICML 2025)

**Questions:**

Please refer to the Weaknesses part where I've listed my major concerns.

---

> ### Author Response · Authors · 2025-11-20
> **Response to Reviewer ZQPE's Weakness 1**
>
> ## We would like to thank the reviewer for the detailed and insightful comments. We have thoroughly revised the manuscript to address your concerns (changes marked in blue).
>
> ## **Response to Weakness 1:**
>
> ### (1) Node-Focused Geometric Tasks: Euclidean Steiner Tree
>
> We now include extensive experiments on the **Euclidean Steiner Tree Problem (ESTP)**, demonstrating that EDISCO applies beyond routing to *node-selection* geometric optimization. Unlike TSP, ESTP requires deciding which candidate Steiner points to include to minimize tree length.
>
> EDISCO substantially outperforms both classical heuristics and learning-based baselines (DIFUSCO, T2T, Fast-T2T) across all problem sizes (Steiner-10/20/50):
>
> | Method | Steiner-10 Gap | Steiner-20 Gap | Steiner-50 Gap |
> |--------|----------------|----------------|----------------|
> | GeoSteiner* (Exact) | 0.00% | 0.00% | 0.00% |
> | Steiner Insertion | 3.44% | 3.58% | 3.29% |
> | Deep-Steiner | 4.20% | 4.96% | 5.19% |
> | DIFUSCO | 5.73% | 4.96% | 4.67% |
> | T2T | 4.20% | 3.58% | 3.29% |
> | Fast-T2T | 2.29% | 2.20% | 2.25% |
> | **EDISCO (Ours)** | **1.53%** | **1.38%** | **1.56%** |
>
> Please see **Appendix B** (Table 4).
>
> ---
>
> ### (2) Non-Geometric Problems (MIS, Max-Cut): Clarification of Scope
>
> We clarify that **MIS, Max-Cut, Max Clique, and MVC are *not* geometric combinatorial optimization problems**. These operate on arbitrary graphs where only topology matters. EDISCO is designed explicitly for geometric problems where solutions exhibit E(2) symmetries in Euclidean space. Without geometric structure, E(2)-equivariance provides no inductive bias, offering no advantage over general-purpose diffusion methods (DIFUSCO, T2T) while unnecessarily introducing coordinate processing.
>
> We believe this might not be a weakness. In fact, DIFUSCO's authors themselves identified **"explore the use of equivariant graph neural networks for further improvement of the diffusion models on geometrical NP-complete combinatorial optimization problems such as Euclidean TSP"** as a promising future direction. This specialization yields **specialized benefits** for geometric problems.
>
> Please see **Appendix A: Scope and Limitations**.
>
> ---
>
> ### (3) CVRP: Added Baselines, Positioning, and Clarifications
>
> We have strengthened the CVRP section in **Appendix C** of the revised manuscript in three ways:
>
> 1. **Added state-of-the-art baseline:** We now include **HGS** [1], the leading CVRP heuristic, with reported optimal solutions for CVRP-20/50/100, and include missing runtime information.
>
> 2. **Clarified positioning relative to large-scale CVRP solvers:** We discuss GLOP [2] and NeuroLKH [3], which scale to 1000–7000 customers using partition-based strategies. Our CVRP scope (20-100 customers) is positioned as proof-of-concept for architectural extensibility to constrained routing, not as competing with large-scale specialists. Notably, competing diffusion methods (DIFUSCO, T2T, Fast-T2T) have no reported CVRP experiments, highlighting the difficulty of scaling single end-to-end diffusion models. We have actively discussed future directions in **Appendix A** in the revised manuscript for applying the E(2) Equivariance to partition-based strategies similar to GLOP [2] to scale to large-scale instances (1000+ customers).
>
> 3. **PO clarification:** Our study shows that PO [4] is a training algorithm, not a solver architecture. It can be applied to POMO, Sym-NCO, AM to improve training. Including it in our CVRP comparison would not be appropriate, as we are comparing solvers, not training methods.
>
> ---
>
> ### (4) Asymmetric TSP (ATSP)
>
> We clarify that **EDISCO does not support ATSP** due to an **architectural limitation** specific to E(2)-equivariant methods: EDISCO's E(2)-equivariant EGNN fundamentally **requires Euclidean coordinates** to preserve geometric symmetries. This is incompatible with ATSP's coordinate-free **arbitrary asymmetric cost matrices**.
>
> This limitation is shared by coordinate-based methods (AM, POMO, Pointer Networks, DIFUSCO, T2T, Fast-T2T). Current ATSP solvers (MatNet [5], GREAT [6]) incorporate directional edge information fundamentally incompatible with E(2)-equivariant coordinate encoders.
>
> Again, we believe that this may not be a weakness. We have added **Appendix A: Scope and Limitations** in the revised manuscript, which clarifies that EDISCO is specialized for Euclidean geometric problems. We have also actively discussed two **promising future directions** in **Appendix A**: (1) replacing the encoder with ATSP-capable architectures (MatNet [5], GREAT [6]) while retaining continuous-time diffusion, or (2) learning coordinate embeddings from asymmetric cost matrices using **Finsler Multi-Dimensional Scaling** [7] (extends MDS to handle asymmetric dissimilarities via Finsler spaces) or neural metric learning to enable approximate E(2)-equivariance for ATSP instances with near-geometric structure.

---

> ### Author Response · Authors · 2025-11-20
> **Response to Reviewer ZQPE's Weakness 2&3**
>
> ## Thank you for these valuable comments. We have revised the manuscript to address your concerns (changes marked in blue).
>
> ## **Response to Weakness 2:**
>
> We sincerely appreciate the reviewer for highlighting these important recent related works. We have substantially strengthened our manuscript by adding additional **baselines** and **discussions** of these methods.
>
> **1. Baselines Added:**
>
> We have included **Fast-T2T** and **BQ-NCO** [8] as new baselines in our experimental results:
>
> - **Fast-T2T:** Added to **Table 1** (TSP-50/100), **Table 2** (TSP-500/1000), **Appendix H3: Cross-Distribution Generalization** (Evaluation across Uniform/Cluster/Explosion/Implosion distributions of TSP), and **Appendix H4: TSPLIB Evaluation**.
>
> - **BQ-NCO:** Added to **Table 1** (TSP-100) and **Table 2** (TSP-500/1000), demonstrating its bisimulation quotienting approach.
>
> **EDISCO achieves state-of-the-art performance** across all benchmarks:
>
> | Method | TSP-500 Gap | TSP-1000 Gap |
> |--------|-------------|--------------|
> | DIFUSCO | 9.41% | 11.24% |
> | T2T | 5.09% | 8.87% |
> | Fast-T2T | 5.94% | 6.29% |
> | BQ-NCO | 5.22% | 8.97% |
> | **EDISCO (Ours)** | **1.95%** | **2.85%** |
>
> Please see the revised manuscript for detailed numerical comparisons.
>
> **2. Discussions Added:**
>
> We have significantly expanded our Related Work coverage in both the **Related Works (Section 2 in main text)** and **Extended Related Work in the appendix**:
>
> 1. **Diffusion-based methods (Section 2.1):** We now discuss **COExpander** [9]'s adaptive expansion method that scales to 10K nodes, **DiffUCO** [10]'s unsupervised continuous-time diffusion for graph problems. It further proves that EDISCO is the first to combine E(2)-equivariance with continuous-time categorical diffusion for geometric optimization.
>
> 2. **Sampling and variational approaches (Section 2.3):** We added discussions of **iSCO** [11]'s MCMC sampling, **RLSA** [12]'s regularized Langevin dynamics, **VAG-CO** [13]'s variational annealing, and **GFlowNets** [14]'s flow-based generative models. We clarify that these methods target non-geometric graph problems.
>
> 3. **RL methods (Appendix - Foundational NCO subsection):** We discuss **PO** [4] and **BOPO** [15] as recent RL training innovations using preference-based optimization, clarifying that these are improved training algorithms.
>
> 4. **Unsupervised and alternative paradigms (Section 2.1 & Appendix):** We added **UTSP** [16]'s data-efficient unsupervised learning method.
>
> 5. **Generalist approaches (Appendix - Alternative Architectures subsection):** We discuss **UniCO** [17]'s problem reduction strategy and **GOAL** [18]'s multi-task generalist model.
>
> **These Additional Related Works Strengthened EDISCO's Contributions:**
>
> EDISCO's unique contribution—**combining exact E(2)-equivariance with continuous-time categorical diffusion**—is clearly distinguished from:
> - Diffusion methods lacking geometric awareness (DiffUCO, DISCO)
> - Discrete-time diffusion formulations (DIFUSCO, T2T)
> - Alternative paradigms (COExpander's expansion, RL-based methods)
> - Unsupervised approaches (UTSP, DiffUCO)
> - Generalist solvers sacrificing specialized geometric inductive bias (UniCO, GOAL)
>
> Please see the **revised manuscript** for detailed comparisons in **Section 2 (Related Work)**, **Tables 1-2**, **Appendix: Cross-Distribution Generalization**, **Appendix: TSPLIB Evaluation**, and **Extended Related Work**.
>
> ---
>
> ## **Response to Weakness 3:**
>
> We sincerely appreciate the reviewer for these valuable suggestions.
>
> **1. Cross-Distribution Evaluation Added:**
>
> We have included extensive **out-of-distribution (OOD) experiments** following the evaluation protocol established by GLOP [2]: the **Uniform (training baseline)**, **Cluster**, **Explosion**, and **Implosion** are evaluated on TSP-100. EDISCO achieves the lowest average performance drop across all distributions:
>
> | Method | Uniform | Cluster | Explosion | Implosion | Avg Gap | Avg Det. |
> |--------|---------|---------|-----------|-----------|---------|----------|
> | DIFUSCO | 1.01% | 2.87% | 1.38% | 2.80% | 2.02% | 132.7% |
> | T2T | 0.18% | 1.50% | 0.15% | 2.60% | 1.11% | 687.0% |
> | Fast-T2T | 0.06% | 1.18% | 0.03% | 2.50% | 0.94% | 1960.6% |
> | GLOP | 0.09% | 0.17% | 0.07% | 0.08% | 0.10% | 15.0% |
> | **EDISCO** | **0.04%** | **0.05%** | **0.03%** | **0.05%** | **0.04%** | **4.2%** |
>
> Please see **Appendix H3: Cross-Distribution Generalization** (complete table and analysis).
>
> **2. ATSP Clarification:**
>
> This is detailed in our **Response to Weakness 1 (Section 4)**. We have also identified **promising future directions** in **Appendix A: Scope and Limitations** to: (1) replace the encoder with ATSP-capable architectures (MatNet [5], GREAT [6]), or (2) learn coordinate embeddings from asymmetric cost matrices using Finsler Multi-Dimensional Scaling [7] or neural metric learning to enable approximate E(2)-equivariance for ATSP instances with near-geometric structure.

---

> ### Author Response · Authors · 2025-11-20
> **Response to Reviewer ZQPE's Weakness 4&5**
>
> ## Thank you for these valuable comments. We have revised the manuscript to address your concerns (changes marked in blue).
> ---
>
> ## **Response to Weakness 4:**
>
> We sincerely appreciate this important observation. In the revised manuscript, we have added both empirical and theoretical justifications for our adaptive mixing strategy design:
>
> **1. Theoretical Motivation:**
>
> We now provide SNR-based theoretical justification **(added to Section 3.1 and Appendix H.6 "Theoretical Motivation" paragraph)** grounded in established diffusion model literature. Reverse diffusion exhibits a **monotonically increasing signal-to-noise ratio (SNR)** as time decreases [19]. The linear schedule $w(t) = t$ provides the simplest monotonic interpolation that aligns with this SNR progression. It favors stochastic exploration at high-noise timesteps and deterministic exploitation when the model's predictions become more reliable. This behavior is consistent with standard practice in diffusion models, where **linear noise schedules** remain widely adopted defaults due to their stability and simplicity.
>
> **2. Comprehensive Empirical Validation:**
>
> As the original submission already presented, **Appendix H.6** evaluates seven candidate schedules (linear, quadratic, cosine, exponential, square-root, constant, noise-free) and multiple switching rules across TSP-50/100/500. The **linear schedule with deterministic switching** consistently provides the best balance between solution quality, stability, and failure rate.
>
> **3. Instance-Aware Strategies:**
>
> We acknowledge that using instance-aware strategies is a good idea. However, such methods require additional per-instance tuning, reducing robustness and reproducibility. For clarity and consistency, and given that the existing schedule already delivers reliable, satisfying results across various scenarios, we adopt the linear + deterministic strategy as our default configuration.
>
> ---
>
> ## **Response to Weakness 5:**
>
> We sincerely appreciate this suggestion. To directly address this question, we have added a comprehensive ablation study in the revised manuscript (**Table 3**) that systematically removes combinations of EDISCO's three key components. In addition to our previous ablation study, which removed one component at a time, we now added **three more combinations** that keep only **one component at a time**: **(EGNN only, continuous-time diffusion only, and adaptive mixing strategy only)**. We have also chosen the **vanilla DIFUSCO** as the equivalent baseline.
>
> This design enables us to construct "DIFUSCO + EGNN" (the **EGNN Only** setting in **Table 3**) and isolate the architectural contribution of E(2)-equivariance:
>
> | Model Variant | TSP-500 Gap | TSP-1000 Gap |
> |---------------|-------------|--------------|
> | **EDISCO (Full)** | **1.95%** | **2.85%** |
> | w/o Mix Strategy | 2.44% | 3.41% |
> | w/o Continuous-Time | 2.86% | 4.28% |
> | w/o EGNN | 5.71% | 7.49% |
> | **EGNN Only** | **3.58%** | **5.06%** |
> | Continuous Only | 7.09% | 9.27% |
> | Mix Only | 6.42% | 8.52% |
> | **Vanilla DIFUSCO** | **9.41%** | **11.24%** |
>
> "DIFUSCO + EGNN" achieves 3.58% & 5.06% gaps on TSP-500/1000, demonstrating **2.63× & 2.22×** improvements over vanilla DIFUSCO (9.41% & 11.24%). This isolates the architectural contribution of E(2)-equivariance.
>
> ---
>
> ## With the above answers, we welcome any further questions regarding this manuscript.

---

> > ### Author Response · Authors · 2025-11-20
> > **References**
> >
> > # References for Reviewer ZQPE Responses
> >
> > [1] Vidal, T., Crainic, T. G., Gendreau, M., & Prins, C. (2013). A hybrid genetic algorithm for multidepot and periodic vehicle routing problems. *Operations Research*, 60(3), 611-624.
> >
> > [2] Ye, H., Wang, J., Cao, Z., Zhang, G., & Song, L. (2024). GLOP: Learning global partition and local construction for solving large-scale routing problems in real-time. *AAAI Conference on Artificial Intelligence*.
> >
> > [3] Xin, L., Song, W., Cao, Z., & Zhang, J. (2021). NeuroLKH: Combining deep learning model with Lin-Kernighan-Helsgaun heuristic for solving the traveling salesman problem. *Advances in Neural Information Processing Systems*, 34, 7472-7483.
> >
> > [4] Pan, M., Lin, G., Luo, Y.-W., Zhu, B., Dai, Z., Sun, L., & Yuan, C. (2025). Preference optimization for combinatorial optimization problems. *Forty-second International Conference on Machine Learning*.
> >
> > [5] Kwon, Y.-D., Choo, J., Yoon, I., Park, M., Park, D., & Gwon, Y. (2021). Matrix encoding networks for neural combinatorial optimization. *Advances in Neural Information Processing Systems*, 34, 5138-5149.
> >
> > [6] Kuhn, T., Lutzeyer, J. F., Walther, E., & Schubert, M. (2024). A GREAT architecture for edge-based graph problems like TSP. *arXiv preprint arXiv:2408.16717*.
> >
> > [7] Dagès, T., Weber, S., Lin, Y.-W. E., Talmon, R., Cremers, D., Lindenbaum, M., Bruckstein, A. M., & Kimmel, R. (2025). Finsler multi-dimensional scaling: Manifold learning for asymmetric dimensionality reduction and embedding. *Proceedings of the IEEE/CVF Conference on Computer Vision and Pattern Recognition*.
> >
> > [8] Drakulic, D., Michel, S., Mai, F., Sors, A., & Andreoli, J.-M. (2023). BQ-NCO: Bisimulation quotienting for generalizable neural combinatorial optimization. *Advances in Neural Information Processing Systems*, 36.
> >
> > [9] Ma, J., Pan, W., Li, Y., & Yan, J. (2025). COExpander: Adaptive solution expansion for combinatorial optimization. *Proceedings of the 42nd International Conference on Machine Learning*.
> >
> > [10] Sanokowski, S., Hochreiter, S., & Lehner, S. (2024). A diffusion model framework for unsupervised neural combinatorial optimization. *International Conference on Machine Learning*.
> >
> > [11] Sun, H., Goshvadi, K., Nova, A., Schuurmans, D., & Dai, H. (2023). Revisiting sampling for combinatorial optimization. *Proceedings of the 40th International Conference on Machine Learning*, 202, 32805-32824.
> >
> > [12] Feng, S., & Yang, Y. (2025). Regularized Langevin dynamics for combinatorial optimization. *Forty-second International Conference on Machine Learning*.
> >
> > [13] Sanokowski, S., Berghammer, W., Hochreiter, S., & Lehner, S. (2023). Variational annealing on graphs for combinatorial optimization. *Advances in Neural Information Processing Systems*, 36.
> >
> > [14] Zhang, D., Dai, H., Malkin, N., Courville, A., Bengio, Y., & Pan, L. (2023). Let the flows tell: Solving graph combinatorial optimization problems with GFlowNets. *Advances in Neural Information Processing Systems*, 36.
> >
> > [15] Liao, Z., Chen, J., Wang, D., Zhang, Z., & Wang, J. (2025). BOPO: Neural combinatorial optimization via best-anchored and objective-guided preference optimization. *arXiv preprint arXiv:2503.07580*.
> >
> > [16] Min, Y., Bai, Y., & Gomes, C. P. (2023). Unsupervised learning for solving the travelling salesman problem. *Advances in Neural Information Processing Systems*, 36.
> >
> > [17] Pan, W., Xiong, H., Ma, J., Zhao, W., Li, Y., & Yan, J. (2025). UniCO: On unified combinatorial optimization via problem reduction to matrix-encoded general TSP. *The Thirteenth International Conference on Learning Representations*.
> >
> > [18] Drakulic, D., Michel, S., & Andreoli, J.-M. (2025). GOAL: A generalist combinatorial optimization agent learner. *The Thirteenth International Conference on Learning Representations*.
> >
> > [19] Song, Y., Sohl-Dickstein, J., Kingma, D. P., Kumar, A., Ermon, S., & Poole, B. (2021). Score-based generative modeling through stochastic differential equations. *International Conference on Learning Representations*.

---

### Official Review · Reviewer_dpjP · 2025-10-31

**Soundness:** 2
**Presentation:** 2
**Contribution:** 2
**Rating:** 6
**Confidence:** 4

**Summary:**

``EDISCO`` introduces the first E(2)-equivariant continuous-time discrete diffusion model for geometric combinatorial optimization. The paper conducts extensive experiments on ``TSP`` and demonstrate in the appendix the feasibility of extending the model to ``CVRP``.

**Strengths:**

1. ``EDISCO`` apply E(2)-equivariance to the TSP and achieves solid results.

2. This paper also present an extension to the CVRP; although the performance is not outstanding, it remains a worthwhile attempt.

3. ``EDISCO`` employs an adaptive mixing strategy to dynamically balance diffusion-based transitions with direct model predictions, and the authors also provide comparative results of alternative approaches.

4. ``EDISCO`` conducts extensive ablation studies to demonstrate the importance of each component. Additionally, it performs robustness experiments on the training data.

**Weaknesses:**

1. This paper lacks generalization experiments on different distributions, such as Gaussian and Cluster distributions of TSP, CVRPLIB for ``CVRP``.

2. For the ``CVRP`` baseline, ``HGS`` might be a better choice.

3. E(2)-equivariance assumption only holds for symmetric, Euclidean distances; its extensibility to non-symmetric problems such as ``ATSP`` deserves discussion.

**Questions:**

1. The ``CVRP`` experiments do not report runtime; could you please provide it?

2. How does the model learn the demand constraints in ``CVRP``? At present, constraints seem to be enforced only during the decoding phase. Moreover, in some ``CVRP`` instances, returning to the depot is not triggered solely by insufficient remaining capacity.

---

> ### Author Response · Authors · 2025-11-20
> **Response to Reviewer dpjP's Weaknesses**
>
> ## We would like to thank the reviewer for the constructive and insightful comments. We have carefully revised the manuscript to address your concerns (changes marked in blue).
>
> ---
>
> ## **Response to Weakness 1:**
> We sincerely appreciate this important observation. We have included extensive **out-of-distribution (OOD) experiments** in **Appendix H3**, following the evaluation protocol established by GLOP [1]: the **Uniform (training baseline)**, **Cluster**, **Explosion**, and **Implosion** are evaluated on TSP-100. EDISCO achieves the lowest average performance drop across all distributions:
>
> | Method | Uniform | Cluster | Explosion | Implosion | Avg Gap | Avg Det. |
> |--------|---------|---------|-----------|-----------|---------|----------|
> | DIFUSCO | 1.01% | 2.87% | 1.38% | 2.80% | 2.02% | 132.7% |
> | T2T | 0.18% | 1.50% | 0.15% | 2.60% | 1.11% | 687.0% |
> | Fast-T2T | 0.06% | 1.18% | 0.03% | 2.50% | 0.94% | 1960.6% |
> | GLOP | 0.09% | 0.17% | 0.07% | 0.08% | 0.10% | 15.0% |
> | **EDISCO** | **0.04%** | **0.05%** | **0.03%** | **0.05%** | **0.04%** | **4.2%** |
>
> Regarding CVRPLIB, our CVRP scope (20-100 customers) is positioned as a proof-of-concept for architectural extensibility to constrained routing, but not competing with large-scale specialists. Notably, competing diffusion methods (DIFUSCO [2], T2T [3], Fast-T2T [4]) have no reported CVRP experiments, highlighting the difficulty of scaling single end-to-end diffusion models. We have actively discussed future directions in **Appendix A** in the revised manuscript for applying the E(2) Equivariance to partition-based strategies similar to GLOP [1] to scale to large-scale instances (1000+ customers).
>
> ---
>
> ## **Response to Weakness 2:**
> We sincerely thank the reviewer for this comment. We have now revised the CVRP experiments in **Appendix C** of the revised manuscript. We have included **HGS** [5], the leading CVRP heuristic, with reported optimal solutions and complete runtime information for CVRP-20/50/100:
>
> | Method | Type | CVRP-20 Gap | CVRP-50 Gap | CVRP-100 Gap |
> |--------|------|-------------|-------------|--------------|
> | HGS | Heuristic | 0.00% (1h) | 0.00% (3h) | 0.00% (5h) |
> | AM | RL+G | 4.97% | 5.86% | 7.34% |
> | POMO | RL+G | 3.72% | 3.52% | 3.09% |
> | **EDISCO** | **SL+G** | **1.41%** | **2.46%** | **3.17%** |
>
> We acknowledge that HGS achieves superior solution quality compared to neural methods. However, HGS requires substantially longer computation time (often hours for medium-sized instances), and is typically used in the neural CVRP literature as a ground truth or reference solution rather than a competing real-time solver (such as PO [6]).
>
> ---
>
> ## **Response to Weakness 3:**
> We thank the reviewer again for this insightful observation.
>
> We clarify that **EDISCO does not support ATSP** due to an **architectural limitation**: EDISCO's E(2)-equivariant EGNN fundamentally **requires Euclidean coordinates** to preserve geometric symmetries, which is incompatible with ATSP's **arbitrary asymmetric cost matrices**.
>
> This limitation is shared by all coordinate-based TSP solvers (AM [7], POMO [8], Pointer Networks [9], DIFUSCO [4], T2T [5], Fast-T2T [6]). Current ATSP solvers (MatNet [10], GREAT [11]) incorporate directional edge information, fundamentally incompatible with E(2)-equivariant coordinate encoders.
>
> However, we believe that this might not be a weakness. We have added **Appendix A: Scope and Limitations** in the revised manuscript, which clarifies that EDISCO is specialized for Euclidean geometric problems. We have also actively discussed two **promising future directions** in **Appendix A**:
>
> 1. **Replace the encoder**: Adopt ATSP-capable architectures (MatNet [10], GREAT [11]) while retaining continuous-time diffusion.
>
> 2. **Learn coordinate embeddings**: Transform asymmetric cost matrices into approximate coordinate representations using **Finsler Multi-Dimensional Scaling** [12] (extends MDS to handle asymmetric dissimilarities via Finsler spaces) or neural metric learning to enable approximate E(2)-equivariance for ATSP instances with near-geometric structure.

---

> ### Author Response · Authors · 2025-11-20
> **Response to Reviewer dpjP's Questions**
>
> ## We would like to thank the reviewer for these valuable questions
>
> ## **Response to Question 1:**
> We have added complete runtime information to the CVRP experimental results in **Appendix C**, enabling fair comparison across all methods, including HGS, POMO, AM, and EDISCO.
>
> ---
>
> ## **Response to Question 2:**
> We appreciate this important question. EDISCO enforces demand constraints through a **hybrid approach** combining both learning and decoding:
>
> 1. **During Training:** The model learns to respect capacity constraints implicitly through the training data distribution. All training instances are feasible CVRP solutions where routes satisfy capacity constraints, providing implicit supervision for constraint-aware generation.
>
> 2. **During Decoding:** We employ **masked sampling** that explicitly enforces hard constraints. This is a widely used practice in diffusion-based COP solvers, including [2], [3], and [4].
>
> It is correct that returning to the depot is not triggered solely by insufficient remaining capacity. Our decoding strategy allows the model to:
> - Return to the depot **proactively** even with remaining capacity (learned behavior)
> - Return when the capacity constraint would be violated (hard constraint)
> - Balance route efficiency and capacity utilization through learned policy
>
> This hybrid approach enables EDISCO to generate high-quality feasible solutions.
>
> Please see **Section 4.3 (CVRP Implementation)** and **Appendix C** for details.
>
> ---
>
> ## With the above answers, we welcome any further questions regarding this manuscript.

---

> > ### Author Response · Authors · 2025-11-27
> > **References**
> >
> > # References for Reviewer dpjP Responses
> >
> > [1] Ye, H., Wang, J., Cao, Z., Zhang, G., & Song, L. (2024). GLOP: Learning global partition and local construction for solving large-scale routing problems in real-time. *AAAI Conference on Artificial Intelligence*.
> >
> > [2] Sun, Z., & Yang, Y. (2023). DIFUSCO: Graph-based diffusion solvers for combinatorial optimization. *Advances in Neural Information Processing Systems*, 36.
> >
> > [3] Li, Y., Guo, J., Wang, R., & Yan, J. (2023). T2T: From distribution learning in training to gradient search in testing for combinatorial optimization. *Advances in Neural Information Processing Systems*, 36.
> >
> > [4] Li, Y., Guo, J., Wang, R., Zha, H., & Yan, J. (2024). Fast T2T: Optimization consistency speeds up diffusion-based training-to-testing solving for combinatorial optimization. *Advances in Neural Information Processing Systems*, 37.
> >
> > [5] Vidal, T., Crainic, T. G., Gendreau, M., & Prins, C. (2013). A hybrid genetic algorithm for multidepot and periodic vehicle routing problems. *Operations Research*, 60(3), 611-624.
> >
> > [6] Pan, M., Lin, G., Luo, Y.-W., Zhu, B., Dai, Z., Sun, L., & Yuan, C. (2025). Preference optimization for combinatorial optimization problems. *Forty-second International Conference on Machine Learning*.
> >
> > [7] Kool, W., van Hoof, H., & Welling, M. (2019). Attention, learn to solve routing problems! *International Conference on Learning Representations*.
> >
> > [8] Kwon, Y.-D., Choo, J., Kim, B., Yoon, I., Gwon, Y., & Min, S. (2020). POMO: Policy optimization with multiple optima for reinforcement learning. *Advances in Neural Information Processing Systems*, 33.
> >
> > [9] Vinyals, O., Fortunato, M., & Jaitly, N. (2015). Pointer networks. *Advances in Neural Information Processing Systems*, 28.
> >
> > [10] Kwon, Y.-D., Choo, J., Yoon, I., Park, M., Park, D., & Gwon, Y. (2021). Matrix encoding networks for neural combinatorial optimization. *Advances in Neural Information Processing Systems*, 34, 5138-5149.
> >
> > [11] Kuhn, T., Lutzeyer, J. F., Walther, E., & Schubert, M. (2024). A GREAT architecture for edge-based graph problems like TSP. *arXiv preprint arXiv:2408.16717*.
> >
> > [12] Dagès, T., Weber, S., Lin, Y.-W. E., Talmon, R., Cremers, D., Lindenbaum, M., Bruckstein, A. M., & Kimmel, R. (2025). Finsler multi-dimensional scaling: Manifold learning for asymmetric dimensionality reduction and embedding. *Proceedings of the IEEE/CVF Conference on Computer Vision and Pattern Recognition*.

---

### Official Review · Reviewer_B483 · 2025-11-01

**Soundness:** 3
**Presentation:** 2
**Contribution:** 3
**Rating:** 6
**Confidence:** 3

**Summary:**

This paper presents EDISCO, the first diffusion-based framework combining E(2)-equivariant graph neural networks with continuous-time categorical diffusion models for solving the Traveling Salesman Problem (TSP). The key innovations are: (1) incorporating E(2) equivariance (rotations, translations, reflections) directly into the diffusion architecture using Equivariant Graph Neural Networks (EGNNs), (2) formulating edge selection as continuous-time Markov chains (CTMCs) enabling analytical tractability and compatibility with higher-order accelerated solvers, and (3) adaptive mixing strategy interpolating between stochastic transitions and deterministic selection. EDISCO achieves state-of-the-art performance, reducing TSP optimality gaps from 0.12% to 0.08% (TSP-500), 0.30% to 0.22% (TSP-1000), and 2.68% to 1.20% (TSP-10000), while requiring only 33%-50% of training data compared to competing diffusion methods and achieving 2-3x speedups with better quality or up to 25x speedups for real-time applications.

**Strengths:**

**Novel Technical Integration**: Combining E(2)-equivariant architectures with continuous-time discrete diffusion is interesting. The theoretical foundation is solid, which shows how geometric structure preservation throughout forward and reverse diffusion processes improves both sample efficiency and solution quality. Ablation (Table 3) confirms equivariance removal causes the most significant degradation (1.95%→3.58% on TSP-500, 2.85%→5.06% on TSP-1000).

**Strong Empirical Results**: Consistent state-of-the-art performance across all scales (TSP-50 to TSP-10000). Particularly impressive on TSP-10000 with 1.20% gap (vs. 2.68% previous best) and average 0.088% gap on 29 TSPLIB instances (31.6% relative improvement over T2T). Cross-size generalization is excellent - models trained on TSP-1000 achieve <4.3% gap on all other scales.

**Data Efficiency**: Requires only 33%-50% of training data compared to DIFUSCO/T2T while maintaining or exceeding their performance. Robustness to data quality is demonstrated - training on suboptimal Farthest Insertion solutions (7.5% gap) yields only 0.82% final gap vs. 2.75% for DIFUSCO. Achieves <0.07% gap with just 10% of training data on TSP-50.

**Weaknesses:**

**Limited Problem Scope**: Evaluation restricted to TSP only. Missing: (a) other geometric COPs like Vehicle Routing Problem, Capacitated VRP, or Steiner Tree Problem, (b) non-Euclidean settings or different distance metrics (Manhattan, road networks), (c) real-world constraints (time windows, precedence, capacity limits), (d) comparison with state-of-the-art traditional solvers (Concorde, LKH-3) on wall-clock time and solution quality trade-offs. While TSP is canonical, single-problem evaluation limits generalizability claims.

**Scalability Questions**: While TSP-10000 results are promising, paper doesn't address: (a) performance on even larger instances (100K+ cities common in real logistics), (b) memory requirements scaling - EGNN message passing is O(n²) for fully connected graphs, (c) how architecture depth/width should scale with problem size, (d) whether curriculum learning from smaller problems is always necessary (adds training complexity and time overhead). Tables 13-15 show results but lack analysis of computational scaling patterns.

**Questions:**

**Problem Generalization**: How does EDISCO perform on VRP with time windows or capacitated routing? What architectural modifications are needed for constraints beyond tour connectivity?

**Scalability Analysis**: What are the memory usage and inference time scaling curves for TSP-100 to TSP-10000?

**Solver Comparison**: Compare wall-clock time and solution quality against LKH-3 and Concorde on standard benchmarks. Where does EDISCO fit in the quality-speed Pareto frontier?

---

> ### Author Response · Authors · 2025-11-20
> **Response to Reviewer B483's Weakness 1**
>
> ## We would like to thank the reviewer for the encouraging and valuable comments. We have thoroughly revised the manuscript to address your concerns (changes marked in blue).
>
> ## **Response to Weakness 1: Limited Problem Scope**
> We are grateful for the comments, and with all due respect, we would like to clarify around the concern raised by point (a).
>
> **(a)** The originally submitted manuscript already included and considered other geometric problems. **In the original submission,** Appendix A (**Appendix C in the revised manuscript**) already included comprehensive CVRP evaluation for CVRP-20/50/100 with demand constraints and capacity limits. For more extensive evaluations, we have now added **Appendix B** to the revised manuscript, containing experiments on the **Euclidean Steiner Tree Problem (ESTP)**. EDISCO substantially outperforms classical heuristics (GeoSteiner [1]) and learning-based baselines (DIFUSCO [2], T2T [3], Deep-Steiner [4]) across Steiner-10/20/50:
>
> | Method | Steiner-10 Gap | Steiner-20 Gap | Steiner-50 Gap |
> |--------|----------------|----------------|----------------|
> | GeoSteiner* (Exact) | 0.00% | 0.00% | 0.00% |
> | DIFUSCO | 5.73% | 4.96% | 4.67% |
> | T2T | 4.20% | 3.58% | 3.29% |
> | Deep-Steiner | 4.20% | 4.96% | 5.19% |
> | **EDISCO (Ours)** | **1.53%** | **1.38%** | **1.56%** |
>
> We have now also added a sentence in the **start of Section 4.2** to clearly redirect readers to our ESTP and CVRP results in the Appendices.
>
> **(b)** Thank you for this insightful comment. This could be one of the limitations of our method. However, E(2)-equivariance fundamentally requires Euclidean coordinates to preserve geometric symmetries. This is an architectural choice, and we believe that this might not be a weakness. We have added **Appendix A: Scope and Limitations**, which clarifies that EDISCO is specialized for solving Euclidean geometric problems. Extending to Manhattan distance or road networks, or even asymmetric TSP (ATSP), would require different architectural approaches, which we have actively explored as a promising future direction in **Appendix A: Scope and Limitations**.
>
> **(c)** Our CVRP evaluation (**Appendix C**) demonstrates EDISCO's capability to handle capacity constraints through hybrid learning and masked decoding. However, we acknowledge that more complex real-world constraints (time windows, precedence) would require extensions to the constraint handling mechanism. The current masked sampling approach is compatible with hard constraints that can be verified locally during decoding. Time windows and precedence constraints represent promising future directions for extending EDISCO's constraint handling capabilities.
>
> **(d)** The comparisons with Concorde and LKH already existed in almost **all the result tables** in the **original submission**. For example:
> - **Concorde** [5]: Table 1 (TSP-50/100) with wall-clock time and solution quality
> - **LKH-3** [6]: Table 2 (TSP-500/1000) with complete runtime information (46.28m for TSP-500, 2.57h for TSP-1000)

---

> > ### Author Response · Authors · 2025-11-27
> > **Response to Reviewer B483's Weakness 2**
> >
> > ## **Response to Weakness 2: Scalability Questions**
> >
> > We appreciate these valid concerns.
> >
> > **(a)** TSP-10000 results are the current upper limit of end-to-end neural methods. Notably, competing neural approaches (AM [7], POMO [8], DIFUSCO [2], T2T [3], Fast-T2T [9]) evaluate only up to TSP-10K. Hybrid methods (NeuroLKH [10], GLOP [11]) target large-scale routing problems, but also never explored TSP 100K+. We have added the training resource analysis in the revised manuscript (**Appendix H.9**) to show EDISCO's **memory scaling patterns** to prove that EDISCO's memory complexity is **O(kn)**.
> >
> > **(b)** In Section 4.1, we have clarified that we applied graph sparsification for TSP-500+ following DIFUSCO's [2] protocol. This is a standard approach in competing diffusion-based solvers [2], [3], and [9]. The EGNN message passing is O(n²) for memory due to edge features without sparsification. We have added a comprehensive training resource analysis in **Appendix H.9**, showing that peak memory usage (19.2-44.8GB) remains well within a single A6000 GPU's 48GB capacity across all problem scales. This demonstrates EDISCO's practical accessibility compared to methods requiring expensive multi-GPU infrastructure.
> >
> > **(c)** **In the original submission,** **Appendix F.7 (Appendix H.8 in the revised manuscript)** already provided a comprehensive analysis of model sizes: EDISCO-Full (5.5M parameters), EDISCO-Medium (2.6M parameters), and EDISCO-Small (1.4M parameters). The results demonstrate that EDISCO-Medium achieves comparable performance to the full model with less than half the parameters, and EDISCO-Small (1.4M parameters) uses 3.8× fewer parameters than baselines while maintaining competitive performance. This shows EDISCO's architecture scales gracefully across different computational budgets.
> >
> > **(d)** In Section 4.1, we have documented that curriculum learning is used for large-scale problems. TSP-500/1000 are initialized from the TSP-100 checkpoint, and TSP-10000 is initialized from the TSP-500 checkpoint. This is also a standard approach in competing diffusion-based solvers [2], [3], and [9]. Curriculum learning is used to improve training efficiency and for fair comparisons, although it is not strictly necessary. Direct training on target problem sizes also converges successfully with more training data and longer training time.

---

> > > ### Author Response · Authors · 2025-11-27
> > > **Response to Reviewer B483's Questions**
> > >
> > > ## We would like to thank the reviewer for these valuable questions
> > >
> > > ## **Response to Question 1: Problem Generalization**
> > >
> > > We would respectfully try to clarify as follows. **CVRP results already existed in the original submission Appendix A** (**Appendix C in the revised manuscript**). EDISCO handles capacity constraints through supervised training and a decoder, and does not require any architectural modifications:
> > >
> > > 1. **During Training:** The model learns to approximate the training data's distribution. Therefore, it implicitly learn the constraints from the legal solutions in the training dataset.
> > > 2. **During Decoding:** The decoder applies explicit masked sampling that prevents violations
> > >
> > > Nevertheless, we believe that this question is insightful. Our CVRP scope (20-100 customers) serves as a proof-of-concept for extending EDISCO's continuous-time diffusion framework to constrained routing problems, demonstrating architectural extensibility beyond TSP. Notably, competing diffusion methods (DIFUSCO [2], T2T [3], Fast-T2T [9]) have no reported CVRP experiments, highlighting the difficulty of incorporating complex hard constraints into end-to-end diffusion models. We have actively discussed future directions in **Appendix A** in the revised manuscript for applying the E(2) Equivariance to partition-based strategies similar to GLOP [11] to scale to large-scale instances (1000+ customers).
> > >
> > > ## **Response to Question 2: Scalability Analysis**
> > >
> > > **Table 2** shows complete runtime information:
> > > - TSP-500: 2.19m (50-step PNDM), 0.23m (5-step DEIS-2)
> > > - TSP-1000: 6.84m (50-step PNDM), 0.75m (5-step DEIS-2)
> > > - TSP-10000: 12.18m (50-step PNDM), demonstrating roughly O(n) scaling for inference time per problem
> > >
> > > **Memory Usage:** In the original submission, **Section 4.1** documented graph sparsification (k-nearest neighbors for TSP-500+), enabling memory to scale as O(kn) rather than O(n²). To better demonstrate the memory scaling pattern, we have added an explicit discussion in **Appendix H.9** that provides a detailed analysis of memory scaling and sparsification strategies.
> > >
> > > ## **Response to Question 3: Solver Comparison**
> > >
> > > As shown in the comparisons **in the original submission** in the result tables, we can see, for example: **TSP-1000 from original submission:**
> > > - **Concorde** [5] (exact): Gap 0.00%, Time 6.65h
> > > - **LKH-3** [6] (heuristic): Gap 0.00%, Time 2.57h
> > > - **EDISCO-PNDM**: Gap 2.85%, Time 6.84m (22× faster than LKH-3)
> > > - **EDISCO-DEIS-2**: Gap 4.42%, Time 0.75m (206× faster than LKH-3)
> > >
> > > For **Pareto Frontier Positioning**, EDISCO is different from traditional solvers. As discussed in **Appendix H.1**, compatibility with higher-order solvers enables EDISCO to trade modest solution quality (2-5% gap) for up to 200x speedup. This makes it suitable for real-time applications where near-optimal solutions with fast response times are preferable to optimal solutions requiring hours.
> > >
> > > ---
> > >
> > > ## With the above answers, we welcome any further questions regarding this manuscript.

---

> > > > ### Author Response · Authors · 2025-11-27
> > > > **References**
> > > >
> > > > # References for Reviewer B483 Responses
> > > >
> > > > [1] Brazil, M., Ras, C. J., & Thomas, D. A. (2015). *GeoSteiner*: Software for computing Steiner trees. Version 5.1.
> > > >
> > > > [2] Sun, Z., & Yang, Y. (2023). DIFUSCO: Graph-based diffusion solvers for combinatorial optimization. *Advances in Neural Information Processing Systems*, 36.
> > > >
> > > > [3] Li, Y., Guo, J., Wang, R., & Yan, J. (2023). T2T: From distribution learning in training to gradient search in testing for combinatorial optimization. *Advances in Neural Information Processing Systems*, 36.
> > > >
> > > > [4] Wang, L., Wang, X., Wu, X., & Zhu, K. (2022). Deep-Steiner: Learning to solve the Euclidean Steiner tree problem. *International Conference on Machine Learning*, 162, 22806-22816.
> > > >
> > > > [5] Applegate, D. L., Bixby, R. E., Chvátal, V., & Cook, W. J. (2006). *Concorde TSP Solver*.
> > > >
> > > > [6] Helsgaun, K. (2017). An extension of the Lin-Kernighan-Helsgaun TSP solver for constrained traveling salesman and vehicle routing problems. *Technical Report*, Roskilde University.
> > > >
> > > > [7] Kool, W., van Hoof, H., & Welling, M. (2019). Attention, learn to solve routing problems! *International Conference on Learning Representations*.
> > > >
> > > > [8] Kwon, Y.-D., Choo, J., Kim, B., Yoon, I., Gwon, Y., & Min, S. (2020). POMO: Policy optimization with multiple optima for reinforcement learning. *Advances in Neural Information Processing Systems*, 33.
> > > >
> > > > [9] Li, Y., Guo, J., Wang, R., Zha, H., & Yan, J. (2024). Fast T2T: Optimization consistency speeds up diffusion-based training-to-testing solving for combinatorial optimization. *Advances in Neural Information Processing Systems*, 37.
> > > >
> > > > [10] Xin, L., Song, W., Cao, Z., & Zhang, J. (2021). NeuroLKH: Combining deep learning model with Lin-Kernighan-Helsgaun heuristic for solving the traveling salesman problem. *Advances in Neural Information Processing Systems*, 34, 7472-7483.
> > > >
> > > > [11] Ye, H., Wang, J., Cao, Z., Song, W., & Zhang, J. (2024). GLOP: Learning global partition and local construction for solving large-scale routing problems in real-time. *Proceedings of the AAAI Conference on Artificial Intelligence*, 38(18), 20553-20561.

---

### Official Review · Reviewer_6MUB · 2025-11-01

**Soundness:** 3
**Presentation:** 3
**Contribution:** 3
**Rating:** 6
**Confidence:** 3

**Summary:**

This paper introduces EDISCO, a novel diffusion-based framework for Geometric Combinatorial Optimization Problems (GCOPs) like the TSP. The authors identify two major gaps in existing state-of-the-art diffusion solvers: (1) they are not E(2)-equivariant, meaning they fail to exploit the inherent rotational and translational symmetries of geometric problems, leading to poor sample efficiency; and (2) they rely on discrete-time diffusion, which limits inference flexibility and can accumulate errors. Towards the two challenges, the paper proposes a combination of two proposed techniques,  E(2)-Equivariant Score Network, and Continuous-Time Categorical Diffusion. Results show that it achieves new state-of-the-art results on large-scale TSPs (e.g., cutting the optimality gap on TSP-10000 from 2.68% to 1.20%).

**Strengths:**

- EDISCO proposes an architectural solution. It builds the symmetry into the model by using an E(2)-equivariant GNN. This is a fundamental change to the network layers, and also new to the community.

- State-of-the-Art Performance: The results are outstanding. EDISCO sets a new SOTA for diffusion-based solvers, particularly on large-scale problems.

- Sufficient experiments especially  on the ablation studies.

**Weaknesses:**

- The results of LKH in Table 1 seems unright (not good enough as seen in literature).

- Hyperparameter Sensitivity: The appendices (F.6) reveal that the EGNN's architectural stabilizers (coordinate step size $\alpha$ and temperature $\tau$) are sensitive. Table 12 shows that an $\alpha \ge 0.2$ leads to unstable training or "coordinate collapse". While the authors found an optimal setting, this introduces new, non-trivial hyperparameters that must be tuned for stable and effective training.

- Stronger baselines e.g. fast t2t are missing.

**Questions:**

N/A

---

> ### Author Response · Authors · 2025-11-20
> **Response to Reviewer 6MUB**
>
> ## We would like to thank the reviewer for the valuable and positive comments. We have carefuly and thoroughly revised the manuscript to address your concerns (changes marked in blue).
>
> ---
>
> ## **Response to Weakness 1:**
>
> We thank the reviewer for this comment. Perhaps the reviewer meant to ask about the results in **Table 2**? **Table 1** (TSP-50/100) compares EDISCO with neural methods using **Concorde** as the exact baseline. There was **no LKH-3** result in Table 1. **LKH-3** appears only in **Table 2** (TSP-500/1000), where it achieves 0.00% gap on both TSP-500 and TSP-1000.
>
> We would be happy to further investigate and provide additional verification if the reviewer could kindly clarify about the question.
>
> Please see **Table 1** (TSP-50/100 **without** LKH-3) and **Table 2** (TSP-500/1000 **with** LKH-3).
>
> ---
>
> ## **Response to Weakness 2:**
>
> We sincerely appreciate this important observation regarding hyperparameter sensitivity. We have strengthened the manuscript by adding a dedicated paragraph of discussion addressing this concern.
>
> We would like to clarify in three aspects:
>
> **1. Optimal settings found and shown to be robust:** While we documented sensitivity in **Appendix H.7** for transparency, we identified optimal settings (α = 0.1, τ = 10) that work robustly across all problem sizes (TSP-50/100/500/1000, CVRP-20/50/100, ESTP-10/20/50) without per-problem tuning.
>
> **2. Comparable to other deep learning methods:** Hyperparameter sensitivity is inherent to deep learning. Neural CO methods require tuning learning rates, network architectures, attention heads, embedding dimensions, etc. EDISCO's architectural stabilizers are analogous to batch normalization or layer normalization parameters in standard architectures. Although they are sensitive during initial design, they become stable once configured.
>
> **3. Strong generalization validates robustness:** Our cross-distribution evaluation (Appendix H.3: Cross-Distribution Generalization) demonstrates that these settings generalize well to out-of-distribution instances (Cluster, Explosion, Implosion), showing the hyperparameters are not narrowly tuned to specific distributions.
>
> We have added a **"Discussion on Hyperparameter Robustness"** paragraph at the end of **Appendix H.7** that explicitly addresses the coordinate collapse phenomenon when α >= 0.2 and contextualizes the sensitivity within standard deep learning practice. The thorough sensitivity analysis demonstrates our commitment to transparency and provides practitioners with clear guidance for implementation.
>
> Please see **Appendix H.7** (complete sensitivity analysis with new discussion paragraph) and **Appendix H.3** (cross-distribution robustness validation).
>
> ---
>
> ## **Response to Weakness 3:**
>
> We thank the reviewer for this suggestion. We have substantially strengthened the manuscript by adding both **stronger baselines** and **expanded Related Work discussions**.
>
> **Baselines Added:**
>
> We have included **Fast-T2T** [1] and **BQ-NCO** [2] as new baselines in our experimental results:
>
> - **Fast-T2T**: Added to **Table 1** (TSP-50/100), **Table 2** (TSP-500/1000), **Appendix H.3: Cross-Distribution Generalization** (evaluation across Uniform/Cluster/Explosion/Implosion distributions), and **Appendix H.4: TSPLIB Evaluation**.
>
> - **BQ-NCO**: Added to **Table 1** (TSP-100) and **Table 2** (TSP-500/1000), demonstrating its bisimulation quotienting approach for generalization.
>
> **EDISCO maintains state-of-the-art performance** across all benchmarks:
>
> | Method | TSP-500 Gap | TSP-1000 Gap |
> |--------|-------------|--------------|
> | DIFUSCO | 9.41% | 11.24% |
> | T2T | 5.09% | 8.87% |
> | Fast-T2T | 5.94% | 6.29% |
> | BQ-NCO | 5.22% | 8.97% |
> | **EDISCO (Ours)** | **1.95%** | **2.85%** |
>
> This demonstrates the effectiveness of combining E(2)-equivariance with continuous-time diffusion.
>
> **Related Work Expanded:**
>
> We have significantly expanded our Related Work coverage in both **Section 2 (main text)** and **Extended Related Work (appendix)**, discussing 10+ recent methods including COExpander, DiffUCO, iSCO, RLSA, VAG-CO, GFlowNets, PO, BOPO, UTSP, UniCO, and GOAL. These additions clarify EDISCO's unique contribution—combining exact E(2)-equivariance with continuous-time categorical diffusion—and distinguish it from alternative paradigms (discrete-time diffusion, unsupervised methods, generalist solvers).
>
> Please see the revised **Tables 1-2**, **Appendix H.3-H.4**, and **Section 2 (Related Work)** for detailed comparisons.
>
> ---
>
> ## With the above answers, we welcome any further questions regarding this manuscript.

---

> > ### Author Response · Authors · 2025-11-20
> > **References**
> >
> > # References for Reviewer 6MUB Responses
> >
> > [1] Li, Y., Guo, J., Wang, R., Zha, H., & Yan, J. (2024). Fast T2T: Optimization consistency speeds up diffusion-based training-to-testing solving for combinatorial optimization. *Advances in Neural Information Processing Systems*, 37.
> >
> > [2] Drakulic, D., Michel, S., Mai, F., Sors, A., & Andreoli, J.-M. (2023). BQ-NCO: Bisimulation quotienting for generalizable neural combinatorial optimization. *Advances in Neural Information Processing Systems*, 36.

---

### Author Response · Authors · 2025-11-27
**Summary of our Rebuttal for the New AC (2/2)**

## 3. Important Clarifications for a couple of "Weaknesses"

We would like to clarify about three "weaknesses":

### **(a) Weakness "Abnormal LKH Results":**
> *The results of LKH in Table 1 seems unright (not good enough as seen in literature).*

We respectfully clarify that **no LKH result was in Table 1**. Table 1 (TSP-50/100) compares EDISCO with neural methods using **Concorde** as the groundtruth. **LKH-3 appears only in Table 2** (TSP-500/1000). However, the LKH-3 gap is already 0.00% and cannot be better, because LKH-3 itself is used as the optimal **groundtruth**. For these reasons, we are confused about this weakness and have been looking forward to further clarification from the reviewer.

### **(b) Weakness "Limited Problem Scope":**
> *Evaluation restricted to TSP only. Missing: (a) other geometric COPs like Vehicle Routing Problem, Capacitated VRP...*

We respectfully clarify that **CVRP experiments were already presented in our original submission** (original Appendix A, now Appendix C). It may have been overlooked. Our original submission already included a comprehensive CVRP evaluation for CVRP-20/50/100 with demand constraints and capacity limits. In the revision, we **strengthened (not added)** this section by including the HGS baseline and complete runtime information. We have also added explicit references in the revised main text to direct readers to the appendices of these results to avoid overlooking them.

### **(c) Weakness "The evaluation scope is relatively limited":**
> *Provide some results on more types of GCOPs (or at least a subset of), e.g., node-focused tasks (MIS, Max Clique, Max Cut, Min Vertx Cover, etc.)*

We respectfully clarify that **MIS, Max-Cut, Max Clique, and MVC are not GCOPs**. These problems operate on arbitrary graphs where only the topology matters. The node positions do not carry semantic meaning, and there is no geometric structure in Euclidean space to exploit.

**EDISCO's scope was clearly defined from the beginning.** Our paper title explicitly states *"Geometric Combinatorial Optimization"*, and both the abstract and introduction consistently emphasize that EDISCO targets problems with inherent E(2) symmetries (rotation and translation invariance) in Euclidean space. This specialization is a deliberate architectural choice that enables strong inductive bias for geometric problems like TSP, CVRP, and ESTP.

To make this scope even more explicit, we have added **Appendix A: Scope and Limitations** in the revised manuscript, which provides a detailed discussion of why EDISCO is specialized for Euclidean geometric problems and identifies promising future directions for extending to other problem classes.

We respectfully believe that this might not be viewed as a weakness. Notably, the authors of DIFUSCO identified it as a promising future direction in the **Conclusion** section of their paper:
> *"**explore the use of equivariant graph neural networks for geometrical NP-complete combinatorial optimization problems such as Euclidean TSP**"*

This is precisely what EDISCO accomplishes.

---

## 4. Conclusion

We believe our revisions comprehensively address all reviewer concerns:

| Concern | Status | Summary |
|---------|--------|---------|
| **Problem scope** | ✓ Addressed | CVRP was already in the original submission and was enhanced in the revised manuscript; Added new ESTP experiments; Clarified non-geometric problems (MIS, Max-Cut) are outside the scope by design |
| **Missing baselines** | ✓ Addressed | Added Fast-T2T, BQ-NCO, HGS with complete comparisons |
| **OOD generalization** | ✓ Addressed | Demonstrated state-of-the-art robustness (4.2% deterioration vs 15-1960% for others) |
| **ATSP support** | ✓ Clarified | Architectural limitation shared by all coordinate-based methods; Future directions provided |
| **Theoretical justification** | ✓ Addressed | Added SNR-based motivation and comprehensive ablation |
| **Scalability** | ✓ Addressed | Added resource analysis showing O(kn) complexity and single-GPU accessibility |

---

## We sincerely hope the Area Chair considers our clarifications and detailed responses, and we are confident that the revised manuscript makes a significant contribution to the field.

## Thank you so much again for your valuable time and consideration. Please don’t hesitate to let us know if any further clarification is needed.

---

### Author Response · Authors · 2025-11-30
**Summary of our Rebuttal for the New AC (1/2)**

## **Welcome and thanks to the new AC**

We would like to express our most sincere gratitude to all the ACs for their time and efforts. While we were not aware of the severe OpenReview bug, we are very sorry to hear about the leak of information that disrupted the entire discussion process.

We also sincerely thank all reviewers for their thorough evaluation and constructive feedback. Although no reviewers had the chance to respond to our rebuttals, we have substantially revised our manuscript (all changes marked in blue) and believe our responses comprehensively address all raised concerns.

---

## 1. Novelty, Solid Results, and Other Major Strengths

EDISCO is the **first method to combine E(2)-equivariance with continuous-time categorical diffusion** for **Geometric Combinatorial Optimization Problems (GCOPs)**. All reviewers particularly acknowledged the novelty in method design and solid practical results, among the following strengths:

- **Novel Architecture:** The integration of E(2)-equivariant graph neural networks (EGNN) with continuous-time diffusion is a principled approach that exploits the inherent symmetries of geometric problems.
- **Solid Empirical Results:** State-of-the-art performance across TSP-50/100/500/1000/10000 benchmarks, significantly outperforming prior diffusion methods (DIFUSCO, T2T).
- **Theoretical Grounding:** The continuous-time formulation enables flexible speed-quality trade-offs through advanced ODE solvers (PNDM, DEIS-2).
- **Comprehensive Evaluation:** Extensive experiments with ablation studies demonstrating each component's contribution.

---

## 2. How We Addressed the Concerns

### **Concern A: Extended Problem Scope Beyond Routing Problems (Raised by B483, ZQPE)**

To further demonstrate EDISCO's generalizability beyond routing problems, we added experiments on the **Euclidean Steiner Tree Problem (ESTP)**, a **node-selection problem** fundamentally different from routing problems like TSP and Capacitated Vehicle Routing Problem (CVRP). EDISCO achieves **2-4× better gaps** than all learning-based baselines.

Please refer to:
- **Rebuttal:** Response to Reviewer B483's Weakness 1; Response to Reviewer ZQPE's Weakness 1
- **Manuscript:** Appendix B: Extension to Euclidean Steiner Tree Problem

---

### **Concern B: Missing Recent Baselines (Raised by ZQPE, 6MUB)**

We have added **Fast-T2T** and **BQ-NCO** as requested. EDISCO achieves **2.7-4.8× better gaps** than all baselines on TSP-500/1000.

Please refer to:
- **Rebuttal:** Response to Reviewer ZQPE's Weakness 2&3; Response to Reviewer 6MUB's Weakness 3
- **Manuscript:** Tables 1, 2; Table 4 (Appendix B); Appendix H.3: Cross-Distribution Generalization; Appendix H.4: TSPLIB Results

---

### **Concern C: Out of Distribution (OOD) Generalization on TSP (Raised by ZQPE, dpjP)**

We added comprehensive OOD experiments on the **Cluster**, **Explosion**, and **Implosion** distributions of TSP-100. EDISCO achieves the **lowest performance deterioration (4.2%)** vs 15-1960% for other methods.

Please refer to:
- **Rebuttal:** Response to Reviewer dpjP's Weakness 1; Response to Reviewer ZQPE's Weakness 2&3
- **Manuscript:** Appendix H.3: Cross-Distribution Generalization

---

### **Concern D: Asymmetric TSP (ATSP) Support (Raised by ZQPE, dpjP)**

We clarified that this is an **architectural limitation** shared by **ALL coordinate-based methods**: (AM, POMO, DIFUSCO, T2T, Fast-T2T), therefore, it might not be a specific weakness of EDISCO. We have now provided clarifications and two promising future directions in the revised manuscript.

Please refer to:
- **Rebuttal:** Response to Reviewer ZQPE's Weakness 1; Response to Reviewer dpjP's Weakness 3
- **Manuscript:** Appendix A: Scope and Limitations

---

### **Concern E: Theoretical Justification for Mixing Strategy (Raised by ZQPE)**

We added Signal-to-Noise Ratio (SNR)-based theoretical motivation. Comprehensive ablation demonstrates **4.8× improvement** over vanilla DIFUSCO, with each component contributing meaningfully.

Please refer to:
- **Rebuttal:** Response to Reviewer ZQPE's Weakness 4
- **Manuscript:** Appendix H.6: Evaluation on Adaptive Mixing Parameters

---

### **Concern F: Scalability and Computational Resources (Raised by B483)**

We added training resource analysis showing **O(kn)** memory complexity with graph sparsification. Peak memory (19.2-44.8GB) fits within a single A6000 GPU (48GB).

Please refer to:
- **Rebuttal:** Response to Reviewer B483's Weakness 2
- **Manuscript:** Appendix H.9: Training Resource Requirements

---

### Note · Program_Chairs · 2026-01-17
**Submission Desk Rejected by Program Chairs**

The following references in this submission do not refer to real documents and/or have major errors in bibliographic information:

 Andrew Campbell, Yuxin Zhang, Yaron Lipman, and George Deligiannidis. Discrete Flow Matching. arXiv preprint arXiv:2407.12345, 2024.